# ACES: AUTOMATIC COHORT EXTRACTION SYSTEM FOR EVENT-STREAM DATASETS

**Justin Xu**
University of Oxford
justin.xu@ndm.ox.ac.uk

**Jack Gallifant**
Massachusetts Institute of Technology
jgally@mit.edu

**Alistair E. W. Johnson**
University of Toronto
alistair.johnson@utoronto.ca

**Matthew B. A. McDermott**
Harvard Medical School
matthew_mcdermott@hms.harvard.edu

## ABSTRACT

Reproducibility remains a significant challenge in machine learning (ML) for healthcare. Datasets, model pipelines, and even task or cohort definitions are often private in this field, leading to a significant barrier in sharing, iterating, and understanding ML results on electronic health record (EHR) datasets. We address a significant part of this problem by introducing the Automatic Cohort Extraction System (ACES) for event-stream data. This library is designed to simultaneously simplify the development of tasks and cohorts for ML in healthcare and also enable their reproduction, both at an exact level for single datasets and at a conceptual level across datasets. To accomplish this, ACES provides: (1) a highly intuitive and expressive domain-specific configuration language for defining both dataset-specific concepts and dataset-agnostic inclusion or exclusion criteria, and (2) a pipeline to automatically extract patient records that meet these defined criteria from real-world data. ACES can be automatically applied to any dataset in either the Medical Event Data Standard (MEDS) or Event Stream GPT (ES-GPT) formats, or to *any* dataset in which the necessary task-specific predicates can be extracted in an event-stream form. ACES has the potential to significantly lower the barrier to entry for defining ML tasks in representation learning, redefine the way researchers interact with EHR datasets, and significantly improve the state of reproducibility for ML studies using this modality. ACES is available at: https://github.com/justin13601/aces.

## 1 INTRODUCTION

Machine learning (ML) for healthcare suffers from a severe and systemic reproducibility crisis (McDermott et al., 2021b). This challenge is further exacerbated by the need to maintain private and secure datasets, but even with public datasets, ML pipelines are not reliably reproducible from published papers alone. For instance, in numerous attempts to reproduce ML for healthcare studies using the MIMIC-III dataset (Johnson et al., 2016), Johnson et al. found that more than half the time, the cohorts described in the studies could not be reliably reconstructed. Specifically, experiments led to many discrepancies of up to 25% in cohort sizes, with one study reaching as high as 11,767 patients (Johnson et al., 2017). This is primarily due to sparse descriptions of cohorts in

study methods with essential reproducibility details often omitted, along with the absence of openly available code.

This burden in reproducing even the basic task and problem definitions in ML for healthcare studies is profoundly detrimental (McDermott, 2025). Beyond the obvious concerns it raises around the robustness of reported results and their readiness for deployment, our inability to reliably define shared, canonical, and reproducible task definitions limits our capacity to perform meaningful model comparisons during methodological development. This is particularly notable in settings where not all researchers have mutual access to all datasets, as is common in healthcare. Given the critical role that open benchmarks play in the advancement of ML methods (Zhang & Hardt, 2024; Salaudeen & Hardt, 2024; Shirali et al., 2023), this deficit directly translates to a significant barrier in our ability, as a research community, to effectively experiment, iterate, and develop new ML methodologies in the healthcare space.

Given the clear import of this problem, the research community has naturally explored a number of prospective solutions. These can largely be categorized into two areas: (1) leveraging existing common data models (CDMs) to define reproducible task cohorts only for datasets within these schemas, and (2) defining static benchmarking tasks on individual public datasets. Both of these areas have generated numerous high-impact works. For example, in the area of CDM-driven tools, systems such as the ATLAS tool (Gold et al., 2024) for OHDSI's OMOP CDM (Reich et al., 2024) and i2b2's PIC-SURE (Stedman et al., 2024) for the i2b2 CDM (Murphy et al., 2010), as well as various institution-specific tools, have all been used to drive numerous new lines of inquiry. Unfortunately, these tools are also extremely limited in that they can only be applied to the specific CDM or institutional data warehouse for which they have been defined. Further, because many of these CDMs have had limited penetrance into healthcare's high-capacity, deep learning ecosystems, they are particularly ill-posed for task and cohort extraction within the healthcare deep learning communities. Conversely, public static benchmarks (Harutyunyan et al., 2019; McDermott et al., 2021a; van de Water et al., 2024) over datasets such as MIMIC-IV (Johnson et al., 2023) or eICU (Pollard et al., 2018) have also been extremely impactful. However, they are all tied to only a single or small set of datasets and tasks. Given the highly dynamic nature of clinical data and healthcare requirements, this is insufficient for the benchmarking and reproducibility needs faced by the ML for healthcare community.

When considering these existing solutions alongside the realities of healthcare data access and methodological development, it is clear that they are insufficient for three key reasons:

**1. The Need for Interoperability** The limited public datasets and only partially used CDMs cannot capture the diverse clinical populations, needs, and model capacities necessary for tangible ML progress in healthcare. To address this, systems for automated task extraction must be *meaningfully interoperable* across both public and private datasets with diverse input schemas.

**2. The Need for Flexibility** A single, static benchmark cannot encompass the variety of clinical tasks relevant to clinicians and informaticians. As existing tools (with limited interfaces for defining queries using per-set vocabularies) may struggle to generalize to new clinical tasks, ideal solutions must be *sufficiently flexible* to accommodate a myriad of new task definitions, criteria formats, and disease or deployment areas.

**3. The Need for Accessibility, Usability, and Applicability in Deep Learning Workflows** While many existing tools feature no-code interfaces (*e.g.*, web platforms to build queries) that are essential for less technically-literate audiences, integrating such tools with deep learning workflows can prove challenging. Deep learning systems are often run in a semi-programmatic manner on siloed, private computational clusters where researchers have minimal control. Hence, existing tools can cause significant hindrance. Instead, successful software must be able to provide a Python and command-line interface (CLI) that offer *significant ease of use* to deep learning researchers, alongside shareable and readable configuration files that specify task definitions in a manner that can be *readily ported* across datasets and environments.

**Our Solution: Automatic Cohort Extraction System for Event-Stream Datasets** In this work, we solve these problems with the Automatic Cohort Extraction System for Event-Stream Datasets (ACES). ACES offers a simple, expressive, and shareable configuration language for task and cohort

definitions, as well as a reliable CLI and a straightforward Python library for extracting labeled task dataframes (Figure 2).

Task definitions in ACES are naturally separated into simple **dataset-specific** event *predicates* and **dataset-agnostic** inclusion or exclusion *criteria*, thereby permitting the same task to be used in a *conceptually identical* manner across diverse datasets. By requiring users to specify predicates to realize their ML tasks on their specific datasets, ACES allows users to produce precise, locally-specific, verifiable cohorts that harmonize only the data elements needed for their task, regardless of how their input dataset is aligned or misaligned with existing ontologies or CDMs. Further, for datasets that are fully harmonized (*e.g.*, through OHDSI vocabularies), ACES predicate definitions can be re-used across datasets without any loss of utility. In this way, ACES accommodates diverse datasets at various levels of data harmonization in a flexible, transparent manner. Overall, this approach not only enhances reproducibility but also facilitates community collaboration on task definitions, inclusion or exclusion criteria, and evaluation metrics for specific clinical use cases.

In contrast to prior task definition systems such as ATLAS, ACES makes minimal assumptions about the input data structure or source vocabularies. In particular, ACES can be run on *any* dataset, provided the necessary task-specific predicates can be pre-extracted in an "event-stream" format (Figure 1). It can further be run from raw data directly for any dataset in the relatively low-level and flexible Medical Event Data Standard (MEDS) (Arnrich et al., 2024) or Event Stream GPT (ESGPT) (McDermott et al., 2023) formats in approximately five lines of template code, offering high efficiency.

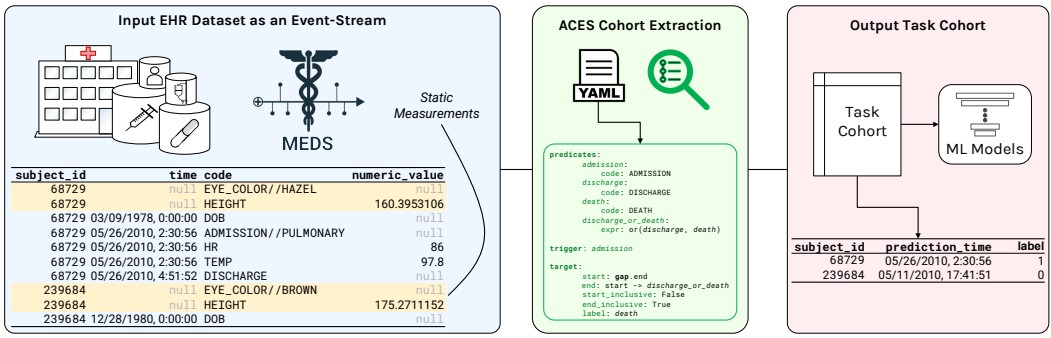

Figure 1: Workflow for extracting cohorts using ACES. The pipeline shows the expected format for ACES-supported event-stream datasets and outcome cohorts. The transformation of raw data into the event-stream format is intentionally designed to be straightforward — primarily merging relational database tables — minimizing data loss risks associated with other CDMs like OMOP.

Further, we align ACES with the concept of frictionless reproducibility for shared tasks proposed by David Donoho (Donoho, 2024), especially for the clinical domain. This addresses the "Bring-Your-Own-Data Challenge", where much research relies on private patient outcomes data, often accessible only to a few credentialed researchers under strict usage agreements. Even when benchmarking platforms and shared code exist, the inability to share data directly remains a significant barrier and often stifles progress. ACES seeks to overcome these challenges by offering a domain-specific language (DSL) and novel infrastructure to ensure reproducibility without necessitating data sharing. Instead of relying on public datasets or reconfiguring code for diverse environments, ACES enables researchers to distribute task definitions through configuration files. These files provide a standardized way to conceptually reproduce cohorts on private datasets or exactly reproduce them on public datasets.

In sum, ACES represents 3 key contributions:

1. ACES defines a shareable, simple, and flexible task configuration language that can define diverse sets of prediction tasks for ML in healthcare on any event-stream dataset.

2. ACES provides an easy-to-use library to automatically extract these tasks from diverse sources of real-world, structured, and longitudinal electronic health record (EHR) data.

3. ACES introduces a novel DSL that leverages event-bounded query aggregations, enabling more expressive and efficient task definitions that better capture clinical timelines.

In the rest of this work, we first explore ACES in depth in Section 2, beginning with an illustration of its key concepts before briefly overviewing its core recursive algorithm. We then present a running example to illustrate how configuration files capture task logic. Next, in Section 3, we demonstrate the use of ACES across diverse problem areas using real-world data, releasing a collection of task definitions based on prior ML for healthcare works — both at the dataset-agnostic criteria level and with dataset-specific predicates for the widely-used MIMIC-IV dataset (Johnson et al., 2023). Finally, we discuss the limitations and future roadmap of ACES in Section 4, and offer concluding thoughts in Section 5.

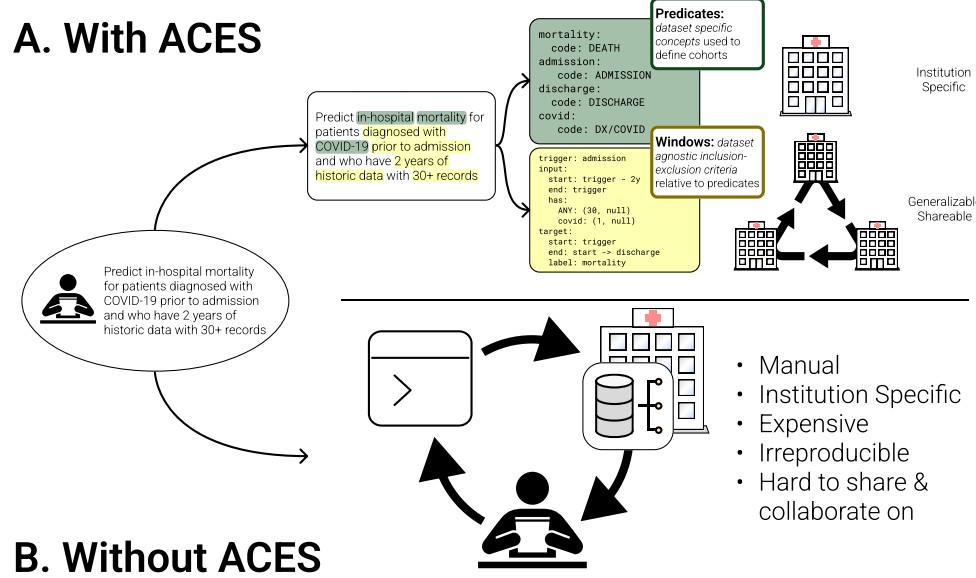

Figure 2: ML task cohort extraction process **(A)** with and **(B)** without ACES. Predicates are dataset-specific concepts that are needed to conceptually capture a ML task. Windows are temporal segments on a patient's health record and are dataset-agnostic, as they are defined relative to the predicates. This distinction allows researchers to easily share the more complex task logic which is independent of datasets, facilitating conceptual reproducibility for ML tasks in healthcare.

## 2 AUTOMATIC COHORT EXTRACTION SYSTEM FOR EVENT-STREAM DATASETS (ACES)

In this section, we introduce ACES, a novel automatic task and cohort extraction system that fills the key gaps in *interoperability*, *flexibility*, and *accessibility* left by the existing tools outlined in Section 1. To use ACES and extract a cohort for downstream ML tasks, a user only needs to do the following simple steps:

**1. Install ACES:** A fully functional version of ACES is pushed to PyPI, and any user can easily install it by simply running `pip install es-aces`. All dependencies are automatically set up with no further actions needed.

**2. Define Dataset:** A dataset in a permitted format, such as MEDS, ESGPT, or as direct predicates, is required. More information on the data formats is available in Section 2.2.

**3. Define Task:** A task configuration file is required to define the task that the user wishes to extract. This configuration language is simple, clear, yet flexible, permitting users to rapidly share and iterate over task definitions for their clinical settings. Configuration specification is given in Section 2.3.

**4. Run the ACES CLI:** ACES can be directly run from the command line. Details about the possible command-line arguments are detailed in Section 2.4. ACES can also be used as a direct Python import, as mentioned in Section 2.5.

Sample command for basic cohort extraction using ACES CLI:

```
$ aces-cli cohort_name='$TASK' data.path='$DATA_PATH'
```

**5. Get Outputs:** ACES outputs a single unified dataframe with all valid patient instances extracted according to task specifications. Users can subsequently leverage the returned columns with original patient identifiers and health record timestamps for downstream ML tasks. Further information is available in Section 2.6.

Critically, after *only these five simple steps*, a user can immediately, reproducibly extract a full cohort from their source dataset that matches their task definition, and begin using this task for downstream representation learning.

## 2.1 ALGORITHM DESIGN

ACES addresses the challenge of extracting meaningful windows of data from patient records by using a recursive approach grounded in a tree-structured configuration file. Each task is represented as a hierarchy of constraints, with nodes defining boundaries of windows of interest and edges specifying temporal or event-based relationships between these windows. The algorithm begins by identifying root anchor events in the dataset that correspond to the triggering criteria of the task. It then recursively evaluates subtrees of constraints, aggregating predicate counts over defined windows either through temporal aggregations (*e.g.*, over fixed time intervals) or event-bound aggregations (*e.g.*, over windows bounded by specific clinical events, such as admissions, diagnoses, etc.). Each step ensures that the criteria of the subtree are met, filtering out invalid realizations before proceeding to child nodes. This recursive process guarantees that the specified configuration can always be resolved into valid windows that meet the task's constraints. The final output is a dataframe containing all valid patients, task-specific labels, and prediction timestamps, and optionally, window start and end times as well as aggregated predicate counts. This ensures systematic and deterministic extraction of datasets for ML tasks. It also maintains flexibility and leverages the simple, transparent, and highly expressive DSL of ACES (Figure 3).

## 2.2 DATASET CONFIGURATIONS

ACES is extremely flexible and can handle different input data formats, including MEDS (`data.standard=meds`), ESGPT (`data.standard=esgpt`), or direct predicates (`data.standard=direct`), where event-stream features are pre-extracted by the user from any given dataset schema. Other CDMs are interchangeable with these formats, such as OHDSI OMOP through the MEDS OMOP ETL[1], which transforms OMOP-compliant datasets into MEDS without data loss or scalability issues.

Using direct predicates to extract cohorts from formats that ACES does not natively support still significantly reduces the burden on users. Simply creating predicate features is much less cumbersome than either fully converting the dataset to a CDM in order to use existing tools like ATLAS or i2b2's platform, or performing the entire task extraction from scratch by writing in-house dataframe querying code. Additionally, as ACES configuration files are shareable and easily portable to other datasets (by simply swapping out predicate definitions), we believe ACES will offer long-term efficiency benefits. This demonstrates the significant improvement in utility that ACES brings across diverse data schemas compared to existing tools.

## 2.3 TASK CONFIGURATIONS

In ACES, tasks are specified through configuration files that define a collection of dataset-specific event predicates, which are simple functions evaluated on individual events within a structured event-stream dataset. Predicate definitions can be stored in a central "database" file specific to each

---

[1]`https://github.com/Medical-Event-Data-Standard/meds_etl/tree/main`

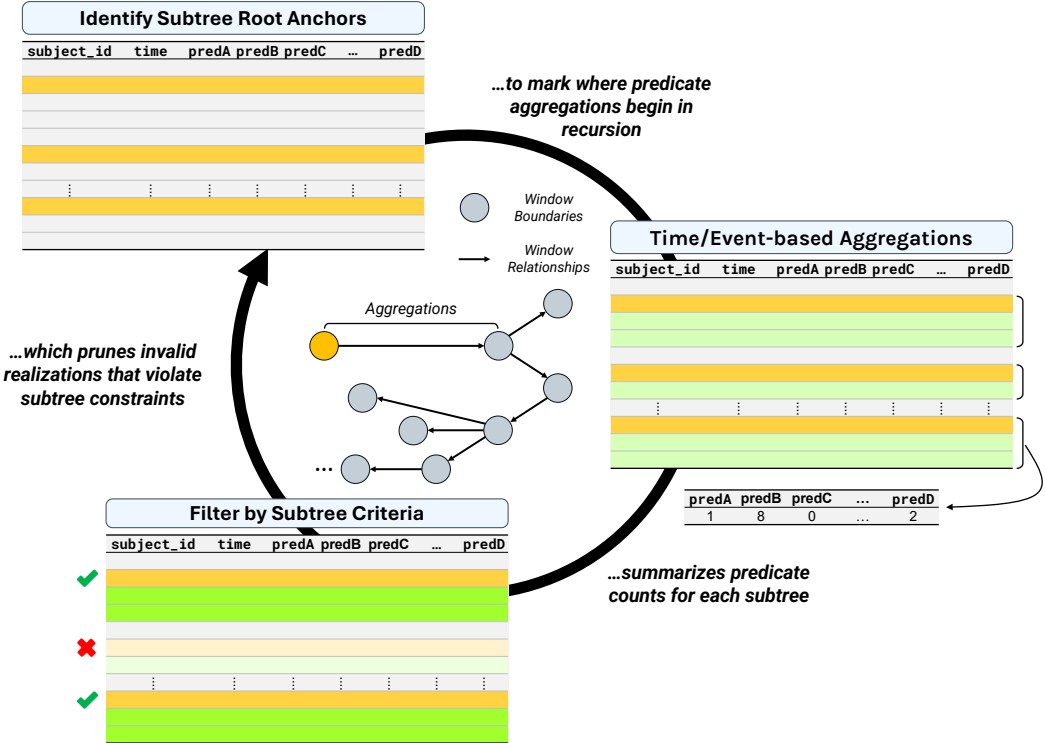

Figure 3: Overview of the ACES recursive algorithm. Given a task tree generated from a configuration file, ACES first identifies possible roots of the tree (task triggers) based on the associated predicate. It then computes aggregations of predicate counts over time-based (*i.e.*, windows with a time interval) or event-based (*i.e.*, windows between specified events) periods to summarize predicates over the edges between the tree nodes. Finally, invalid branches are filtered out if their predicate counts do not meet the specified criteria. This process is recursed for all child nodes of the task tree.

dataset, such that previously defined features could be easily reused for a variety of downstream tasks without further effort. Community predicate contributions for public datasets also streamline collaborative efforts for reproducibility. Additionally, task criteria are defined in a dataset-agnostic manner through a collection of interrelated windows, which specify segments of a patient's record and are constrained by certain relationships. Please see Figure 4 for an example of a task configuration.

## 2.4 COMMAND-LINE INTERFACE

**Hydra Arguments**    The Hydra framework (Yadan, 2019) enhances the CLI by enabling flexible run configurations and argument parsing for cohort extractions. For instance, specific arguments are required to define the external source dataset for data loading. Depending on the chosen format (the `data.standard` argument), either the path to the data file (for `meds` or `direct`) or the path to the dataset directory (for `esgpt`) must be specified to indicate the external source data from which ACES will extract the cohort. Additionally, `cohort_dir` and `cohort_name` are essential for locating and loading the task configuration file, as well as for results and operational logging.

**Scaling to Large Datasets**    An overview of the computational profile of ACES is available in Section 3.1. Additionally, for users dealing with large datasets, ACES can also be run over a collection of sharded files, extracting and storing the matching cohort for each shard individually in corresponding file paths. This can greatly increase computational efficiency by facilitating the processing of different shards in parallel via Hydra's multi-run launchers[2].

---

[2] `https://hydra.cc/docs/1.0/plugins/joblib_launcher/`

Sample command for cohort extraction over multiple MEDS shards using ACES CLI and Hydra:

```
$ aces-cli \
    --multirun \
    cohort_name="<task_config_name>" \
    cohort_dir="/directory/to/task/config/" \
    data=sharded \
    data.standard=meds \
    data.root="/directory/to/dataset/shards/" \
    data.shard="$(expand_shards <folder>/<num>)" # Sweeps over shards
```

## In-hospital Mortality Prediction

## Task Configuration

Figure 4: Example configuration file for the binary prediction of in-hospital mortality 48 hours after admission. References to predicates and windows are italicized and bolded, respectively. **(A)** Dataset-specific task predicates. These concepts are needed to conceptually capture this task and are used as constraints and boundaries for windows of the patient record. For instance, in this example, the value of "$ADMISSION$" denotes a hospital admission event in the source dataset. **(B)** A window of the task specifying the task inputs for downstream models. Suppose we'd like to use all historic patient data up to and including 24 hours past the admission. We could also place an arbitrary criterion requiring more than 5 records in this window to ensure that the extracted cohort contains sufficient input data. **(C)** Trigger events for the task, which are hospital admissions as we'd like to make a mortality prediction for each admission. **(D)** A window of the task specifying a gap in the patient timeline. Suppose we'd like to set a minimum length of admission for our cohort (*e.g.*, 48 hours). A temporal constraint (minimum window duration) of 48 hours could then be set to represent this requirement. **(E)** A window of the task specifying the task target, which is set from the end of *(D)* to the immediately subsequent $discharge$ or $death$ predicate. This creates our binary label classes for the task (*i.e.*, $discharge = 0$; $death = 1$). All windows are interrelated on the patient timeline, as shown by how each window references another in the configuration file.

## 2.5 PYTHON API

In addition to the command-line tool, we also provide a Python interface to allow researchers to easily leverage ACES for cohort extraction in their deep learning code pipelines. A full tutorial is available on the ACES online documentation.

## 2.6 EXTRACTION OUTPUT

Finally, with a dataset configured for predicates and a task configuration file, ACES will execute the extraction for the cohort and return a table where each row is a valid instance as per the criteria defined in the configuration file. Hence, each instance can be included in our cohort used for the downstream ML task. At the most basic level, the table contains the patient identifiers of our cohort, a user-defined timestamp that indexes prediction time, and a task label derived from a user-specified predicate. In addition, for each of the interrelated windows, a **start** and **end** timestamp is provided to segment the patient record, along with a summary of the number of predicates evaluated in that window.

## 3 USING ACES: A REPOSITORY OF EXAMPLE TASK CONFIGURATION FILES

To demonstrate the flexibility and utility of ACES, we define and publicly release the task configuration files described in Table 1, both with dataset-agnostic criteria and with dataset-specific predicate realizations based on the MEDS version of the public MIMIC-IV dataset. These various tasks have been previously studied, and ACES will facilitate their conceptual reproducibility to encourage benchmarking efforts and ensure robustness in ML for healthcare.

Table 1: A collection of sample configuration files for various common predictive tasks on MIMIC-IV. These tasks can be easily generalized to other datasets, such as e-ICU or other private intensive care unit (ICU) and inpatient datasets by simply swapping out appropriate predicate definitions.

| Task Name | Description |
| --- | --- |
| First 24h in-hospital mortality | Predict **mortality** within a *hospital admission* using the first 24 hours of data from that admission. |
| First 48h in-hospital mortality | Predict **mortality** within a *hospital admission* using the first 48 hours of data from that admission. |
| First 24h in-ICU mortality | Predict **mortality** within an *ICU admission* using the first 24 hours of data from that admission. |
| First 48h in-ICU mortality | Predict **mortality** within a *ICU admission* using the first 48 hours of data from that admission. |
| 30d post-hospital-discharge mortality | Predict **mortality** within 30 days of *discharge*. |
| 30d re-admission | Predict **hospital readmission** within 30 days of *discharge*. |
| Myocardial infarction 1-5Y phenotyping | Predict **myocardial infarction (MI) incidence** 1-5 years after *hospital admission*. |
| Reduced echo-derived LVEF 9m post-ECG | Predict **reduced echo-derived left ventricular ejection fraction (LVEF)** within 9 months of any *ECG*. |
| CKD onset in diabetics 5Y from kidney panel | Predict **chronic kidney disease (CKD) onset** in diabetic patients within 5 years of any *kidney panel laboratory test*. |

## 3.1 COMPUTATIONAL PROFILE

To establish an overview of the computational profile of ACES, the collection of tasks from Table 1 was extracted on MIMIC-IV. The MIMIC-IV MEDS schema has approximately 50,000 patients *per shard* with an average of approximately 80,500,000 total event rows *per shard* over seven shards. However, only a single shard was used to provide the bounded computational overview of ACES in Table 2, as the results are applicable even when scaled to larger datasets using Hydra[2]. For instance, if one shard costs $M$ memory and $T$ time, then $N$ shards may be executed in parallel with about $N * M$ memory and $T$ time, or in series with about $M$ memory and $T * N$ time.

Table 2: Performance statistics for various common predictive tasks on a single MEDS shard of MIMIC-IV. This particular shard was 5,494.39 MiBs on disk as a parquet file and 9,661.41 MiBs when loaded in memory as a dataframe. It consisted of 80,301,208 rows corresponding to 47,954 unique patients. All experiments were executed on a Linux server with 36 cores and 340 GBs of RAM available.

| Task | # Patients | # Samples | Total Time (s) | Max Memory (MiBs) |
|---|---|---|---|---|
| First 24h in-hospital mortality | 20,971 | 58,823 | 363.09 | 106,367.14 |
| First 48h in-hospital mortality | 18,847 | 60,471 | 364.62 | 108,913.95 |
| First 24h in-ICU mortality | 4,768 | 7,156 | 216.81 | 39,594.37 |
| First 48h in-ICU mortality | 4,093 | 7,112 | 217.98 | 39,451.86 |
| 30d post-hospital-discharge mortality | 28,416 | 68,547 | 182.91 | 30,434.68 |
| 30d re-admission | 18,908 | 464,821 | 367.41 | 106,064.04 |
| Myocardial Infarction 1-5Y phenotyping | 3,329 | 8,319 | 198.04 | 33,427.70 |
| Reduced echo-derived LVEF 9m post-ECG | 14 | 17 | 210.02 | 35,385.79 |
| CKD onset in diabetics 5Y from kidney panel | 736 | 3,503 | 238.65 | 44,221.81 |

## 4 DISCUSSION

### 4.1 ADDITIONAL RELATED WORK

In addition to the existing tools discussed in Section 1, there are several other areas of related work relevant to ACES. Firstly, ACES serves as a middle ground between solutions that focus on specific CDMs, such as OHDSI's ATLAS Gold et al. (2024) and i2b2's PIC-SURE (Stedman et al., 2024). Compared to these tools, ACES balances capability with greater ease of use and improved communicative value. ACES is also not tied to a particular CDM. Built on a flexible event-stream format, ACES is a no-code solution with a descriptive input format, permitting easy and wide iteration over task definitions. It can be applied to a variety of schemas, making it a versatile tool suitable for diverse research needs. ACES could also be directly connected with existing health CDMs through ETLs, such as OMOP (Reich et al., 2024), i2b2 (Murphy et al., 2010), FHIR (Bender & Sartipi, 2013), and PCORnet (Fleurence et al., 2014). These models provide already-accepted standardized frameworks for organizing and analyzing healthcare data, and supporting them could greatly enhance the utility and interoperability of ACES. Similarly, frameworks such as DescEmb (Hur et al., 2022) and GenHPF (Hur et al., 2024) hold great synergistic potential with ACES, and we believe that they can be complementary in enabling new kinds of cross-dataset training, transfer learning, and evaluation. Static benchmarks that provide standardized datasets, metrics, and baseline methods for a range of clinical problems, such as YAIB (van de Water et al., 2024), multitask learning clinical prediction benchmarks (Harutyunyan et al., 2019), and EHR-PT (McDermott et al., 2021a), can also be directly integrated with ACES to facilitate robust ML in healthcare. Lastly, ACES can be used in conjunction with various health data management tools, such as TemporAI (Saveliev & van der Schaar, 2023), PyHealth (Yang et al., 2023), OMOP-learn (Kodialam et al., 2021), and DPM360 (Suryanarayanan et al., 2021). These tools offer functionalities for pre-processing, managing, and analyzing health data for downstream tasks, and integrating ACES with them directly can streamline ML workflows.

Beyond healthcare, ACES is applicable to data from a variety of other domains, such as for finance, climate, or social media data — essentially, ACES could be used for *any* **structured**, **longitudinal** data that can be reformatted as an event-stream. This versatility makes ACES a powerful library for extracting and analyzing complex event-based datasets across different fields.

### 4.2 LIMITATIONS & FUTURE ROADMAP

ACES has limitations that can be addressed in future work. Firstly, while already very expressive, the ACES task configuration language can still be further expanded. Expressing more complex kinds of predicates, window aggregations, labeling functions, and criteria would expand the scope of ACES significantly. ACES also seeks to provide direct support for cohort extraction based on unstructured data (notes and memos) in the future. Currently, such predicates need to be manually extracted by the user, but with the help of community contributions, we hope to be able to incorporate automatic

feature extraction from clinical notes, or even images, and integrate them into configuration files for cohort extraction.

ACES is also very well poised to capture more complex patterns of task and cohort relationships, including prescribed systems of case-control matching, automated bias analyses, or propensity re-weighting over excluded populations. It is also possible to enable users to nest ACES configuration files to leverage extracted task labels as new predicates in more complex tasks and querying processes.

To enhance the scalability of ACES, we will seek to maintain support for the expanding MEDS standard. Direct interoperability with other existing resources in this space, in particular ATLAS and its OHDSI vocabulary-derived cohort definitions, is a high priority area for future work.

Finally, with the standardization that ACES offers, new opportunities for human interaction with data are also made available, such as via a natural language interface to define ACES predicates or configuration files and, thus, to extract downstream tasks, patient cohorts, or derived datasets in a **code-free manner** on diverse input EHR formats. We aim to explore the viability of leveraging large language models (LLMs) to directly format predicate and criteria definitions given a data dictionary, and to automatically construct configuration files from natural language.

### 4.3    ACES as a Catalyst for a New Era of Benchmarking

In addition to the clear impact of ACES on reproducibility, robustness, and accessibility of ML for healthcare, we also feel that ACES is critical for a "new kind of benchmark" in the field — and, in so being, is a portent of what **needs to come** should ML for healthcare progress to a more productive, communal, and impactful stage (McDermott, 2025).

In particular, we argue that for this field to progress in the manner desired by the community and to maximize positive impact for all patients, we need to develop methodologies to test, share, and develop ML solutions across diverse datasets in a meaningful and reproducible manner, **even without said datasets being publicly available** to general researchers. This capability is critical because, without it, we will never be able to offer new inductive insights about which methods are most likely to work best on novel, private data. In other words, if we cannot test our model training recipes across the diverse sets of clinical care settings, populations, and conceptual dataset schemas that exist in the real world, we similarly cannot expect those training recipes to generalize well to a myriad of downstream deployment areas.

Libraries like ACES, which make it as easy as possible for users to share the *conceptual* definitions of their tasks and prediction areas across datasets — in such a way that their colleagues can use them **even over independent, private datasets** — can help transform the kinds of benchmarking studies that we can perform in ML for healthcare. For instance, the MEDS Decentralized Extensible Validation (MEDS-DEV) (McDermott et al., 2024) effort is one such step towards enabling the generalizable assessment of ML training recipes across datasets, clinical areas, and beyond.

## 5    Conclusion

In this work, we present the Automatic Cohort Extraction System for Event-Stream Datasets (ACES). ACES is a system designed to intuitively define cohorts and downstream tasks of interest for representation learning and reliably extract those cohorts from arbitrary datasets in event-stream formats. This system enables significantly greater shareability of task definitions, reproducibility of ML training and evaluation recipes, and is as easy to use as installing a package via `pip` and running a simple command-line tool. We feel that ACES will be integral in the development of new kinds of benchmarks in ML for healthcare, which can be explored across both public and private datasets alike, as well as help characterize populations and tasks of interest in a manner that cleanly separates dataset-specific components from shareable dataset-agnostic components. To learn more about ACES and use it today in your work, please visit our GitHub repository at: `https://github.com/justin13601/aces`, and the ACES online documentation at: `https://eventstreamaces.readthedocs.io/en/latest`.

## ACKNOWLEDGMENTS AND DISCLOSURE OF FUNDING

MBAM gratefully acknowledges support from a Berkowitz Postdoctoral Fellowship at Harvard Medical School. JG is funded by the National Institutes of Health through NIH-USA R01CA294033. JX greatly appreciates support from supervisors David Eyre (University of Oxford) and Curtis Langlotz (Stanford University). We also acknowledge valuable contributions by Tom Pollard (Massachusetts Institute of Technology) and by the broader MEDS ecosystem of contributors and users.

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

# A   ACES ONLINE DOCUMENTATION INDEX

The full ACES online documentation is available at: **`https://eventstreamaces.`**
**`readthedocs.io/en/latest`**. We have also included a compiled PDF version of this documentation in the Supplementary Material.

To answer specific questions about ACES, please see the below index for the PDF documentation (links to the associated online documentation are also provided).

**How do you use ACES?**

1. *What is a task and how do you specify one?*

   Sample task descriptions and specifications are provided in the Task Examples section in Chapter 3 (`https://eventstreamaces.readthedocs.io/en/latest/notebooks/examples.html`).

   1.1. *What are predicates and how do you specify them?*

      For an overview of predicates and how they form the foundation of ACES, please refer to the Predicates DataFrame section in Chapter 4 (`https://eventstreamaces.readthedocs.io/en/latest/notebooks/predicates.html`).

   1.2. *What are windows and how do you specify them?*

      A window in ACES represents a segment in the patient record. For details on how to define a window, please refer to Chapter 1.3.3 (`https://eventstreamaces.readthedocs.io/en/latest/readme.html#windows`).

2. *How do you extract a task from a dataset?*

   For general ACES usage instructions, please refer to Chapter 2.1 (`https://eventstreamaces.readthedocs.io/en/latest/usage.html#quick-start`). Additionally, brief end-to-end instructions are also available in Chapters 1.2 and 1.3.

   2.1. *Detailed Usage Instructions for ACES CLI*

      For detailed instructions on using ACES CLI, please refer to the Usage Guide in Chapter 2.2 (`https://eventstreamaces.readthedocs.io/en/latest/usage.html#detailed-instructions`).

   2.2. *Tutorial for the ACES Python API*

      For a step-by-step tutorial on using the ACES Python API, please refer to the Code Example Notebook in Chapter 5 (`https://eventstreamaces.readthedocs.io/en/latest/notebooks/tutorial_meds.html`).

3. *ACES with / vs. Other Tools*

   For an overview of how ACES could be used with other existing complementary tools for reproducible ML, please refer to Chapter 1.4.2 (`https://eventstreamaces.readthedocs.io/en/latest/readme.html#complementary-tools`).

   For an overview of how ACES compares to other existing alternative tools for semi- or fully-automated cohort extraction, please refer to Chapter 1.4.3 (`https://eventstreamaces.readthedocs.io/en/latest/readme.html#alternative-tools`).

**How does ACES work?**

1. *What is the formal configuration language specification for ACES?*

For technical details on the ACES configuration language, please refer to the Configuration Language Specification section in Chapter 6.1 (`https://eventstreamaces.readthedocs.io/en/latest/technical.html#configuration-language-specification`).

2. *Glossary of ACES Terminology*

   For a glossary of terminology used throughout ACES, please refer to the Algorithm Terminology section in Chapter 6.3 (`https://eventstreamaces.readthedocs.io/en/latest/technical.html#algorithm-terminology`).

3. *What is the ACES extraction algorithm?*

   For technical details on the ACES algorithm, please refer to the Algorithm Design section in Chapter 6.4 (`https://eventstreamaces.readthedocs.io/en/latest/technical.html#algorithm-design`).

4. *Full ACES Module API Documentation*

   For the complete ACES module documentation, including `doctests` that ensure algorithm correctness, please refer to the Module API sections in Chapter 8 (`https://eventstreamaces.readthedocs.io/en/latest/api/modules.html`).

**How well does ACES work?**

1. *Computational Profile*

   For an overview of the computational profile of ACES, please refer to the Computational Profile section in Chapter 7 (`https://eventstreamaces.readthedocs.io/en/latest/profiling.html`).

2. *Further Examples*

   For additional examples of configuration files and criteria of different ML for healthcare tasks, please refer to the MEDS-DEV benchmarking effort on GitHub: `https://github.com/mmcdermott/MEDS-DEV`.

# B    COMPARATIVE EXPERIMENTS

We conducted preliminary experiments to quantitatively compare ACES with other alternative tools. Using the OMOP version of the MIMIC-IV Demo, as well as a synthetic dataset of 1,000 patients generated using Synthea (Walonoski et al., 2017) and converted into OMOP, we queried four tasks using ACES and two of the most comparable tools, OMOP-learn and DPM360.

We collected metrics including script runtime, peak memory usage (in MiBs), lines of code required (including configuration files and any template code needed to execute extraction), and human time spent. All experiments were conducted on a default A100 GPU instance with 84 GB of RAM and 12 CPU cores from Google Cloud Platform's Compute Engine.

Table B.1: Quantitative comparison of ACES and other comparable cohort extraction tools across datasets and tasks.

| Dataset | Method | Task | Runtime (s) | Peak Memory (MiB) | Lines of Code | Human Time (s) |
|---|---|---|---|---|---|---|
| Synthea-1000 | ACES via MEDS | First 24h in-hospital mortality | 0.386 | 389 | 35 | 120 |
| | | 30d post-hospital-discharge mortality | 0.236 | 351 | 32 | 90 |
| | | 30d re-admission | 0.337 | 355 | 22 | 60 |
| | | End-of-life prediction | 0.449 | 421 | 28 | 120 |
| | DPM360 via OMOP | First 24h in-hospital mortality | 5.932 | 390 | 205 | 2,126 |
| | | 30d post-hospital-discharge mortality | 4.188 | 550 | 257 | 1,200 |
| | | 30d re-admission | 6.260 | 870 | 288 | 2,020 |
| | | End-of-life prediction | 4.901 | 387 | 222 | 1,500 |
| MIMIC-IV Demo | ACES via MEDS | First 24h in-hospital mortality | 0.617 | 545 | 35 | 180 |
| | | 30d post-hospital-discharge mortality | 0.301 | 509 | 32 | 90 |
| | | 30d re-admission | 0.455 | 532 | 22 | 90 |
| | | End-of-life prediction | 0.349 | 589 | 28 | 300 |
| | OMOP-learn via OMOP | First 24h in-hospital mortality | 12.220 | 688 | 172 | 3,623 |
| | | 30d post-hospital-discharge mortality | 8.608 | 587 | 199 | 2,441 |
| | | 30d re-admission | 19.710 | 640 | 168 | 2,998 |
| | | End-of-life prediction | 24.540 | 932 | 251 | 12,000 |

We also qualitatively compared the adaptability of these approaches to any other given ML task. While ACES requires simple modifications to the task configuration file to capture new task logic or cohort criteria, new ATLAS executions, bespoke SQL queries, and changes to Python parameters may be needed for DPM360 and OMOP-learn.

While we acknowledge potential biases in these results due to our familiarity with ACES and only surface exposure to other extraction tools, we have aimed for a fair and objective evaluation. Based on our experience and preliminary user feedback, we find ACES to be intuitive and believe it offers a significantly improved workflow with more comprehensive functionality.

