# ACES Documentation

**Justin Xu & Matthew McDermott**

**Feb 28, 2025**

# CONTENTS

ACES is a library designed for the automatic extraction of cohorts from event-stream datasets for downstream machine learning tasks. Check out below for an overview of ACES and how it could be useful in your workflows!

# AUTOMATIC COHORT EXTRACTION SYSTEM FOR EVENT-STREAMS

**Updates**

- **[2025-01-22]** ACES accepted to ICLR'25!

- **[2024-12-10]** Latest `polars` version (`1.17.1`) is now supported.

- **[2024-10-28]** Nested derived predicates and derived predicates between static variables and plain predicates can now be defined.

- **[2024-09-01]** Predicates can now be defined in a configuration file separate to task criteria files.

- **[2024-08-29]** Latest `MEDS` version (`0.3.3`) is now supported.

- **[2024-08-10]** Expanded predicates configuration language to support regular expressions, multi-column constraints, and multi-value constraints.

- **[2024-07-30]** Added ability to place constraints on static variables, such as patient demographics.

- **[2024-06-28]** Paper available at arXiv:2406.19653.

Automatic Cohort Extraction System (ACES) is a library that streamlines the extraction of task-specific cohorts from time series datasets formatted as event-streams, such as Electronic Health Records (EHR). ACES is designed to query these EHR datasets for valid subjects, guided by various constraints and requirements defined in a YAML task configuration file. This offers a powerful and user-friendly solution to researchers and developers. The use of a human-readable YAML configuration file also eliminates the need for users to be proficient in complex dataframe querying, making the extraction process accessible to a broader audience.

There are diverse applications in healthcare and beyond. For instance, researchers can effortlessly define subsets of EHR datasets for training of foundation models. Retrospective analyses can also become more accessible to clinicians as it enables the extraction of tailored cohorts for studying specific medical conditions or population demographics. Finally, ACES can help realize a new era of benchmarking over tasks instead of data - please check out MEDS-DEV!

Currently, two data standards are directly supported: the Medical Event Data Standard (MEDS) standard and the EventStreamGPT (ESGPT) standard. You must format your data in one of these two formats by following instructions in their respective repositories. ACES also supports *any* arbitrary dataset schema, provided you extract the necessary dataset-specific plain predicates and format it as an event-stream. More information about this is available below and here.

This README provides a brief overview of this tool, instructions for use, and a description of the fields in the task configuration file (see representative configs in `sample_configs/`). Please refer to the ACES Documentation for more detailed information.

## 1.1 Installation

### 1.1.1 For MEDS v0.3.3

```
pip install es-aces
```

### 1.1.2 For ESGPT

1. Install EventStreamGPT (ESGPT):

Clone EventStreamGPT:

```
git clone https://github.com/mmcdermott/EventStreamGPT.git
```

Install with dependencies from the root directory of the cloned repo:

```
pip install -e .
```

**Note**: To avoid potential dependency conflicts, please install ESGPT first before installing ACES. This ensures compatibility with the `polars` version required by ACES.

## 1.2 Instructions for Use

1. **Prepare a Task Configuration File**: Define your predicates and task windows according to your research needs. Please see below or here for details regarding the configuration language.

2. **Prepare Dataset & Predicates DataFrame**: Process your dataset according to instructions for the MEDS or ESGPT standard so you can leverage ACES to automatically create the predicates dataframe. Alternatively, you can also create your own predicates dataframe directly (more information below and here).

3. **Execute Query**: A query may be executed using either the command-line interface or by importing the package in Python:

### 1.2.1 Command-Line Interface:

```
aces-cli data.path='/path/to/data/directory/or/file' data.standard='<meds|esgpt|direct>'␣
↪cohort_dir='/directory/to/task/config/' cohort_name='<task_config_name>'
```

For help using `aces-cli`:

```
aces-cli --help
```

## 1.2.2 Python Code:

```python
from aces import config, predicates, query
from omegaconf import DictConfig

# create task configuration object
cfg = config.TaskExtractorConfig.load(config_path="/path/to/task/config.yaml")

# get predicates dataframe
data_config = DictConfig(
    {
        "path": "/path/to/data/directory/or/file",
        "standard": "<meds|esgpt|direct>",
        "ts_format": "%m/%d/%Y %H:%M",
    }
)
predicates_df = predicates.get_predicates_df(cfg=cfg, data_config=data_config)

# execute query and get results
df_result = query.query(cfg=cfg, predicates_df=predicates_df)
```

4. **Results**: The output will be a dataframe of subjects who satisfy the conditions defined in your task configuration file. Timestamps for the start/end boundaries of each window specified in the task configuration, as well as predicate counts for each window, are also provided. Below are sample logs for the successful extraction of an in-hospital mortality cohort:

```
aces-cli cohort_name="inhospital_mortality" cohort_dir="sample_configs" data.standard=
↪"meds" data.path="MEDS_DATA"
2024-09-24 02:06:57.362 | INFO     | aces.__main__:main:153 - Loading config from
↪'sample_configs/inhospital_mortality.yaml'
2024-09-24 02:06:57.369 | INFO     | aces.config:load:1258 - Parsing windows...
2024-09-24 02:06:57.369 | INFO     | aces.config:load:1267 - Parsing trigger event...
2024-09-24 02:06:57.369 | INFO     | aces.config:load:1282 - Parsing predicates...
2024-09-24 02:06:57.380 | INFO     | aces.__main__:main:156 - Attempting to get␣
↪predicates dataframe given:
standard: meds
ts_format: '%m/%d/%Y %H:%M'
path: MEDS_DATA/
_prefix: ''

2024-09-24 02:07:58.176 | INFO     | aces.predicates:generate_plain_predicates_from_meds:
↪268 - Loading MEDS data...
2024-09-24 02:07:01.405 | INFO     | aces.predicates:generate_plain_predicates_from_
↪esgpt:272 - Generating plain predicate columns...
2024-09-24 02:07:01.579 | INFO     | aces.predicates:generate_plain_predicates_from_
↪esgpt:276 - Added predicate column 'admission'.
2024-09-24 02:07:01.770 | INFO     | aces.predicates:generate_plain_predicates_from_
↪esgpt:276 - Added predicate column 'discharge'.
2024-09-24 02:07:01.925 | INFO     | aces.predicates:generate_plain_predicates_from_
↪esgpt:276 - Added predicate column 'death'.
2024-09-24 02:07:07.155 | INFO     | aces.predicates:generate_plain_predicates_from_
↪esgpt:279 - Cleaning up predicates dataframe...
2024-09-24 02:07:07.156 | INFO     | aces.predicates:get_predicates_df:642 - Loaded␣
```

---

```
→plain predicates. Generating derived predicate columns...
2024-09-24 02:07:07.167 | INFO     | aces.predicates:get_predicates_df:645 - Added␣
→predicate column 'discharge_or_death'.
2024-09-24 02:07:07.772 | INFO     | aces.predicates:get_predicates_df:654 - Generating␣
→special predicate columns...
2024-09-24 02:07:07.841 | INFO     | aces.predicates:get_predicates_df:681 - Added␣
→predicate column '_ANY_EVENT'.
2024-09-24 02:07:07.841 | INFO     | aces.query:query:76 - Checking if '(subject_id,␣
→timestamp)' columns are unique...
2024-09-24 02:07:08.221 | INFO     | aces.utils:log_tree:57 -

trigger
|- input.end
|    +- input.start
+- gap.end
     +- target.end

2024-09-24 02:07:08.221 | INFO     | aces.query:query:85 - Beginning query...
2024-09-24 02:07:08.221 | INFO     | aces.query:query:89 - Static variable criteria␣
→specified, filtering patient demographics...
2024-09-24 02:07:08.221 | INFO     | aces.query:query:99 - Identifying possible trigger␣
→nodes based on the specified trigger event...
2024-09-24 02:07:08.233 | INFO     | aces.constraints:check_constraints:110 - Excluding␣
→14,623,763 rows as they failed to satisfy '1 <= admission <= None'.
2024-09-24 02:07:08.249 | INFO     | aces.extract_subtree:extract_subtree:252 -␣
→Summarizing subtree rooted at 'input.end'...
2024-09-24 02:07:13.259 | INFO     | aces.extract_subtree:extract_subtree:252 -␣
→Summarizing subtree rooted at 'input.start'...
2024-09-24 02:07:26.011 | INFO     | aces.constraints:check_constraints:176 - Excluding␣
→12,212 rows as they failed to satisfy '5 <= _ANY_EVENT <= None'.
2024-09-24 02:07:26.052 | INFO     | aces.extract_subtree:extract_subtree:252 -␣
→Summarizing subtree rooted at 'gap.end'...
2024-09-24 02:07:30.223 | INFO     | aces.constraints:check_constraints:176 - Excluding␣
→631 rows as they failed to satisfy 'None <= admission <= 0'.
2024-09-24 02:07:30.224 | INFO     | aces.constraints:check_constraints:176 - Excluding␣
→18,165 rows as they failed to satisfy 'None <= discharge <= 0'.
2024-09-24 02:07:30.224 | INFO     | aces.constraints:check_constraints:176 - Excluding␣
→221 rows as they failed to satisfy 'None <= death <= 0'.
2024-09-24 02:07:30.226 | INFO     | aces.extract_subtree:extract_subtree:252 -␣
→Summarizing subtree rooted at 'target.end'...
2024-09-24 02:07:41.512 | INFO     | aces.query:query:113 - Done. 44,318 valid rows␣
→returned corresponding to 11,606 subjects.
2024-09-24 02:07:41.513 | INFO     | aces.query:query:129 - Extracting label 'death'␣
→from window 'target'...
2024-09-24 02:07:41.514 | INFO     | aces.query:query:142 - Setting index timestamp as␣
→'end' of window 'input'...
2024-09-24 02:07:41.606 | INFO     | aces.__main__:main:188 - Completed in 0:00:44.
→243514. Results saved to 'sample_configs/inhospital_mortality.parquet'.
```

## 1.3 Task Configuration File

The task configuration file allows users to define specific predicates and windows to query your dataset. Below is a sample generic configuration file in its most basic form:

```
predicates:
  predicate_1:
    code: ???
  ...

trigger: ???

windows:
  window_1:
    start: ???
    end: ???
    start_inclusive: ???
    end_inclusive: ???
    has:
      predicate_1: (???, ???)

    label: ???
    index_timestamp: ???
  ...
```

Sample task configuration files for 6 common tasks are provided in `sample_configs/`. All task configurations can be directly extracted using `'direct'` mode on `sample_data/sample_data.csv` as this predicates dataframe was designed specifically to capture concepts needed for all tasks. However, only `inhospital_mortality.yaml` and `imminent-mortality.yaml` would be able to be extracted on `sample_data/esgpt_sample` and `sample_data/meds_sample` due to a lack of required concepts in the datasets (predicates are defined as per the MEDS sample data by default; modifications will be needed for ESGPT).

### 1.3.1 Predicates

Predicates describe the event at a timestamp. Predicate columns are created to contain predicate counts for each row of your dataset. If the MEDS or ESGPT data standard is used, ACES automatically computes the predicates dataframe needed for the query from the `predicates` fields in your task configuration file. However, you may also choose to construct your own predicates dataframe should you not wish to use the MEDS or ESGPT data standard.

Example predicates dataframe `.csv`:

```
subject_id,timestamp,death,admission,discharge,covid,death_or_discharge,_ANY_EVENT
1,12/1/1989 12:03,0,1,0,0,0,1
1,12/1/1989 13:14,0,0,0,0,0,1
1,12/1/1989 15:17,0,0,0,0,0,1
1,12/1/1989 16:17,0,0,0,0,0,1
1,12/1/1989 20:17,0,0,0,0,0,1
1,12/2/1989 3:00,0,0,0,0,0,1
1,12/2/1989 9:00,0,0,0,0,0,1
1,12/2/1989 15:00,0,0,1,0,1,1
```

There are two types of predicates that can be defined in the configuration file, "plain" predicates, and "derived" predicates.

### Plain Predicates

"Plain" predicates represent explicit values (either `str` or `int`) in your dataset at a particular timestamp and has 1 required `code` field (for string categorical variables) and 4 optional fields (for integer or float continuous variables). For instance, the following defines a predicate representing normal SpO2 levels (a range of 90-100 corresponding to rows where the `lab` column is `O2 saturation pulseoxymetry (%)`):

```
normal_spo2:
  code: lab//O2 saturation pulseoxymetry (%)    # required <str>//<str>
  value_min: 90                                 # optional <float/int>
  value_max: 100                                # optional <float/int>
  value_min_inclusive: true                     # optional <bool>
  value_max_inclusive: true                     # optional <bool>
  other_cols: {}                                # optional <dict>
```

Fields for a "plain" predicate:

- `code` (required): Must be one of the following:

    - a string matching values in a column named `code` (for `MEDS` only).

    - a string with a `//` sequence separating the column name and the matching column value (for `ESGPT` only).

    - a list of strings as above in the form of `{any: \[???, ???, ...\]}` (or the corresponding expanded indented `YAML` format), which will match any of the listed codes.

    - a regex in the form of `{regex: "???"}` (or the corresponding expanded indented `YAML` format), which will match any code that matches that regular expression.

- `value_min` (optional): Must be float or integer specifying the minimum value of the predicate, if the variable is presented as numerical values.

- `value_max` (optional): Must be float or integer specifying the maximum value of the predicate, if the variable is presented as numerical values.

- `value_min_inclusive` (optional): Must be a boolean specifying whether `value_min` is inclusive or not.

- `value_max_inclusive` (optional): Must be a boolean specifying whether `value_max` is inclusive or not.

- `other_cols` (optional): Must be a 1-to-1 dictionary of column name and column value, which places additional constraints on further columns.

**Note**: For memory optimization, we strongly recommend using either the List of Values or Regular Expression formats whenever possible, especially when needing to match multiple values. Defining each code as an individual string will increase memory usage significantly, as each code generates a separate predicate column. Using a list or regex consolidates multiple matching codes under a single column, reducing the overall memory footprint.

### Derived Predicates

"Derived" predicates combine existing "plain" predicates using `and` / `or` keywords and have exactly 1 required `expr` field: For instance, the following defines a predicate representing either death or discharge (by combining "plain" predicates of `death` and `discharge`):

```
# plain predicates
discharge:
  code: event_type//DISCHARGE
death:
  code: event_type//DEATH
```

```
# derived predicates
discharge_or_death:
  expr: or(discharge, death)
```

Field for a "derived" predicate:

- `expr`: Must be a string with the 'and()' / 'or()' key sequences, with "plain" predicates as its constituents.

A special predicate `_ANY_EVENT` is always defined, which simply represents any event, as the name suggests. This predicate can be used like any other predicate manually defined (ie., setting a constraint on its occurrence or using it as a trigger - more information below!).

### Special Predicates

There are also a few special predicates that you can use. These *do not* need to be defined explicitly in the configuration file, and can be directly used:

`_ANY_EVENT`: specifies any event in the data (ie., effectively set to 1 for every single row in your predicates dataframe)

`_RECORD_START`: specifies the beginning of a patient's record (ie., effectively set to 1 in the first chronological row for every `subject_id`)

`_RECORD_END`: specifies the end of a patient's record (ie., effectively set to 1 in the last chronological row for every `subject_id`)

## 1.3.2 Trigger Event

The trigger event is a simple field with a value of a predicate name. For each trigger event, a prediction by a model can be made. For instance, in the following example, the trigger event is an admission. Therefore, in your task, a prediction by a model can be made for each valid admission (ie., samples remaining after extraction according to other task specifications are considered valid). You can also simply filter to a cohort of one event (ie., just a trigger event) should you not have any further criteria in your task.

```
predicates:
  admission:
    code: event_type//ADMISSION

trigger: admission                        # trigger event <predicate>
```

## 1.3.3 Windows

Windows can be of two types, a temporally-bounded window or an event-bounded window. Below is a sample temporally-bounded window configuration:

```
trigger: admission

input:
  start: NULL
  end: trigger + 24h
  start_inclusive: True
  end_inclusive: True
```

                                                

```
has:
    _ANY_EVENT: (5, None)
```

In this example, the window `input` begins at NULL (ie., the first event or the start of the time series record), and ends at 24 hours after the `trigger` event, which is specified to be a hospital admission. The window is inclusive on both ends (ie., both the first event and the event at 24 hours after the admission, if any, is included in this window). Finally, a constraint of 5 events of any kind is placed so any valid window would include sufficient data.

Two fields (`start` and `end`) are required to define the size of a window. Both fields must be a string referencing a predicate name, or a string referencing the `start` or `end` field of another window. In addition, it may express a temporal relationship by including a positive or negative time period expressed as a string (ie., `+ 2 days`, `- 365 days`, `+ 12h`, `- 30 minutes`, `+ 60s`). It may also express an event relationship by including a sequence with a directional arrow and a predicate name (ie., `-> predicate_1` indicating the period until the next occurrence of the predicate, or `<- predicate_1` indicating the period following the previous occurrence of the predicate). Finally, it may also contain NULL, indicating the first/last event for the `start/end` field, respectively.

`start_inclusive` and `end_inclusive` are required booleans specifying whether the events, if present, at the `start` and `end` points of the window are included in the window.

The `has` field specifies constraints relating to predicates within the window. For each predicate defined previously, a constraint for occurrences can be set using a string in the format of (`<min>`, `<max>`). Unbounded conditions can be specified by using `None` or leaving it empty (ie., `(5, None)`, `(8,)`, `(None, 32)`, `(,10)`).

`label` is an optional field and can only exist in ONE window in the task configuration file if defined (an error is thrown otherwise). It must be a string matching a defined predicate name, and is used to extract the label for the task.

`index_timestamp` is an optional field and can only exist in ONE window in the task configuration file if defined (an error is thrown otherwise). It must be either `start` or `end`, and is used to create an index column used to easily manipulate the results output. Usually, one would set it to be the time at which the prediction would be made (ie., set to `end` in your window containing input data). Please ensure that you are validating your interpretation of `index_timestamp` for your task. For instance, if `index_timestamp` is set to the `end` of a particular window, the timestamp would be the event at the window boundary. However, in some cases, your task may want to exclude this boundary event, so ensure you are correctly interpreting the timestamp during extraction.

## 1.4 FAQs

### 1.4.1 Static Data

In MEDS, static variables are simply stored in rows with `null` timestamps. In ESGPT, static variables are stored in a separate `subjects_df` table. In either case, it is feasible to express static variables as a predicate and apply the associated criteria normally using the `patient_demographics` heading of a configuration file. Please see here and here for examples and details.

## 1.4.2 Complementary Tools

ACES is an integral part of the MEDS ecosystem. To fully leverage its capabilities, you can utilize it alongside other complementary MEDS tools, such as:

- MEDS-ETL, which can be used to transform various data schemas, including some common data models, into the MEDS format.

- MEDS-TAB, which can be used to generate automated tabular baseline methods (ie., XGBoost over ACES-defined tasks).

- MEDS-Polars, which contains polars-based ETL scripts.

## 1.4.3 Alternative Tools

There are existing alternatives for cohort extraction that focus on specific common data models, such as i2b2 PIC-SURE and OHDSI ATLAS.

ACES serves as a middle ground between PIC-SURE and ATLAS. While it may offer less capability than PIC-SURE, it compensates with greater ease of use and improved communication value. Compared to ATLAS, ACES provides greater capability, though with slightly lower ease of use, yet it still maintains a higher communication value.

Finally, ACES is not tied to a particular common data model. Built on a flexible event-stream format, ACES is a no-code solution with a descriptive input format, permitting easy and wide iteration over task definitions. It can be applied to a variety of schemas, making it a versatile tool suitable for diverse research needs.

# 1.5 Future Roadmap

## 1.5.1 Usability

- Extract indexing information for easier setup of downstream tasks (#37)

## 1.5.2 Coverage

- Directly support nested configuration files (#43)

- Support timestamp binning for use in predicates or as qualifiers (#44)

- Support additional label types (#45)

- Allow chaining of multiple task configurations (#49)

- Additional predicates expansions (#66)

## 1.5.3 Generalizability

- Promote generalizability across other common data models (#50)

### 1.5.4 Causal Usage

- Directly support case-control matching (#51)

### 1.5.5 Additional Tasks

- Support for additional task types and outputs (#53)
- Directly support tasks with multiple endpoints (#54)

### 1.5.6 Natural Language Interface

- LLM integration for extraction (#55)

## 1.6 Video Demonstration

## 1.7 Acknowledgements

**Matthew McDermott**, PhD | *Harvard Medical School*

**Alistair Johnson**, DPhil | *Independent*

**Jack Gallifant**, MD | *Massachusetts Institute of Technology*

**Tom Pollard**, PhD | *Massachusetts Institute of Technology*

**Curtis Langlotz**, MD, PhD | *Stanford University*

**David Eyre**, BM BCh, DPhil | *University of Oxford*

For any questions, enhancements, or issues, please file a GitHub issue. For inquiries regarding MEDS or ESGPT, please refer to their respective repositories. Contributions are welcome via pull requests.

# USAGE GUIDE

## 2.1 Quick Start

### 2.1.1 Installation

To use ACES, first determine which data standard you'd like to use. Currently, ACES can be automatically applied to the Medical Event Data Standard (MEDS) and EventStreamGPT (ESGPT). Please first follow instructions on their respective repositories to install and/or transform your data into one of these standards. Alternatively, ACES also supports *any* arbitrary dataset schema, provided you extract the necessary dataset-specific plain predicates and format it **directly** as an event-stream - details are provided here.

**Note:** If you choose to use the ESGPT standard, please install ESGPT first before installing ACES. This ensures compatibility with the `polars` version required by ACES.

**To install ACES:**

```
pip install es-aces
```

### 2.1.2 Task Configuration Example

**Example: `inhospital_mortality.yaml`**

Please see the Task Configuration File Overview for details on how to create this configuration for your own task! More examples are available here and in the GitHub repository.

This particular task configuration defines a cohort for the binary prediction of in-hospital mortality 48 hours after admission. Patients with 5 or more records between the start of their record and 24 hours after the admission will be included. The cohort includes both those that have been discharged (label=**0**) and those that have died (label=**1**).

```yaml
predicates:
  admission:
    code: code//ADMISSION
  discharge:
    code: code//DISCHARGE
  death:
    code: code//DEATH
  discharge_or_death:
    expr: or(discharge, death)

trigger: admission
```

```
windows:
  input:
    start:
    end: trigger + 24h
    start_inclusive: true
    end_inclusive: true
    has:
      _ANY_EVENT: (5, None)
    index_timestamp: end
  gap:
    start: trigger
    end: start + 24h
    start_inclusive: false
    end_inclusive: true
    has:
      admission: (None, 0)
      discharge: (None, 0)
      death: (None, 0)
  target:
    start: gap.end
    end: start -> discharge_or_death
    start_inclusive: false
    end_inclusive: true
    label: death
```

**Note**: Each configuration file contains `predicates`, a `trigger`, and `windows`. Additionally, the `label` field is used to extract the predicate count from the window it was defined in, which acts as the task label. This has been set to the `death` predicate from the `target` window in this example. The `index_timestamp` is used to specify the timestamp at which a prediction is made and can be set to `start` or `end` of a particular window. In most tasks, including this one, it can be set to `end` in the window containing input data (`input` in this example).

### 2.1.3 Run the CLI

You can now run `aces-cli` in your terminal!

#### MEDS

With MEDS, ACES supports the simultaneous extraction of tasks over multiple shards with just a single command. Suppose we have a directory structure like the following:

```
ACES/
├── sample_data/
│    ├── meds_sample/
│    │    ├── held_out/
│    │    │    └── 0.parquet
│    │    ├── train/
│    │    │    ├── 0.parquet
│    │    │    └── 1.parquet
│    │    └── tuning/
│    │         └── 0.parquet
├── sample_configs/
```

```
|    └─ inhospital_mortality.yaml
└─ ...
```

You can run the following to execute Hydra jobs in series or parallel to extract over all MEDS shards:

```
aces-cli cohort_name="inhospital_mortality" cohort_dir="sample_configs/" data.
↪standard=meds data=sharded data.root="sample_data/meds_sample/" "data.shard=$(expand_
↪shards train/1 test/0)" -m
```

If you'd like to just extract a cohort from a singular shard, you can also use the following:

```
aces-cli cohort_name="inhospital_mortality" cohort_dir="sample_configs/" data.
↪standard=meds data.path="sample_data/meds_sample/train/0.parquet"
```

### ESGPT

Given the following directory structure containing an appropriate formatted ESGPT dataset with `events_df` and `dynamic_measurements_df`:

```
ACES/
├─ sample_data/
|    ├─ esgpt_sample/
|    |    ├─ ...
|    |    ├─ events_df.parquet
|    |    └─ dynamic_measurements_df.parquet
├─ sample_configs/
|    └─ inhospital_mortality.yaml
└─ ...
```

You can extract a cohort using the following:

```
aces-cli cohort_name="inhospital_mortality" cohort_dir="sample_configs/" data.
↪standard=esgpt data.path="sample_data/esgpt_sample/"
```

### Direct Predicates

To extract from a direct predicates dataframe (`.csv` | `.parquet`) from the following directory structure:

```
ACES/
├─ sample_data/
|    └─ sample_data.csv
├─ sample_configs/
|    └─ inhospital_mortality.yaml
└─ ...
```

You can use the following:

```
aces-cli cohort_name="inhospital_mortality" cohort_dir="sample_configs/" data.
↪standard=direct data.path="sample_data/sample_data.csv"
```

**For help using `aces-cli`:**

```
aces-cli --help
```

## 2.1.4 Results

By default, results from the above examples would be saved to `sample_configs/inhospital_mortality/` containing [`train/0.parquet`, `train/1.parquet`, `test/0.parquet`] for MEDS with multiple shards, and `sample_configs/inhospital_mortality.parquet` otherwise. However, these can be overridden using `output_filepath='/path/to/output.parquet'`.

```
shape: (2, 8)
+----+----+---+----+----+----+----+----+
| subject_id | index_time | label | trigger     | input.end_ | input.star | gap.end_su |␣
→target.end |
| ---        | stamp      | ---   | ---         | summary    | t_summary  | mmary      | _
→summary    |
| i64        | ---        | i64   | datetime[μ  | ---        | ---        | ---        | -
→--         |
|            | datetime[μ |       | s]          | struct[8]  | struct[8]  | struct[8]  |␣
→struct[8]  |
|            | s]         |       |             |            |            |            | ␣
→           |
|----|----|---|----|----|----|----|----|
| 1          | 1991-01-28 | 0     | 1991-01-27  | {"input.en | {"input.st | {"gap.end" | {
→"target.e |
|            | 23:32:00   |       | 23:32:00    | d",1991-01 | art",1989- | ,1991-01-2 |␣
→nd",1991-0 |
|            |            |       |             | -27        | 12-01      | 7          | ␣
→1-29       |
|            |            |       |             | 23:32:...  | 12:0...     | 23:32:    |␣
→00... | 23:32...   |
| 2          | 1996-06-06 | 1     | 1996-06-05  | {"input.en | {"input.st | {"gap.end" | {
→"target.e |
|            | 00:32:00   |       | 00:32:00    | d",1996-06 | art",1996- | ,1996-06-0 |␣
→nd",1996-0 |
|            |            |       |             | -05        | 03-08      | 5          | ␣
→6-07       |
|            |            |       |             | 00:32:...  | 02:2...     | 00:32:    |␣
→00... | 00:32...   |
└----+----+---+----+----+----+----+----+
```

## 2.2 Detailed Instructions

### 2.2.1 Hydra

Hydra configuration files are leveraged for cohort extraction runs. All fields can be overridden by specifying their values in the command-line.

#### Data Configuration

**To set a data standard**:

*data.standard*: String specifying the data standard, must be 'meds' OR 'esgpt' OR 'direct'

**To query from multiple MEDS shards**, you must set `data=sharded`. Additionally:

*data.root*: Root directory of MEDS dataset containing shard directories

*data.shard*: Expression specifying MEDS shards using expand_shards (`$(expand_shards <str>/<int>)`)

**To query from a single MEDS shard**, you must set `data=single_file`. Additionally:

*data.path*: Path to the `.parquet` shard file

**To query from an ESGPT dataset**:

*data.path*: Directory of the full ESGPT dataset

**To query from a direct predicates dataframe**:

*data.path* Path to the `.csv` or `.parquet` file containing the predicates dataframe

*data.ts_format*: Timestamp format for predicates. Defaults to "%m/%d/%Y %H:%M"

#### Task Configuration

*cohort_dir*: Directory of your task configuration file

*cohort_name*: Name of the task configuration file

The above two fields are used below for automatically loading task configurations, saving results, and logging:

*config_path*: Path to the task configuration file. Defaults to `${cohort_dir}/${cohort_name}.yaml`

*output_filepath*: Path to store the outputs. Defaults to `${cohort_dir}/${cohort_name}/${data.shard}.parquet` for MEDS with multiple shards, and `${cohort_**dir}/${cohort_name}.parquet` otherwise

*log_dir*: Path to store logs. Defaults to `${cohort_dir}/${cohort_name}/.logs`

Additionally, predicates may be specified in a separate predicates configuration file and loaded for overrides:

*predicates_path*: Path to the separate predicates-only file. Defaults to null

### Tab Completion

Shell completion can be enabled for the Hydra configuration fields. For Bash, please run:

```
eval "$(aces-cli -sc install=bash)"
```

**Note**: you may have to run this command for every terminal - please visit Hydra's Documentation for more details.

## 2.2.2 MEDS

### Multiple Shards

A MEDS dataset can have multiple shards, each stored as a `.parquet` file containing subsets of the full dataset. We can make use of Hydra's launchers and multi-run (`-m`) capabilities to start an extraction job for each shard (`data=sharded`), either in series or in parallel (e.g., using `joblib`, or `submitit` for Slurm). To load data with multiple shards, a data root needs to be provided, along with an expression containing a comma-delimited list of files for each shard. We provide a function `expand_shards` to do this, which accepts a sequence representing `<shards_location>/<number_of_shards>`. It also accepts a file directory, where all `.parquet` files in its directory and subdirectories will be included.

```
aces-cli cohort_name="foo" cohort_dir="bar/" data.standard=meds data=sharded data.root=
→"baz/" "data.shard=$(expand_shards qux/#)" -m
```

### Single Shard

Shards are stored as `.parquet` files in MEDS. As such, the data can be loading by providing a path pointing to the `.parquet` file directly, and specifying `data=single_file`.

```
aces-cli cohort_name="foo" cohort_dir="bar/" data.standard=meds data.path="baz.parquet"
```

## 2.2.3 ESGPT

A ESGPT dataset will be encapsulated in a directory with two key files, `events_df.parquet` and `dynamic_measurements_df.parquet`. To load data formatting using the ESGPT standard, a directory of a valid ESGPT dataset containing these two tables is needed.

```
aces-cli cohort_name="foo" cohort_dir="bar/" data.standard=esgpt data.path="baz/"
```

## 2.2.4 Direct

A direct predicates dataset could also be used instead of MEDS or ESGPT to support *any* dataset schema. You will need to handle the transformation of your dataset into a predicates dataframe (see Predicates Dataframe), and save it either to a `.csv` or `.parquet` file. It can then be loaded by passing this file into ACES.

```
aces-cli cohort_name="foo" cohort_dir="bar/" data.standard=direct data.path="baz.csv |
→baz.parquet"
```

## 2.2.5 Python

You can also use the `aces.query.query()` function to extract a cohort in Python directly. Please see the Module API Reference for specifics.

`aces.query.`**`query`**(cfg: *TaskExtractorConfig*, predicates_df: *DataFrame*) → DataFrame

> Query a task using the provided configuration file and predicates dataframe.
>
> > **Parameters**
> >
> > > **cfg:** *TaskExtractorConfig*
> > > > TaskExtractorConfig object of the configuration file.
> > >
> > > **predicates_df: DataFrame**
> > > > Polars predicates dataframe.
> >
> > **Returns**
> >
> > > **The result of the task query, containing subjects who satisfy the conditions**
> > > > defined in cfg. Timestamps for the start/end boundaries of each window specified in the task configuration, as well as predicate counts for each window, are provided.
> >
> > **Return type**
> > > polars.DataFrame
> >
> > **Raises**
> >
> > > • **TypeError** – If predicates_df is not a polars.DataFrame.
> > >
> > > • **ValueError** – If the (subject_id, timestamp) columns are not unique.
>
> Examples: These examples are limited for now; see the `tests` directory for full examples.

```
>>> import logging
>>> from io import StringIO
>>> log_stream = StringIO()
>>> logger.addHandler(logging.StreamHandler(log_stream))
>>> logger.setLevel(logging.INFO)
>>> from datetime import datetime
>>> from .config import PlainPredicateConfig, WindowConfig, EventConfig
```

```
>>> cfg = None # This is obviously invalid, but we're just testing the error case.
>>> predicates_df = {"subject_id": [1, 1], "timestamp": [1, 1]}
>>> query(cfg, predicates_df)
Traceback (most recent call last):
    ...
TypeError: Predicates dataframe type must be a polars.DataFrame. Got: <class 'dict'>
→.
>>> query(cfg, pl.DataFrame(predicates_df))
Traceback (most recent call last):
    ...
ValueError: The (subject_id, timestamp) columns must be unique.
>>> cfg = TaskExtractorConfig(
...     predicates={"A": PlainPredicateConfig("A")},
...     trigger=EventConfig("_ANY_EVENT"),
...     windows={
...         "pre": WindowConfig(None, "trigger", True, False, index_timestamp="start
→"),
```

```
...          "post": WindowConfig("pre.end", None, True, True, label="A"),
...      },
...      index_timestamp_window="pre",
...      label_window="post",
... )
>>> predicates_df = pl.DataFrame({
...      "subject_id": [1, 1, 3],
...      "timestamp": [datetime(1980, 12, 28), datetime(2010, 6, 20), datetime(2010,
↪5, 11)],
...      "A": [False, False, False],
...      "_ANY_EVENT": [True, True, True],
... })
>>> result = query(cfg, predicates_df)
>>> result.select("subject_id", "trigger")
shape: (3, 2)
+----+-------+
| subject_id | trigger              |
| ---        | ---                  |
| i64        | datetime[µs]         |
|----|-------|
| 1          | 1980-12-28 00:00:00 |
| 1          | 2010-06-20 00:00:00 |
| 3          | 2010-05-11 00:00:00 |
└----+-------+
>>> "index_timestamp" in result.columns
True
>>> "label" in result.columns
True
>>> cfg = TaskExtractorConfig(
...      predicates={"A": PlainPredicateConfig("A", static=True)},
...      trigger=EventConfig("_ANY_EVENT"),
...      windows={},
... )
>>> query(cfg, predicates_df)
shape: (0, 0)
++
||
└+
>>> log_output = log_stream.getvalue()
>>> "Static variable criteria specified, filtering patient demographics..." in log_
↪output
True
>>> "No static variable criteria specified, removing all rows with null timestamps..
↪." in log_output
True
>>> predicates_df = pl.DataFrame({
...      "subject_id": [1, 1, 3],
...      "timestamp": [None, datetime(2010, 6, 20), datetime(2010, 5, 11)],
...      "A": [True, False, False],
...      "_ANY_EVENT": [False, False, False],
... })
>>> result = query(cfg, predicates_df)
```

```
>>> log_output = log_stream.getvalue()
>>> "No valid rows found for the trigger event" in log_output
True
```

The `cfg` parameter must be of type *aces.config.TaskExtractorConfig*, and the `predicates_df` parameter must be of type `polars.DataFrame`.

Details about the configuration language used to define the `cfg` parameter can be found in /configuration.

For example, to query an in-hospital mortality task on the sample data (both the configuration file and data are provided in the repository) using the `'direct'` predicates method:

```
>>> from aces import query, predicates, config
>>> from omegaconf import DictConfig

>>> cfg = config.TaskExtractorConfig.load(config_path="sample_configs/inhospital_
→mortality.yaml")

>>> data_config = DictConfig({"path": "sample_data.csv", "standard": "direct", "ts_format
→": "%m/%d/%Y %H:%M"})
>>> predicates_df = predicates.get_predicates_df(cfg=cfg, data_config=data_config)

>>> query.query(cfg=cfg, predicates_df=predicates_df)
```

## 2.2.6 Separate Predicates-Only File

For more complex tasks involving a large number of predicates, a separate predicates-only "database" file can be created and passed into `TaskExtractorConfig.load()`. Only referenced predicates will have a predicate column computed and evaluated, so one could create a dataset-specific deposit file with many predicates and reference as needed to ensure the cleanliness of the dataset-agnostic task criteria file.

```
>>> cfg = config.TaskExtractorConfig.load(config_path="criteria.yaml", predicates_path=
→"predicates.yaml")
```

If the same predicates are defined in both the task configuration file and the predicates-only file, the predicates-only definition takes precedent and will be used to override previous definitions. As such, one may create a predicates-only "database" file for a particular dataset, and override accordingly for various tasks.

# TASK EXAMPLES

Provided below are two examples of mortality prediction tasks that ACES could easily extract subject cohorts for. The configurations have been tested all the provided synthetic data in the repository (sample_data/), as well as the MIMIC-IV dataset loaded using MEDS & ESGPT (with very minor changes to the below predicate definition). The configuration files for both of these tasks are provided in the repository (sample_configs/), and cohorts can be extracted using the `aces-cli` tool:

```
aces-cli data.path='/path/to/MIMIC/ESGPT/schema/' data.standard='esgpt' cohort_dir=
↪'sample_configs/' cohort_name='...'
```

For simplicity and consistency of these examples, we will use the following 4 window types and names. In practice, ACES supports arbitrary window types and window names, so you have full flexibility and control in how you define your windows in your task logic.

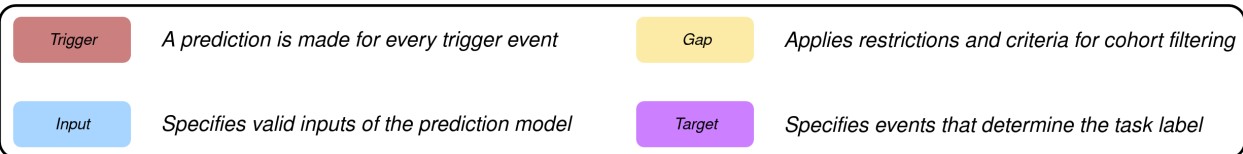

```
[1]: import json

     import yaml
     from bigtree import print_tree

     from aces import config
```

```
[2]: config_path = "../../../sample_configs"
```

## 3.1 In-hospital Mortality

The below timeline specifies a binary in-hospital mortality prediction task where we aim to predict whether the patient dies (label=1) or is discharged (label=0):

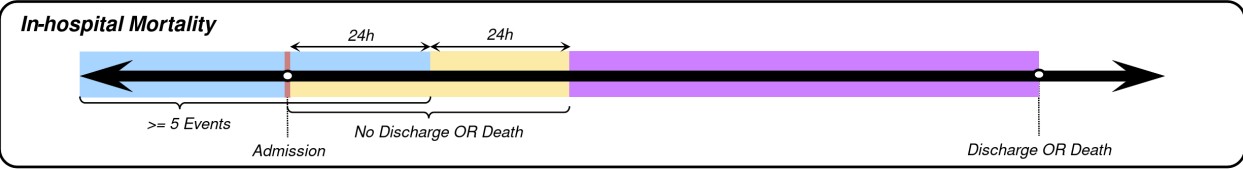

Suppose we'd like to use all patient data up to and including 24 hours past an admission. We can therefore define the `input` window as above. We can also place criteria on the windows to filter out cohort. In this case, we'd like to ensure there is sufficient prior input data for our model, so we place a constraint that there must be at least 5 or more records (ie., with unique timestamps) within `input.

Next, suppose we'd like to only include hospital admissions that were longer than 48 hours. To represent this clause, we can specify `gap` as above with a length of 48 hours (overlapping the initial 24 hours of `input`). If we then place constraints on `gap`, preventing it to have any discharge or death events, then the admission must then be at least 48 hours.

Finally, we specify `target`, which is our prediction horizon and lasts until the immediately next discharge or death event. This allows us to extract a cohort that includes both patients who have died and those who did not (ie., successfully discharged).

In addition to constructing a cohort based on dynamic variables, we can also place constraints on static variables (ie., eye color). Suppose we'd like to filter our cohort to only those with blue eyes.

We can then specify a task configuration as below:

```
predicates:
  admission:
    code: event_type//ADMISSION
  discharge:
    code: event_type//DISCHARGE
  death:
    code: event_type//DEATH
  discharge_or_death:
    expr: or(discharge, death)

patient_demographics:
  eye_color:
    code: EYE//blue

trigger: admission

windows:
  input:
    start: NULL
    end: trigger + 24h
    start_inclusive: True
    end_inclusive: True
    has:
      _ANY_EVENT_: (5, None)
    index_timestamp: end
  gap:
    start: trigger
    end: start + 48h
    start_inclusive: False
    end_inclusive: True
    has:
      admission: (None, 0)
      discharge: (None, 0)
      death: (None, 0)
  target:
    start: gap.end
```

```
    end: start -> discharge_or_death
    start_inclusive: False
    end_inclusive: True
    label: death
```

### 3.1.1 Predicates

To capture our task definition, we must define at least three predicates. Recall that these predicates are dataset-specific, and thus may be different depending on the data standard used or data schema.

For starters, we are specifically interested in mortality "in the hospital". As such, an `admission` and a `discharge` predicate would be needed to represent events where patients are officially admitted "into" the hospital and where patients are officially discharged "out of" the hospital. We also need the `death` predicate to capture death events so we can accurately capture the mortality component.

Since our task endpoints could be either `discharge` or `death` (ie., binary label prediction), we may also create a derived predicate `discharge_or_death` which is expressed by an `OR` relationship between `discharge` and `death`.

### 3.1.2 Trigger

A prediction can be made for each event specified in `trigger`. This field must contain one of the previously defined dataset-specific predicates. In our case, we'd like to make a prediction of mortality for each valid admission in our cohort, and thus we set `trigger` to be the `admission` predicate.

### 3.1.3 Windows

The windows section contains the remaining three windows we defined previously - `input`, `gap`, and `target`.

`input` begins at the start of a patient's record (ie., `NULL`), and ends 24 hours past `trigger` (ie., `admission`). As we'd like to include the events specified at both the start and end of `input`, if present, we can set both `start_inclusive` and `end_inclusive` as `True`. Our constraint on the number of records is specified in `has` using the `_ANY_EVENT` predicate, with its value set to be greater or equal to 5 (ie., unbounded parameter on the right as seen in `(5, None)`).

**Note**: Since we'd like to make a prediction at the end of `input`, we can set `index_timestamp` to be `end`, which corresponds to the timestamp of `trigger + 24h`.

`gap` also begins at `trigger`, and ends 48 hours after. As we have included included the left boundary event in `trigger` (ie., `admission`), it would be reasonable to not include it again as it should not play a role in `gap`. As such, we set `start_inclusive` to `False`. As we'd like our admission to be at least 48 hours long, we can place constraints specifying that there cannot be any `admission`, `discharge`, or `death` in `gap` (ie., right-bounded parameter at `0` as seen in `(None, 0)`).

`target` beings at the end of `gap`, and ends at the next discharge or death event (ie., `discharge_or_death` predicate). We can use this arrow notation which ACES recognizes as event references (ie., `->` and `<-`; see Time Range Fields). In our case, we end `target` at the next `discharge_or_death`. Similarly, as we included the event at the end of `gap`, if any, already in `gap`, we can set `start_inclusive` to `False`.

**Note**: Since we'd like to make a binary mortality prediction, we can extract the `death` predicate as a label from `target`, by specifying the `label` field to be `death`.

### 3.1.4 Task Tree

ACES is then able to parse our configuration file and generate the below task tree that captures our task. You can see the relationships between nodes in the tree reflect that of the task timeline:

```
[3]: inhospital_mortality_cfg_path = f"{config_path}/inhospital_mortality.yaml"
     cfg = config.TaskExtractorConfig.load(config_path=inhospital_mortality_cfg_path)
     tree = cfg.window_tree
     print_tree(tree)
```

```
trigger
├─ input.end
│   └─ input.start
└─ gap.end
    └─ target.end
```

## 3.2 Imminent Mortality

The below timeline specifies a binary imminent mortality prediction task where we aim to predict whether the patient dies (label=1) or not (label=0) in the immediate 24 hours following a 2 hour period from any given time:

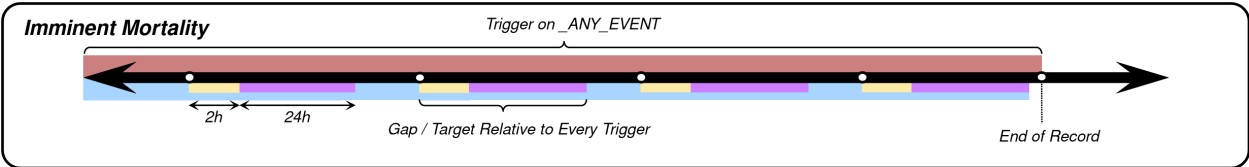

In this case, we'd like to use all patient data up to and including the triggers (ie., every event). However, as we won't be placing any constraints on this window, we actually do not need to add it into our task configuration, as ultimately, any and all data rows prior to the `trigger` timestamp will be included.

You can see that the `trigger` window essentially encapsulates the entire patient record. This is because we'd like to define a task to predict mortality at every single event in the record for simplicity. In practice, this might not be reasonable or feasible. For instance, you may only be interested in predicting imminent mortality within an admission. In this case, you might create `admission_window`, starting from `admission` predicates to `discharge_or_death` predicates. ACES would create a branch in the task tree from this window, and since the results ensure that all output rows satisfy all tree branches, the cohort would only include triggers on events in `admission_window`.

For this particular example, we create `gap` of 2 hours and `target` of 24 hours following `gap`. No specific constraints are set for either window, except for the time durations.

We can then specify a task configuration as below:

```
predicates:
  death:
    code: event_type//DEATH

trigger: _ANY_EVENT

windows:
  gap:
    start: trigger
    end: start + 2 hours
    start_inclusive: True
```

```
      end_inclusive: True
      index_timestamp: end
  target:
    start: gap.end
    end: start + 24 hours
    start_inclusive: False
    end_inclusive: True
    label: death
```

### 3.2.1 Predicates

Only a `death` predicate is required in this example to capture our `label` in `target`. However, as noted here, certain special predicates can be used without explicit definition. In this case, we will make use of `_ANY_EVENT`.

### 3.2.2 Trigger

A prediction can be made for each and every event. As such, `trigger` is set to the special predicate `_ANY_EVENT`.

### 3.2.3 Windows

The windows section contains the two windows we defined - `gap` and `target`. In this case, the `gap` and `target` windows are defined relative to every single event (ie., `_ANY_EVENT`). `gap` begins at `trigger`, and ends 2 hours after. `target` beings at the end of `gap`, and ends 24 hours after.

**Note**: Since we'd again like to make a binary mortality prediction, we can extract the `death` predicate as a label from `target`, by specifying the `label` field to be `death`. Additionally, since a prediction would be made at the end of each `gap`, we can set `index_timestamp` to be `end`, which corresponds to the timestamp of `_ANY_EVENT + 24h`.

### 3.2.4 Task Tree

As in the in-hospital mortality case, ACES is able to parse our configuration file and generate a task tree:

```
[4]: imminent_mortality_cfg_path = f"{config_path}/imminent_mortality.yaml"
     cfg = config.TaskExtractorConfig.load(config_path=imminent_mortality_cfg_path)

     tree = cfg.window_tree
     print_tree(tree)
```

```
trigger
└─ gap.end
    └─ target.end
```

## 3.3 Other Examples

A few other examples are provided in sample_configs/ of the repository. We will continue to add task configurations to MEDS-DEV, a benchmarking effort for EHR representation learning - stay tuned!

# PREDICATES

In ACES, predicates specify how particular concepts relevant to your task of interest is expressed in your dataset of interest. These dataset-specific items form a large foundation of the cohort extraction algorithm as the more complex dataset-agnostic windowing logic of your task is defined based on your predicates, ultimately facilitating ease-of-sharing for your task configurations.

## 4.1 Predicate Columns

A predicate column is simply a column in a dataframe containing numerical counts (often just `0`'s and `1`'s), representing the number of times a given predicate (concept) occurs at a given timestamp for a given patient.

Suppose you had a simple time-sorted dataframe as follows:

| subject_id | timestamp | code | value |
|---|---|---|---|
| 1 | null | SEX//male | null |
| 1 | 1989-01-01 00:00:00 | ADMISSION | null |
| 1 | 1989-01-01 01:00:00 | LAB//HR | 90 |
| 1 | 1989-01-01 01:00:00 | PROCEDURE_START | null |
| 1 | 1989-01-01 02:00:00 | DISCHARGE | null |
| 1 | 1989-01-01 02:00:00 | PROCEDURE_END | null |
| 2 | null | SEX//female | null |
| 2 | 1991-05-06 12:00:00 | ADMISSION | null |
| 2 | 1991-05-06 20:00:00 | DEATH | null |
| 3 | null | SEX//male | null |
| 3 | 1980-10-17 22:00:00 | ADMISSION | null |
| 3 | 1980-10-17 22:00:00 | LAB//HR | 120 |
| 3 | 1980-10-18 01:00:00 | LAB//temp | 37 |
| 3 | 1980-10-18 09:00:00 | DISCHARGE | null |
| 3 | 1982-02-02 02:00:00 | ADMISSION | null |
| 3 | 1982-02-02 04:00:00 | DEATH | null |

The `code` column contains a string of an event that occurred at the given `timestamp` for a given `subject_id`. **Note**: Static variables are shown as rows with `null` timestamps.

You may then create a series of predicate columns depending on what suits your needs. For instance, here are some plausible predicate columns that could be created:

| sub-ject_id | timestamp | admis-sion | dis-charge | death | dis-charge_or_dea | lab | proce-dure_start | HR_over_1 | male |
|---|---|---|---|---|---|---|---|---|---|
| 1 | 1989-01-01 00:00:00 | 1 | 0 | 0 | 0 | 0 | 0 | 0 | 1 |
| 1 | 1989-01-01 01:00:00 | 0 | 0 | 0 | 0 | 1 | 1 | 1 | 1 |
| 1 | 1989-01-01 02:00:00 | 0 | 1 | 0 | 1 | 0 | 0 | 0 | 1 |
| 2 | 1991-05-06 12:00:00 | 1 | 0 | 0 | 0 | 0 | 0 | 0 | 0 |
| 2 | 1991-05-06 20:00:00 | 0 | 0 | 1 | 1 | 0 | 0 | 0 | 0 |
| 3 | 1980-10-17 22:00:00 | 1 | 0 | 0 | 0 | 1 | 0 | 0 | 1 |
| 3 | 1980-10-18 01:00:00 | 0 | 0 | 0 | 0 | 1 | 0 | 0 | 1 |
| 3 | 1980-10-18 09:00:00 | 0 | 1 | 0 | 1 | 0 | 0 | 0 | 1 |
| 3 | 1982-02-02 02:00:00 | 1 | 0 | 0 | 0 | 0 | 0 | 0 | 1 |
| 3 | 1982-02-02 04:00:00 | 0 | 0 | 1 | 1 | 0 | 0 | 0 | 1 |

**Note**: This set of predicates are all `plain` predicates (ie., explicitly expressed as a value in the dataset), with the exception of the `derived` predicate `discharge_or_death`, which can be expressed by applying boolean logic on the `discharge` and `death` predicates (ie., `or(discharge, death)`). You may choose to create these columns for `derived` predicates explicitly (as you would `plain` predicates). Or, ACES can automatically create them from `plain` predicates if the boolean logic is provided in the task configuration file. Please see Predicates for more information.

Additionally, you may notice that the tables differ in shape. In the original raw data, (`subject_id`, `timestamp`) is not unique. However, a final predicates dataframe must have unique (`subject_id`, `timestamp`) pairs. If the MEDS or ESGPT standard is used, ACES will automatically collapse rows down into unique per-patient per-timestamp levels (ie., grouping by these two columns and aggregating by summing predicate counts). However, if creating predicate columns directly, please ensure your dataframe is unique over (`subject_id`, `timestamp`).

## 4.2 Sample Predicates DataFrame

A sample predicates dataframe is provided in the repository (sample_data/sample_data.csv). This dataframe holds completely synthetic data and was designed such that the accompanying sample configuration files in the repository (sample_configs/) could be directly extracted.

```
[1]: import polars as pl

     pl.read_csv("../../../sample_data/sample_data.csv")
```

```
[1]: shape: (54, 17)
     +----+-----+--+---+-+----+----+----+----+
     | subject_id | timestamp   | male | female | ... | procedure_ | ventilatio | diagnosis_ |
     →| diagnosis_ |
     | ---        | ---         | ---  | ---    |     | | end       | n          | ICD9CM_410 | |
     →ICD10CM_I2 |
```

(continues on next page)

```
| i64           | str        | i64 | i64 |   | ---        | ---        | 71    |␣
↪14         |
|             |            |     |     |   | i64        | i64        | ---   | -
↪--      |
|             |            |     |     |   |            |            | i64   |␣
↪i64       |
|----|-----|--|---|-|----|----|----|----|
| 1           | null       | 1   | 0   |...| 0          | 0          | 0     |␣
↪| 0         |
| 1           | 12/1/1989  | 0   | 0   |...| 0          | 0          | 0     |␣
↪| 0         |
|             | 12:03      |     |     |   |   |        |            |       | |␣
↪            |
| 1           | 12/1/1989  | 0   | 0   |...| 0          | 0          | 0     |␣
↪| 0         |
|             | 13:14      |     |     |   |   |        |            |       | |␣
↪            |
| 1           | 12/1/1989  | 0   | 0   |...| 0          | 0          | 0     |␣
↪| 0         |
|             | 15:17      |     |     |   |   |        |            |       | |␣
↪            |
| 1           | 12/1/1989  | 0   | 0   |...| 0          | 0          | 0     |␣
↪| 0         |
|             | 16:17      |     |     |   |   |        |            |       | |␣
↪            |
| ...         | ...        | ... | ... |   | ...| ...   | ...        | ...   |␣
↪...         | ...        |
| 3           | 3/9/1996   | 0   | 0   |...| 0          | 0          | 0     |␣
↪| 0         |
|             | 11:00      |     |     |   |   |        |            |       | |␣
↪            |
| 3           | 3/9/1996   | 0   | 0   |...| 0          | 0          | 0     |␣
↪| 0         |
|             | 19:00      |     |     |   |   |        |            |       | |␣
↪            |
| 3           | 3/9/1996   | 0   | 0   |...| 0          | 0          | 0     |␣
↪| 0         |
|             | 22:00      |     |     |   |   |        |            |       | |␣
↪            |
| 3           | 3/11/1996  | 0   | 0   |...| 1          | 1          | 0     |␣
↪| 0         |
|             | 21:00      |     |     |   |   |        |            |       | |␣
↪            |
| 3           | 3/12/1996  | 0   | 0   |...| 0          | 0          | 0     |␣
↪| 0         |
|             | 0:00       |     |     |   |   |        |            |       | |␣
↪            |
└----+-----+--+---+-+----+----+----+----+
```

## 4.3 Generating the Predicates DataFrame

The predicates dataframe will always have the `subject_id` and `timestamp` columns. They should be unique between these two columns, as each row can capture multiple events.

ACES is able to automatically compute the predicates dataframe from your dataset and the fields defined in your task configuration if you are using the MEDS or ESGPT data standard. Should you choose to not transform your dataset into one of these two currently supported standards, you may also navigate the transformation yourself by creating your own predicates dataframe.

Again, it is acceptable if your own predicates dataframe only contains `plain` predicate columns, as ACES can automatically create `derived` predicate columns from boolean logic in the task configuration file. However, for complex predicates that would be impossible to express (outside of `and/or`) in the configuration file, we direct you to create them manually prior to using ACES. Support for additional complex predicates is planned for the future, including the ability to use SQL or other expressions (see #66).

**Note**: When creating `plain` predicate columns directly, you must still define them in the configuration file (they could be with an arbitrary value in the `code` field) - ACES will verify their existence after data loading (ie., by validating that a column exists with the predicate name in your dataframe). You will also need them for referencing in your windows.

Example of the `derived` predicate `discharge_or_death`, expressed as an `or()` relationship between `plain` predicates `discharge` and `death`, which have been directly defined (ie., arbitrary values for their codes, `defined in data`, are present).

```
predicates:
  death:
    code: defined in data
  discharge:
    code: defined in data
  discharge_or_death:
    expr: or(discharge, death)
  ...
```

# FIVE

# CODE TUTORIAL WITH SYNTHETIC MEDS DATA

## 5.1 Set-up

### 5.1.1 Imports

First, let's import ACES! Three modules - `config`, `predicates`, and `query` - are required to execute an end-to-end cohort extraction. `omegaconf` is also required to express our data config parameters in order to load our `MEDS` dataset. Other imports are only needed for visualization!

```
[1]: import json
     from pathlib import Path

     import pandas as pd
     import yaml
     from bigtree import print_tree
     from IPython.display import display
     from omegaconf import DictConfig

     from aces import config, predicates, query
```

### 5.1.2 Directories

Next, let's specify our paths and directories. In this tutorial, we will extract a cohort for a typical in-hospital mortality prediction task from the MEDS synthetic sample dataset. The task configuration file and sample data are both shipped with the repository in sample_configs/ and sample_data/ folders in the project root, respectively.

```
[2]: config_path = "../../../sample_configs/inhospital_mortality.yaml"
     data_path = "../../../sample_data/meds_sample/"
```

## 5.2 Configuration File

The task configuration file is the core configuration language that ACES uses to extract cohorts. Details about this configuration language is available in Configuration Language. In brief, the configuration file contains `predicates`, `patient_demographics`, `trigger`, and `windows` sections.

The `predicates` section is used to define dataset-specific concepts that are needed for the task. In our case of binary mortality prediction, we are interested in extracting a cohort of patients that have been admitted into the hospital and who were subsequently discharged or died. As such `admission`, `discharge`, `death`, and `discharge_or_death` would be handy predicates.

The `patient_demographics` section is used to define static concepts that remain constant for subjects over time. For instance, sex is a common static variable. Should we want to filter out cohort to patients with a specific sex, we can do so here in the same way as defining predicates. For more information on predicates, please refer to this guide. In this example, let's say we are only interested in male patients.

We'd also like to make a prediction of mortality for each admission. Hence, a reasonable `trigger` event would be an `admission` predicate.

Suppose in our task, we'd like to set a constraint that the admission must have been more than 48 hours long. Additionally, for our prediction inputs, we'd like to use all information in the patient record up until 24 hours after admission, which must contain at least 5 event records (as we'd want to ensure there is sufficient input data). These clauses are captured in the `windows` section where each window is defined relative to another.

```
[3]: with open(config_path, "r") as stream:
         data_loaded = yaml.safe_load(stream)
         print(json.dumps(data_loaded, indent=4))
```

```
{
    "predicates": {
        "admission": {
            "code": {
                "regex": "ADMISSION//.*"
            }
        },
        "discharge": {
            "code": {
                "regex": "DISCHARGE//.*"
            }
        },
        "death": {
            "code": "DEATH"
        },
        "discharge_or_death": {
            "expr": "or(discharge, death)"
        }
    },
    "patient_demographics": {
        "male": {
            "code": "SEX//male"
        }
    },
    "trigger": "admission",
    "windows": {
        "input": {
```

(continues on next page)

```
                "start": null,
                "end": "trigger + 24h",
                "start_inclusive": true,
                "end_inclusive": true,
                "has": {
                    "_ANY_EVENT": "(5, None)"
                },
                "index_timestamp": "end"
            },
            "gap": {
                "start": "trigger",
                "end": "start + 48h",
                "start_inclusive": false,
                "end_inclusive": true,
                "has": {
                    "admission": "(None, 0)",
                    "discharge": "(None, 0)",
                    "death": "(None, 0)"
                }
            },
            "target": {
                "start": "gap.end",
                "end": "start -> discharge_or_death",
                "start_inclusive": false,
                "end_inclusive": true,
                "label": "death"
            }
        }
    }
}
```

We can see that the `input` window begins at `null` (start of the patient record) and ends 24 hours after `trigger` (`admission`). A `gap` window is defined for 24 hours after the end of the `input` window, constraining the admission to be longer than 48 hours at minimum. Finally, a `target` window is specified from the end of the `gap` window to either the next `discharge` or `death` event (ie., `discharge_or_death`). This would allow us to extract a binary label for each patient in our cohort to be used in the prediction task (ie., field `label` in the `target` window, which will extract `0`: discharged, `1`: died). Additionally, an `index_timestamp` field is set as the `end` of the `input` window to denote when a prediction is made (ie., at the end of the `input` window when all input data is fed into the model), and can be used to index extraction results.

We now load our configuration file by passing its path (`str`) into `config.TaskExtractorConfig.load()`. This parses the configuration file for each of the three key sections indicated above and prepares ACES for extraction based on our defined constraints (inclusion/exclusion criteria for each window).

```
[4]: cfg = config.TaskExtractorConfig.load(config_path=config_path)
```

### 5.2.1 Task Tree

With the configuration file loaded and parsed, we can access a visualization of a tree structure that is representative of our task of interest. As seen, the tree nodes are `start` and `end` time points of the windows that were defined in the configuration file, and the tree edges express the relationships between these windows. ACES will traverse this tree and recursively compute aggregated predicate counts for each subtree. This would allow us to filter our dataset to valid realizations of this task tree, which would make up our task cohort.

```
[5]: tree = cfg.window_tree
     print_tree(tree)
```

```
trigger
├── input.end
│      └── input.start
└── gap.end
       └── target.end
```

## 5.3 Data

This tutorial uses synthetic data of 100 patients stored in the MEDS standard. For more information about this data, please refer to the generation of this synthetic data in the ESGPT Documentation (separately converted to MEDS). Here is what the data looks like:

```
[6]: pd.read_parquet(f"{data_path}/train/0.parquet").head()
```

```
[6]:    subject_id                time               code  numeric_value
     0           0                 NaT           SEX//male            NaN
     1           0 2010-06-24 13:23:00   ADMISSION//CARDIAC            NaN
     2           0 2010-06-24 13:23:00                  HR      -0.266073
     3           0 2010-06-24 13:23:00            LAB//SpO2       0.283409
     4           0 2010-06-24 13:23:00                TEMP       0.618533
```

### 5.3.1 Predicate Columns

The next step in our cohort extraction is the generation of predicate columns. Our defined dataset-agnostic windows (ie., complex task logic) are linked to dataset-specific predicates (ie., dataset observations and concepts), which facilitates the sharing of tasks across datasets. As such, the predicates dataframe is the foundational unit on which ACES acts upon.

A predicate column is simply a column containing numerical counts (often just `0`'s and `1`'s), representing the number of times a given predicate (concept) occurs at a given timestamp for a given patient.

In the case of MEDS (and ESGPT), ACES support the automatic generation of these predicate columns from the configuration file. However, some fields need to be provided via a `DictConfig` object. These include the path to the directory of the MEDS dataset (`str`) and the data standard (which is `meds` in this case).

Given this data configuration, we then call `predicates.get_predicates_df()` to generate the relevant predicate columns for our task. Due to the nature of the specified predicates, the resulting dataframe simply contains the unique (`subject_id`, `timestamp`) pairs and binary columns for each predicate. An additional predicate `_ANY_EVENT` is also generated - this will be used to enforce our constraint of the number of events in the `input` window.

```
[7]: data_config = DictConfig({"path": data_path, "standard": "meds"})

     predicates_df = predicates.get_predicates_df(cfg=cfg, data_config=data_config)
     display(predicates_df)
```

Expand shards is not enabled but your data path is a directory. If you are working with␣
→sharded datasets or large-scale queries, using `expand_shards` and`data=sharded` will␣
→improve efficiency and completeness.

```
shape: (31_025, 8)
+----+------+----+----+---+--+------+----+
| subject_id | timestamp     | admission | discharge | death | male | discharge_or_d | _
→ANY_EVENT |
| ---        | ---           | ---       | ---       | ---   | ---  | eath           | -
→--         |
| i64        | datetime[µs]  | i64       | i64       | i64   | i64  | ---            |␣
→i64         |
|            |               |           |           |       |      | i64            | ␣
→            |
|----|------|----|----|---|--|------|----|
| 0          | null          | 0         | 0         | 0     | 1    | 0              |␣
→null        |
| 0          | 2010-06-24    | 1         | 0         | 0     | 0    | 0              |␣
→1           |
|            | 13:23:00      |           |           |       |      |                | ␣
→            |
| 0          | 2010-06-24    | 0         | 0         | 0     | 0    | 0              |␣
→1           |
|            | 14:23:00      |           |           |       |      |                | ␣
→            |
| 0          | 2010-06-24    | 0         | 0         | 0     | 0    | 0              |␣
→1           |
|            | 15:23:00      |           |           |       |      |                | ␣
→            |
| 0          | 2010-06-24    | 0         | 0         | 0     | 0    | 0              |␣
→1           |
|            | 16:23:00      |           |           |       |      |                | ␣
→            |
| ...        | ...           | ...       | ...       | ...   | ...  | ...            ␣
→            | ...           |
| 99         | 2010-11-20    | 0         | 0         | 0     | 0    | 0              |␣
→1           |
|            | 08:20:06      |           |           |       |      |                | ␣
→            |
| 99         | 2010-11-20    | 0         | 0         | 0     | 0    | 0              |␣
→1           |
|            | 09:20:06      |           |           |       |      |                | ␣
→            |
| 99         | 2010-11-20    | 0         | 0         | 0     | 0    | 0              |␣
→1           |
|            | 10:20:06      |           |           |       |      |                | ␣
→            |
| 99         | 2010-11-20    | 0         | 0         | 0     | 0    | 0              |␣
→1           |
```

(continues on next page)

```
|            | 11:20:06    |        |        |       |      |      |       |␣
↪           |
| 99         | 2010-11-20  | 0      | 1      | 0     | 0    | 1    |␣
↪1          |
|            | 12:20:06    |        |        |       |      |      |       |␣
↪           |
└----+------+----+----+---+--+------+----+
```

## 5.4 End-to-End Query

Finally, with our task configuration object and the computed predicates dataframe, we can call `query.query()` to execute the extraction of our cohort.

Each row of the resulting dataframe is a valid realization of our task tree. Hence, each instance can be included in our cohort used for the prediction of in-hospital mortality as defined in our task configuration file. The output contains:

- `subject_id`: subject IDs of our cohort (since we'd like to treat individual admissions as separate samples, there will be duplicate subject IDs)

- `index_timestamp`: timestamp of when a prediction is made, which coincides with the `end` timestamp of the `input` window (as specified in our task configuration)

- `label`: binary label of mortality, which is derived from the `death` predicate of the `target` window (as specified in our task configuration)

- `trigger`: timestamp of the `trigger` event, which is the `admission` predicate (as specified in our task configuration)

Additionally, it also includes a column for each node of our task tree in a pre-order traversal order. Each column contains a `pl.Struct` object containing the name of the node, the start and end times of the window it represents, and the counts of all defined predicates in that window.

```
[8]: df_result = query.query(cfg=cfg, predicates_df=predicates_df)
     display(df_result)
```

All labels in the extracted cohort are the same: '0'. This may indicate an issue with␣
↪the task logic. Please double-check your configuration file if this is not expected.

```
shape: (87, 8)
+----+----+---+----+----+----+----+----+
| subject_id | index_time | label | trigger    | input.end_ | input.star | gap.end_su |␣
↪target.end |
| ---        | stamp      | ---   | ---        | summary    | t_summary  | mmary      | _
↪summary    |
| i64        | ---        | i64   | datetime[μ | ---        | ---        | ---        | -
↪--         |
|            | datetime[μ |       | s]         | struct[8]  | struct[8]  | struct[8]  |␣
↪struct[8]  |
|            | s]         |       |            |            |            |            |␣
↪           |
|----|----|---|----|----|----|----|----|
| 0          | 2010-10-05 | 0     | 2010-10-04 | {"input.en | {"input.st | {"gap.end" | {
↪"target.e |
|            | 17:23:00   |       | 17:23:00   | d",2010-10 | art",2010- | ,2010-10-0 |␣
```

(continued from previous page)

```
↪nd",2010-1 |
|              |               |          |                 | -04          | 06-24       | 4             |␣
↪0-06          |
|              |               |          |                 | 17:23:...    | 13:2...      | 17:23:
↪00... | 17:23...      |
| 1            | 2010-02-13 | 0         | 2010-02-12 | {"input.en | {"input.st | {"gap.end" | {
↪"target.e |
|              | 20:16:13   |           | 20:16:13   | d",2010-02 | art",2010- | ,2010-02-1 |␣
↪nd",2010-0 |
|              |               |          |                 | -12          | 02-12       | 2             |␣
↪2-14          |
|              |               |          |                 | 20:16:...    | 20:1...      | 20:16:
↪13... | 20:16...      |
| 2            | 2010-01-19 | 0         | 2010-01-18 | {"input.en | {"input.st | {"gap.end" | {
↪"target.e |
|              | 23:07:07   |           | 23:07:07   | d",2010-01 | art",2010- | ,2010-01-1 |␣
↪nd",2010-0 |
|              |               |          |                 | -18          | 01-18       | 8             |␣
↪1-20          |
|              |               |          |                 | 23:07:...    | 23:0...      | 23:07:
↪07... | 23:07...      |
| 4            | 2010-06-30 | 0         | 2010-06-29 | {"input.en | {"input.st | {"gap.end" | {
↪"target.e |
|              | 07:20:14   |           | 07:20:14   | d",2010-06 | art",2010- | ,2010-06-2 |␣
↪nd",2010-0 |
|              |               |          |                 | -29          | 06-29       | 9             |␣
↪7-01          |
|              |               |          |                 | 07:20:...    | 07:2...      | 07:20:
↪14... | 07:20...      |
| 4            | 2010-08-03 | 0         | 2010-08-02 | {"input.en | {"input.st | {"gap.end" | {
↪"target.e |
|              | 14:20:14   |           | 14:20:14   | d",2010-08 | art",2010- | ,2010-08-0 |␣
↪nd",2010-0 |
|              |               |          |                 | -02          | 06-29       | 2             |␣
↪8-04          |
|              |               |          |                 | 14:20:...    | 07:2...      | 14:20:
↪14... | 14:20...      |
| ...          | ...           | ...      | ...             | ...          | ...         |␣
↪...           | ...           |
| 98           | 2010-06-29 | 0         | 2010-06-28 | {"input.en | {"input.st | {"gap.end" | {
↪"target.e |
|              | 22:25:52   |           | 22:25:52   | d",2010-06 | art",2010- | ,2010-06-2 |␣
↪nd",2010-0 |
|              |               |          |                 | -28          | 04-05       | 8             |␣
↪6-30          |
|              |               |          |                 | 22:25:...    | 19:2...      | 22:25:
↪52... | 22:25...      |
| 98           | 2010-08-29 | 0         | 2010-08-28 | {"input.en | {"input.st | {"gap.end" | {
↪"target.e |
|              | 00:25:52   |           | 00:25:52   | d",2010-08 | art",2010- | ,2010-08-2 |␣
↪nd",2010-0 |
|              |               |          |                 | -28          | 04-05       | 8             |␣
```

(continues on next page)

```
↪8-30        |
|           |           |      |           | 00:25:...  | 19:2...     | 00:25:
↪52... | 00:25...      |
| 99        | 2010-04-16 | 0    | 2010-04-15 | {"input.en | {"input.st | {"gap.end" | {
↪"target.e |
|           | 18:20:06   |      | 18:20:06   | d",2010-04 | art",2010- | ,2010-04-1 |⎵
↪nd",2010-0 |
|           |           |      |           | -15        | 04-15      | 5          |⎵
↪4-17       |
|           |           |      |           | 18:20:...  | 18:2...     | 18:20:
↪06... | 18:20...      |
| 99        | 2010-10-13 | 0    | 2010-10-12 | {"input.en | {"input.st | {"gap.end" | {
↪"target.e |
|           | 22:20:06   |      | 22:20:06   | d",2010-10 | art",2010- | ,2010-10-1 |⎵
↪nd",2010-1 |
|           |           |      |           | -12        | 04-15      | 2          |⎵
↪0-14       |
|           |           |      |           | 22:20:...  | 18:2...     | 22:20:
↪06... | 22:20...      |
| 99        | 2010-11-15 | 0    | 2010-11-14 | {"input.en | {"input.st | {"gap.end" | {
↪"target.e |
|           | 08:20:06   |      | 08:20:06   | d",2010-11 | art",2010- | ,2010-11-1 |⎵
↪nd",2010-1 |
|           |           |      |           | -14        | 04-15      | 4          |⎵
↪1-16       |
|           |           |      |           | 08:20:...  | 18:2...     | 08:20:
↪06... | 08:20...      |
└----+----+---+----+----+----+----+----+
```

… and that's a wrap! We have used ACES to perform an end-to-end extraction on a MEDS dataset for a cohort that can be used to predict in-hospital mortality. Similar pipelines can be made for other tasks, as well as using the ESGPT data standard. You may also pre-compute predicate columns and use the `direct` flag when loading in `.csv` or `.parquet` data files. More information about this is available in Predicates DataFrame.

As always, please don't hesitate to reach out should you have any questions about ACES!

# ACES TECHNICAL DETAILS

## 6.1 Configuration Language Specification

This document specifies the configuration language for the automatic extraction of task dataframes and cohorts from structured EHR data organized either via the MEDS format (recommended) or the ESGPT format. This extraction system works by defining a configuration object that details the underlying concepts, inclusion/exclusion, and labeling criteria for the cohort/task to be extracted, then using a recursive algorithm to identify all realizations of valid patient time-ranges of data that satisfy those constraints from the raw data. For more details on the recursive algorithm, see Algorithm Design.

As indicated above, these cohorts are specified through a combination of concepts (realized as event *predicate* functions, *aka* "predicates") which are *dataset specific* and inclusion/exclusion/labeling criteria which, conditioned on a set of predicate definitions, are *dataset agnostic*.

Predicates are currently limited to "count" predicates, which are predicates that count the number of times a boolean condition is satisfied over a given time window, which can either be a single timestamp, thus tracking whether how many observations there were that satisfied the boolean condition in that event (*aka* at that timestamp) or over 1-dimensional windows. In the future, predicates may expand to include other notions of functional characterization, such as tracking the average/min/max value a concept takes on over a time-period, etc.

Constraints are specified in terms of time-points that can be bounded by events that satisfy predicates or temporal relationships on said events. The windows between these time-points can then either be constrained to contain events that satisfy certain aggregation functions over predicates for these time frames.

---

In the machine form used by ACES, the configuration file consists of three parts:

- `predicates`, stored as a dictionary from string predicate names (which must be unique) to either `aces.config.PlainPredicateConfig` objects, which store raw predicates with no dependencies on other predicates, or `aces.config.DerivedPredicateConfig` objects, which store predicates that build on other predicates.

- `trigger`, stored as a string to `EventConfig`

- `windows`, stored as a dictionary from string window names (which must be unique) to `aces.config.WindowConfig` objects.

Below, we will detail each of these configuration objects.

---

### 6.1.1 Predicates: `PlainPredicateConfig` and `DerivedPredicateConfig`

**`aces.config.PlainPredicateConfig`: Configuration of Predicates that can be Computed Directly from Raw Data**

These configs consist of the following four fields:

- `code`: The string expression for the code object that is relevant for this predicate. An observation will only satisfy this predicate if there is an occurrence of this code in the observation. The field can additionally be a dictionary with either a `regex` key and the value being a regular expression (satisfied if the regular expression evaluates to True), or a `any` key and the value being a list of strings (satisfied if there is an occurrence for any code in the list).

  **Note**: Each individual definition of `PlainPredicateConfig` and `code` will generate a separate predicate column. Thus, for memory optimization, it is strongly recommended to match multiple values using either the List of Values or Regular Expression formats whenever possible.

- `value_min`: If specified, an observation will only satisfy this predicate if the occurrence of the underlying `code` with a reported numerical value that is either greater than or greater than or equal to `value_min` (with these options being decided on the basis of `value_min_inclusive`, where `value_min_inclusive=True` indicating that an observation satisfies this predicate if its value is greater than or equal to `value_min`, and `value_min_inclusive=False` indicating a greater than but not equal to will be used).

- `value_max`: If specified, an observation will only satisfy this predicate if the occurrence of the underlying `code` with a reported numerical value that is either less than or less than or equal to `value_max` (with these options being decided on the basis of `value_max_inclusive`, where `value_max_inclusive=True` indicating that an observation satisfies this predicate if its value is less than or equal to `value_max`, and `value_max_inclusive=False` indicating a less than but not equal to will be used).

- `value_min_inclusive`: See `value_min`

- `value_max_inclusive`: See `value_max`

- `other_cols`: This optional field accepts a 1-to-1 dictionary of column names to column values, and can be used to specify further constraints on other columns (ie., not `code`) for this predicate.

A given observation will be gauged to satisfy or fail to satisfy this predicate in one of two ways, depending on its source format.

1. If the source data is in MEDS format (recommended), then the `code` will be checked directly against MEDS' `code` field and the `value_min` and `value_max` constraints will be compared against MEDS' `numeric_value` field.

   **Note**: This syntax does not currently support defining predicates that also rely on matching other, optional fields in the MEDS syntax; if this is a desired feature for you, please let us know by filing a GitHub issue or pull request or upvoting any existing issue/PR that requests/implements this feature, and we will add support for this capability.

2. If the source data is in ESGPT format, then the `code` will be interpreted in the following manner: a. If the code contains a "//", it will be interpreted as being a two element list joined by the "//" character, with the first element specifying the name of the ESGPT measurement under consideration, which should either be of the multi-label classification or multivariate regression type, and the second element being the name of the categorical key corresponding to the code in question within the underlying measurement specified. If either of `value_min` and `value_max` are present, then this measurement must be of a multivariate regression type, and the corresponding `values_column` for extracting numerical observations from ESGPT's `dynamic_measurements_df` will be sourced from the ESGPT dataset configuration object. b. If the code does not contain a "//", it will be interpreted as a direct measurement name that must be of the univariate regression type and its value, if needed, will be pulled from the corresponding column.

### `aces.config.DerivedPredicateConfig`: Configuration of Predicates that Depend on Other Predicates

These configuration objects consist of only a single string field–`expr`–which contains a limited grammar of accepted operations that can be applied to other predicates, containing precisely the following:

- `and(pred_1_name, pred_2_name, ...)`: Asserts that all of the specified predicates must be true.

- `or(pred_1_name, pred_2_name, ...)`: Asserts that any of the specified predicates must be true.

**Note**: Currently, `and`'s and `or`'s cannot be nested. Upon user request, we may support further advanced analytic operations over predicates.

---

## 6.1.2 Events: `aces.config.EventConfig`

The event config consists of only a single field, `predicate`, which specifies the predicate that must be observed with value greater than one to satisfy the event. There can only be one defined "event" with an "EventConfig" in a valid configuration, and it will define the "trigger" event of the cohort.

The value of its field can be any defined predicate.

---

## 6.1.3 Windows: `aces.config.WindowConfig`

Windows contain a tracking `name` field, and otherwise are specified with two parts: (1) A set of four parameters (`start`, `end`, `start_inclusive`, and `end_inclusive`) that specify the time range of the window, and (2) a set of constraints specified through two fields, dictionary of constraints (the `has` field) that specify the constraints that must be satisfied over the defined predicates for a possible realization of this window to be valid.

### Time Range Fields

### `start` and `end`

Valid windows always progress in time from the `start` field to the `end` field. These two fields define, in symbolic form, the relationship between the start and end time of the window. These two fields must obey the following rules:

1. *Linkage to other windows*: Firstly, exactly one of these two fields must reference an external event, as specified either through the name of the trigger event or the start or end event of another window. The other field must either be `null`/`None`/omitted (which has a very specific meaning, to be explained shortly) or must reference the field that references the external event.

2. *Linkage reference language*: Secondly, for both events, regardless of whether they reference an external event or an internal event, that reference must be expressed in one of the following ways.

   1. `$REFERENCING = $REFERENCED + $TIME_DELTA`, `$REFERENCING = $REFERENCED - $TIME_DELTA`, etc. In this case, the referencing event (either the start or end of the window) will be defined as occurring exactly `$TIME_DELTA` either after or before the event being referenced (either the external event or the end or start of the window).

      **Note**: If `$REFERENCED` is the `start` field, then `$TIME_DELTA` must be positive, and if `$REFERENCED` is the `end` field, then `$TIME_DELTA` must be negative to preserve the time ordering of the window fields.

---

2. `$REFERENCING = $REFERENCED -> $PREDICATE,` $REFERENCING = $REFERENCED <-
`$PREDICATE` In this case, the referencing event will be defined as the next or previous event satisfying the predicate, `$PREDICATE`.

**Note**: If the `$REFERENCED` is the `start` field, then the "next predicate ordering" (`$REFERENCED ->
$PREDICATE`) must be used, and if the `$REFERENCED` is the `end` field, then the "previous predicate ordering" (`$REFERENCED <- $PREDICATE`) must be used to preserve the time ordering of the window fields. These forms can lead to windows being defined as single point events, if the `$REFERENCED` event itself satisfies `$PREDICATE` and the appropriate constraints are satisfied and inclusive values are set.

3. `$REFERENCING = $REFERENCED` In this case, the referencing event will be defined as the same event as the referenced event.

3. *null/None/omitted*: If `start` is `null/None`/omitted, then the window will start at the beginning of the patient's record. If `end` is `null/None`/omitted, then the window will end at the end of the patient's record. In either of these cases, the other field must reference an external event, per rule 1.

**start_inclusive and end_inclusive**

These two fields specify whether the start and end of the window are inclusive or exclusive, respectively. This applies both to whether they are included in the calculation of the predicate values over the windows, but also, in the `$REFERENCING = $REFERENCED -> $PREDICATE` and `$REFERENCING = $PREDICATE -> $REFERENCED` cases, to which events are possible to use for valid next or prior `$PREDICATE` events. E.g., if we have that `start_inclusive=False` and the `end` field is equal to `start -> $PREDICATE`, and it so happens that the `start` event itself satisfies `$PREDICATE`, the fact that `start_inclusive=False` will mean that we do not consider the `start` event itself to be a valid start to any window that ends at the same `start` event, as its timestamp when considered as the prospective "window start timestamp" occurs "after" the effective timestamp of itself when considered as the `$PREDICATE` event that marks the window end given that `start_inclusive=False` and thus we will think of the window as truly starting an iota after the timestamp of the `start` event itself.

**Constraints Field**

The constraints field is a dictionary that maps predicate names to tuples of the form `(min_valid, max_valid)` that define the valid range the count of observations of the named predicate that must be found in a window for it to be considered valid. Either `min_valid` or `max_valid` constraints can be `None`, in which case those endpoints are left unconstrained. Likewise, unreferenced predicates are also left unconstrained.

**Note**: As predicate counts are always integral, this specification does not need an additional inclusive/exclusive endpoint field, as one can simply increment the bound by one in the appropriate direction to achieve the result. Instead, this bound is always interpreted to be inclusive, so a window would satisfy the constraint for predicate `name` with constraint `name: (1, 2)` if the count of observations of predicate `name` in a window was either 1 or 2. All constraints in the dictionary must be satisfied on a window for it to be included.

## 6.2 Algorithm Overview

We will assume that we are given a dataframe `df` which details events that have happened to subjects. Each row in the dataframe will have a `subject_id` column which identifies the subject, and a `timestamp` column which identifies the timestamp at which the event the row is describing happened. `df` would be constructed to have unique `subject_id` and `timestamp` pairs.

We will also assume this dataframe has a collection of columns which describe the event in a variety of ways. These columns can either have a binary value (1/0) representing whether certain properties are True/False for each row's event, or a count (integer) for the number of times that certain properties hold within each row's event. We'll call

these additional properties/columns "predicates" over the events, as they can often be interpreted as boolean or count functions over the event.

For example, we may consider a dataframe `df_clinical_events` that quantifies clinical events happening to patients, with predicates `"admission"`, `"discharge"`, `"death"`, and `"covid_dx"`, like this:

| subject_id | timestamp | admission | discharge | death | covid_dx |
|------------|-----------|-----------|-----------|-------|----------|
| 1 | 2020-01-01 12:03:31 | 1 | 0 | 0 | 0 |
| 1 | 2020-01-01 12:33:01 | 0 | 0 | 0 | 0 |
| 1 | 2020-01-01 13:02:58 | 0 | 0 | 0 | 0 |
| 1 | 2020-01-01 15:00:00 | 0 | 0 | 0 | 0 |
| 1 | 2020-01-04 11:12:00 | 0 | 1 | 0 | 0 |
| 1 | 2022-04-22 07:45:00 | 0 | 0 | 1 | 0 |
| 2 | 2020-01-01 12:03:31 | 1 | 0 | 0 | 0 |
| 2 | 2020-01-02 10:18:29 | 0 | 0 | 0 | 0 |
| 2 | 2020-01-02 16:18:29 | 0 | 0 | 0 | 1 |
| 2 | 2020-01-03 14:47:31 | 0 | 0 | 1 | 0 |
| 3 | 2020-01-01 12:03:31 | 1 | 0 | 0 | 0 |
| 3 | 2020-01-02 12:03:31 | 0 | 1 | 0 | 0 |
| 3 | 2022-01-01 12:03:31 | 1 | 0 | 0 | 0 |
| 3 | 2022-01-06 12:03:31 | 0 | 0 | 1 | 0 |

In this case, we have 3 subjects (patients), which have the following respective approximate time series of events:

- Subject 1 is admitted, has 3 events that don't satisfy any predicates, is discharged, dies, and has no further events.

- Subject 2 is admitted, has an event that satisfies no predicate, has a COVID diagnosis, dies, and has no further events.

- Subject 3 is admitted, then is discharged, then is admitted again, and then dies.

Events that don't satisfy any predicates in this particular case could represent a variety of other events in the medical record, such as a lab test, a procedure, or a non-COVID diagnosis, just to name a few.

Given data like this, our algorithm is designed to extract valid start and end times of "windows" within a subject's time series that satisfy certain inclusion and exclusion criteria and are defined with temporal and event-bounded constraints. We can use this algorithm to automatically extract windows of interest from the record, including but not limited to data cohorts and downstream task labeled datasets for machine learning applications.

We will specify these windows using a configuration file language that is ultimately interpreted into a tree structure. For example, suppose we wish to extract a dataset for the prediction of in-hospital mortality from the data defined in the above `df_clinical_events` dataframe, such that we wish to include the first 24 hours of data of each hospitalization as an input to a model, and predict whether the patient will die within the hospital. Suppose we also subject the dataset to constraints where the admission in question must be at least 48 hours in length and that the patient must not have a COVID diagnosis within that admission.

We might then specify these windows using the defined predicates in the configuration file language as follows:

```
trigger: admission

windows:
  input:
    start:
    end: trigger + 24h
  gap:
    start: trigger
```

(continues on next page)

```
    end: start + 48h
    has:
      admission: (None, 0)
      discharge: (None, 0)
      death: (None, 0)
      covid_dx: (None, 0)
  target:
    start: gap.end
    end: start -> discharge_or_death
    has:
      covid_dx: (None, 0)
    label: death
```

Given that our machine learning model seeks to predict in-hospital mortality, our dataset should include both positive and negative samples (patients that died in the hospital and patients that didn't die). Hence, the `target` "window" concludes at either a `"death"` event (patients that died) or a `"discharge"` event (patients that didn't die).

We can see that this set of specifications can be realized in a "valid" form for a patient if there exist a set of time points such that, within 48 hours after an admission, there are no discharges, deaths, or COVID diagnoses, and that there exists a discharge or death event after the first 48 hours of an admission where there were no COVID diagnoses between the end of that first 48 hours and the subsequent discharge or death event.

These windows form a naturally hierarchical, tree-based structure based on their relative dependencies on one another. In particular, we can realize the following tree structure constructed by nodes inferred for the above configuration:

```
- Trigger
  - Gap Start (Trigger)
    - Gap End (Gap Start + 48h)
      - Target Start (Gap End)
        - Target End (subsequent "discharge" or "death")
  - Input End
```

Our algorithm will naturally rely on this hierarchical structure by performing a set of recursive database search operations to extract the windows that satisfy the constraints of the configuration file by recursing over each subtree to find windows that satisfy the constraints of those subtrees individually.

In the rest of this document, we will detail how our algorithm automatically extracts records that meet these criteria and the terminology we use to describe our algorithm (both here and in the raw source code and code comments). There are certain limitations of this algorithm where some kinds of tasks cannot yet be expressed directly (more information available in the FAQs and the Future Roadmap). Details about the true configuration language that is used in practice to specify "windows" can be found in /configuration. Some task examples are available in *Task Examples*.

## 6.3 Algorithm Terminology

### 6.3.1 Event

An "event" in our dataset is a unique timestamp that occurs for a given subject.

### 6.3.2 Predicate

A "predicate" is a boolean or count function that can be applied to an event to describe the observations that an underlying dataset included within the timestamp of that event. They will often be boolean functions at the beginning of the process, but become aggregated into count functions when summarizing windows, so will be thought of as count functions to capture this generality throughout the algorithm as it rarely, if ever, necessary to distinguish between the two.

### 6.3.3 Window

A "window" is just a time range capturing some portion of a subject's record. It can be inclusive or exclusive on either endpoint, and may or may not have endpoints corresponding to an extant event in the dataset, as opposed to a time point at which no event occurred.

Time is treated as strictly increasing in our algorithm (ie., the start of a "window" will always be before or equal to the end of that "window").

### 6.3.4 A "Root" of a Subtree

A subtree in the hierarchy of constraint windows has a "root" node in the tree, which corresponds to the start or end of a "window" in the set of constraints. For example, the "Gap End" node in the tree above is the root of the subtree `Gap End -> Target Start -> Target End`.

### 6.3.5 A "Realized" Subtree of Constraint Windows

A subtree in the hierarchy of constraint windows can be *realized* in a patient dataset by finding a set of timestamps such that the windows of events they bound satisfy the constraints of the subtree. For instance, using our example in-hospital mortality task above, the subtree `Gap End -> Target Start -> Target End` would be *realized* if, given the "Gap End" timestamp, we can find:

- A timestamp for "Target Start", which is equal to the timestamp of "Gap End" in this example.

- A timestamp for "Target End", which should be equal to the timestamp of a `"death"` or `"discharge"` event and there are no `"covid_dx"` events between the timestamp of "Target Start" and the timestamp of "Target End".

### 6.3.6 An "Anchor" or "Anchor Event" of a Subtree

A subtree in the hierarchy of constraint windows that can be *realized* in a real patient's record will have one **most recent** ancestor node whose timestamp will correspond to the timestamp of a real event in the patient record. This node is called the "anchor" of the subtree. For example, in any realization of the tree above, the admission event matched by the "Trigger" node will be the anchor of the realization of the `Gap End -> Target Start -> Target End` subtree, as the Gap End is defined via a relative time gap to the admission event and thus cannot be guaranteed to correspond to an extant event in the patient record. However, the admission event of the `Trigger` node will always correspond to an extant event in the patient record and exist in the dataset proper.

This notion of an *anchor* will be useful in the algorithm as it will correspond to rows from which we will perform temporal and event-based aggregations to determine whether windows satisfy subtree constraints.

## 6.4 Algorithm Design

### 6.4.1 I. Initialization

#### Inputs

During initialization, we will be given the following inputs:

#### `cfg`

`cfg` is a *aces.config.TaskExtractorConfig* object containing our task definition, include all information about predicates, the trigger event, and windows.

#### `predicates_df`

The `predicates_df` dataframe will contain all events and their predicates.

#### Computation

During initialization, we will first ensure that the predicates dataframe contains unique (`subject_id`, `timestamp`) pairs. This is to ensure that no memory leaks occur over mismatched/extra rows when joining dataframes.

#### Identify Prospective Root Anchors

Prior to summarizing the rest of the task tree, we first identify prospective root anchors by checking the constraints of the trigger event. The trigger event represents the node of the tree we aim to realize, and thus this first step can significantly filter our cohort.

#### Recurse over Each Subtree

With this dataframe, we can proceed to traverse the tree and recurse over each subtree rooted at each node.

### 6.4.2 II. Recursive Step

#### Inputs

In our recursive step, we will be given the following inputs:

**predicates_df**

The `predicates_df` dataframe will contain all events and their predicates. This will not be modified across recursive steps.

**subtree_anchor_to_subtree_root_df**

The `subtree_anchor_to_subtree_root_df` dataframe will contain rows corresponding to the timestamps of a superset of all possible valid anchor events for realizations of the subtree over which we are recursing (a superset, as if there exist no valid realizations of subtrees, then a prospective anchor would be invalid - if we can find a valid subtree realization for a prospective anchor in this input dataframe, said anchor would be a true valid anchor).

This dataframe will also contain the counts of predicates between the prospective anchor events indexed by the rows of this dataframe and the corresponding possible root timestamps of the subtree over which we are recursing. This information will be necessary to compute the proper counts within a "window" during the recursive step.

**offset**

In the event that the subtree root timestamp is not the same as the subtree anchor timestamp (there may be a temporal offset between the two), the `offset` will be the difference between the two timestamps. If the two are not the same, they will guaranteed to be separated by a constant `offset` because the subtree root will either correspond to a fixed time delta from the subtree anchor or will be an actual event itself, in which case it will be the subtree anchor.

## Computation

In the recursive step, we will iterate over all children of the subtree root node. For each child, we will do the following:

### Aggregate Predicates over the Relevant "Window"

First, we will aggregate the predicates from `predicates_df` over the rows corresponding to the "window" spanning the root of the subtree to the root of the selected child. This aggregation step will always return a dataframe keyed by the `subject_id` column as well as by any possible prospective realizations of anchor events for the *subtree rooted at the selected child node*. This computation will take one of two forms:

### Temporal Aggregation

If the edge linking the subtree root to the child is a temporal relationship (e.g., in our example above, the "Gap End" node is defined as a fixed time delta from the "Gap Start" node), we will aggregate the predicates by using a "rolling" (or "temporal" group-by) operation on the `predicates_df` dataframe, summarizing time windows of the appropriate size and grouping by the `subject_id` column. We will perform this aggregation globally over the predicates dataframe, leveraging the determined edge time delta and the passed `offset` parameter (such that we compute the aggregation over the correct "window" in time from any possible realization of the subtree anchor) and then filter the resulting dataframe to only include rows corresponding to said possible subtree anchors. As a temporal edge means the anchor of the child subtree is the same as the anchor of the passed subtree, this suffices for our intended computation, and we can return it directly.

**Event-bound Aggregation**

If the edge linking the subtree root to the child is an event-bound relationship (e.g., in our example above, the "Target End" node is defined as the first subsequent `"discharge"` or `"death"` event after the "Target Start" node), we will aggregate the predicates by using a custom row-predicate-bound aggregation over the database that will be implemented using differences of cumulative sums within the global `predicates_df` dataframe. In particular, we will first construct the following three dataframes from our inputs:

1. A dataframe that contains the cumulative count of all predicates seen up until each event (row) in the `predicates_df`.

2. A dataframe that contains nulls in each row that does not correspond to a possible prospective realization of a child anchor event given the edge constraints and the possible prospective subtree anchor events and the specified `offset`, and contains a `True` value otherwise.

3. A dataframe that contains nulls in each row that does not correspond to a possible prospective realization of this subtree's anchor event and contains a `True` value otherwise.

From these three dataframes, we can then forward fill (in time) the cumulative counts of each predicate seen at each prospective subtree anchor event up to the next subsequent possible child anchor node, and take the difference between the two to compute the relative counts for each predicate column between each successive pair of subtree anchor events and child anchor events, keyed by child anchor. We must then also subtract the counts seen between the subtree anchor events and the subtree root events to ensure we are only capturing the events between the correct subtree root and child root.

**Filter on Constraints**

Next, with these new "window" counts, we can validate that any inclusion or exclusion criteria are upheld, and if not, remove those subtrees as possible realizations of the "window" before proceeding to the next computational step.

**Recurse through Child Subtree**

With this filtered set of possible prospective child anchor nodes, we can now recurse through the child subtree.

---

## 6.4.3 III. Clean-Up

**Inputs**

After recursion, we will have a `result` dataframe:

**`result`**

This dataframe contains rows that represent valid realizations of the task tree. Each node of the tree will have a column with a `pl.Struct` object containing the name of the window the node represents, the start and end times of the window, and counts of all defined predicates.

### Computation

With this result, we can then proceed with some clean-up to optimize the output and streamline downstream tasks by doing the following:

### Labeling

If a `label` field is specified in exactly one defined window in the task configuration, a column will be created to serve as the label for the task. The field corresponds to a defined predicate, and as such, that predicate count for that window will be extracted.

### Indexing Timestamp

If an 'index_timestamp' field is specified in exactly one defined window in the task configuration, a column will be created to serve as an index for the output cohort. This timestamp can be manually specified to any start or end timestamp of any desired window; however, it should represent the timestamp at which point a prediction can be made (ie., at the end of the `input` windows).

### Matching Input Schemas

For queries on MEDS-formatted dataset, ACES will automatically typecast columns and filter dataframes appropriately to match the label schema defined in MEDS v0.3.

### Re-order & Return

Finally, given this dataframe, the algorithm will sort the columns by placing `subject_id`, `index_timestamp`, `label`, and `trigger` first, if available and in that order, followed by all other window summary columns in the order of a pre-order traversal of the task tree.

---

# COMPUTATIONAL PROFILE

To establish an overview of the computational profile of ACES, a collection of various common tasks was queried on the MIMIC-IV dataset in MEDS format.

The MIMIC-IV MEDS schema has approximately 50,000 patients per shard with an average of approximately 80,500,000 total event rows per shard over five shards.

All tests were executed on a Linux server with 36 cores and 340 GBs of RAM available. A single MEDS shard was used, which provides a bounded computational overview of ACES. For instance, if one shard costs $M$ memory and $T$ time, then $N$ shards may be executed in parallel with $N * M$ memory and $T$ time, or in series with $M$ memory and $T * N$ time.

| Task | # Patients | # Samples | Total Time (secs) | Max Memory (MiBs) |
|---|---|---|---|---|
| First 24h in-hospital mortality | 20,971 | 58,823 | 363.09 | 106,367.14 |
| First 48h in-hospital mortality | 18,847 | 60,471 | 364.62 | 108,913.95 |
| First 24h in-ICU mortality | 4,768 | 7,156 | 216.81 | 39,594.37 |
| First 48h in-ICU mortality | 4,093 | 7,112 | 217.98 | 39,451.86 |
| 30d post-hospital-discharge mortality | 28,416 | 68,547 | 182.91 | 30,434.86 |
| 30d re-admission | 18,908 | 464,821 | 367.41 | 106,064.04 |

# ACES

## 8.1 aces package

### 8.1.1 Subpackages

**aces.configs package**

**Module contents**

This subpackage contains the Hydra configuration groups for ACES, which can be used for `aces-cli`.

Configuration Group File Structure:

```
config/
├─ data/
│   ├─ single_file.yaml
│   ├─ defaults.yaml
│   ├─ sharded.yaml
├─ aces.yaml
```

`aces-cli` help message:

```
================= aces-cli ===================
Welcome to the command-line interface for ACES!

This end-to-end tool extracts a cohort from the external dataset based on a defined task␣
↪configuration
file and saves the output file(s). Several data standards are supported, including␣
↪`meds` (requires a
dataset in the MEDS format, either with a single shard or multiple shards), `esgpt`␣
↪(requires a dataset
in the ESGPT format), and `direct` (requires a pre-computed predicates dataframe as well␣
↪as a timestamp
format). Hydra multi-run (`-m`) and sweep capabilities are supported, and launchers can␣
↪be configured.

------------ Configuration Groups ------------
$APP_CONFIG_GROUPS
`data` is defaulted to `data=single_file`. Use `data=sharded` to enable extraction with␣
↪multiple shards
```

*(continues on next page)*

```
on MEDS.

----------------- Arguments ------------------
data.*:
    - path (required): path to the data directory if using MEDS with multiple shards or␣
↪ESGPT, or path to
    the data `.parquet` if using MEDS with a single shard, or path to the predicates␣
↪dataframe
    (`.csv` or `.parquet`) if using `direct`
    - standard (required): data standard, one of  'meds', 'esgpt', or 'direct'
    - ts_format (required if data.standard is 'direct'): timestamp format for the data
    - root (required, applicable when data=sharded): root directory for the data shards
    - shard (required, applicable when data=sharded): shard number of specific shard␣
↪from a MEDS dataset.

    Note: data.shard can be expanded using the `expand_shards` function. Please refer to
    https://eventstreamaces.readthedocs.io/en/latest/usage.html#multiple-shards and
    https://github.com/justin13601/ACES/blob/main/src/aces/expand_shards.py for more␣
↪information.

cohort_dir (required): cohort directory, used to automatically load configs, saving␣
↪results, and logging
cohort_name (required): cohort name, used to automatically load configs, saving results,␣
↪and logging
config_path (optional): path to the task configuration file, defaults to '<cohort_dir>/
↪<cohort_name>.yaml'
predicates_path (optional): path to a separate predicates-only configuration file for␣
↪overriding
output_filepath (optional): path to the output file, defaults to '<cohort_dir>/<cohort_
↪name>.parquet'

---------------- Default Config ----------------
$CONFIG
------------------------------------------------
All fields may be overridden via the command-line interface. For example:

    aces-cli cohort_name="..." cohort_dir="..." data.standard="..." data="..." data.root=
↪"..."
            "data.shard=$$(expand_shards .../...)" ...

For more information, visit: https://eventstreamaces.readthedocs.io/en/latest/usage.html

Powered by Hydra (https://hydra.cc)
Use --hydra-help to view Hydra specific help
================================================
```

## 8.1.2 Submodules

### aces.aggregate module

This module contains the functions for aggregating windows over conditional row bounds.

aces.aggregate.**aggregate_event_bound_window**(predicates_df: *DataFrame*, endpoint_expr:
                                    ToEventWindowBounds | *tuple[bool, str, bool,*
                                    *datetime.timedelta | None]*) → DataFrame

Aggregates `predicates_df` between each row plus an offset and the next per-subject matching event.

# TODO: Use https://hypothesis.readthedocs.io/en/latest/quickstart.html to test this function.

See the testing for `boolean_expr_bound_sum` for more comprehensive examples and test cases for the underlying API here, and the testing for `ToEventWindowBounds` for how the bounding syntax is converted into arguments for the `boolean_expr_bound_sum` function.

> **Parameters**
>
> > **predicates_df: DataFrame**
> >
> > > The dataframe containing the predicates. The input must be sorted in ascending order by timestamp within each subject group. It must contain the following columns:
> > >
> > > - A column `subject_id` which contains the subject ID.
> > >
> > > - A column `timestamp` which contains the timestamp at which the event contained in any given row occurred.
> > >
> > > - A set of "predicate" columns that contain counts of the number of times a given predicate is satisfied in the event contained in any given row.
> >
> > **endpoint_expr:** *ToEventWindowBounds* **| tuple[bool, str, bool, datetime.timedelta | None]**
> >
> > > The expression defining the event bound window endpoints. Can be specified as a tuple or a ToEventWindowBounds object, which is just a named tuple of the expected form. Said expected form is as follows:
> > >
> > > - The first element is a boolean indicating whether the start of the window is inclusive.
> > >
> > > - The second element is a string indicating the name of the column in which non-zero counts indicate the event is a valid "end event" of the window.
> > >
> > > - The third element is a boolean indicating whether the end of the window is inclusive.
> > >
> > > - The fourth element is a timedelta object indicating the offset from the timestamp of the row to the start of the window. The offset here can _only_ be positive.
>
> **Returns**
>
> > The dataframe that has been aggregated in an event bound manner according to the specified `endpoint_expr`. This aggregation means the following:
> >
> > - The output dataframe will contain the same number of rows and be in the same order (in terms of `subject_id` and `timestamp`) as the input dataframe.
> >
> > - The column names will also be the same as the input dataframe, plus there will be two additional column, `timestamp_at_end` that contains the timestamp of the end of the aggregation window for each row (or null if no such aggregation window exists) and `timestamp_at_start` that contains the timestamp at the start of the aggregation window corresponding to the row.

- The values in the predicate columns of the output dataframe will contain the sum of the values in the predicate columns of the input dataframe from the timestamp of the event in any given row plus the specified offset (`endpoint_expr[3]`) to the timestamp of the next row for that subject in the input dataframe such that the specified event predicate (`endpoint_expr[1]`) has a value greater than zero for that patient, less either the start or end of the window pending the `left_inclusive` (`endpoint_expr[0]`) and `right_inclusive` (`endpoint_expr[2]`) values. If there is no valid "next row" for a given event, the values in the predicate columns of the output dataframe will be 0, as the sum of an empty set is 0. The sum of the predicate columns over the rows in the original dataframe spanning each row's `timestamp + offset` to each row's `timestamp_at_end` should be exactly equal to the values in the predicate columns of the output dataframe.

**Raises**
> **ValueError** – If the offset is negative.

## Examples

```
>>> import polars as pl
>>> _ = pl.Config.set_tbl_width_chars(150)
>>> from datetime import datetime
>>> df = pl.DataFrame({
...     "subject_id": [1, 1, 1, 2, 2, 2, 2, 2],
...     "timestamp": [
...         # Subject 1
...         datetime(year=1989, month=12, day=1,  hour=12, minute=3),
...         datetime(year=1989, month=12, day=3,  hour=13, minute=14), # HAS EVENT␣
→BOUND
...         datetime(year=1989, month=12, day=5,  hour=15, minute=17),
...         # Subject 2
...         datetime(year=1989, month=12, day=2,  hour=12, minute=3),
...         datetime(year=1989, month=12, day=4,  hour=13, minute=14),
...         datetime(year=1989, month=12, day=6,  hour=15, minute=17), # HAS EVENT␣
→BOUND
...         datetime(year=1989, month=12, day=8,  hour=16, minute=22),
...         datetime(year=1989, month=12, day=10, hour=3,  minute=7),  # HAS EVENT␣
→BOUND
...     ],
...     "is_A": [1, 0, 1, 1, 1, 1, 0, 0],
...     "is_B": [0, 1, 0, 1, 0, 1, 1, 1],
...     "is_C": [0, 1, 0, 0, 0, 1, 0, 1],
... })
>>> aggregate_event_bound_window(df, ToEventWindowBounds(True, "is_C", True, None))
shape: (8, 7)
+----+-------+-------+-------+--+--+--+
| subject_id | timestamp           | timestamp_at_start  | timestamp_at_end    | is_
→A | is_B | is_C |
| ---        | ---                 | ---                 | ---                 | ---
→  | ---  | ---  |
| i64        | datetime[μs]        | datetime[μs]        | datetime[μs]        |␣
→i64  | i64  | i64  |
|----|-------|-------|-------|--|--|--|
| 1          | 1989-12-01 12:03:00 | 1989-12-01 12:03:00 | 1989-12-03 13:14:00 | 1 ␣
→  | 1    | 1    |
```

```
| 1          | 1989-12-03 13:14:00 | 1989-12-03 13:14:00 | 1989-12-03 13:14:00 | 0 ␣
↪  | 1     | 1     |
| 1          | 1989-12-05 15:17:00 | null                | null                | 0 ␣
↪  | 0     | 0     |
| 2          | 1989-12-02 12:03:00 | 1989-12-02 12:03:00 | 1989-12-06 15:17:00 | 3 ␣
↪  | 2     | 1     |
| 2          | 1989-12-04 13:14:00 | 1989-12-04 13:14:00 | 1989-12-06 15:17:00 | 2 ␣
↪  | 1     | 1     |
| 2          | 1989-12-06 15:17:00 | 1989-12-06 15:17:00 | 1989-12-06 15:17:00 | 1 ␣
↪  | 1     | 1     |
| 2          | 1989-12-08 16:22:00 | 1989-12-08 16:22:00 | 1989-12-10 03:07:00 | 0 ␣
↪  | 2     | 1     |
| 2          | 1989-12-10 03:07:00 | 1989-12-10 03:07:00 | 1989-12-10 03:07:00 | 0 ␣
↪  | 1     | 1     |
└────+───────+───────+───────+──+──+──+
>>> aggregate_event_bound_window(df, ToEventWindowBounds(True, "is_C", False, None))
shape: (8, 7)
+────+───────+───────+───────+──+──+──+
| subject_id | timestamp           | timestamp_at_start  | timestamp_at_end    | is_
↪A | is_B | is_C |
| ---        | ---                 | ---                 | ---                 | ---
↪  | ---   | ---   |
| i64        | datetime[μs]        | datetime[μs]        | datetime[μs]        |␣
↪i64   | i64   | i64   |
|────|───────|───────|───────|──|──|──|
| 1          | 1989-12-01 12:03:00 | 1989-12-01 12:03:00 | 1989-12-03 13:14:00 | 1 ␣
↪  | 0     | 0     |
| 1          | 1989-12-03 13:14:00 | null                | null                | 0 ␣
↪  | 0     | 0     |
| 1          | 1989-12-05 15:17:00 | null                | null                | 0 ␣
↪  | 0     | 0     |
| 2          | 1989-12-02 12:03:00 | 1989-12-02 12:03:00 | 1989-12-06 15:17:00 | 2 ␣
↪  | 1     | 0     |
| 2          | 1989-12-04 13:14:00 | 1989-12-04 13:14:00 | 1989-12-06 15:17:00 | 1 ␣
↪  | 0     | 0     |
| 2          | 1989-12-06 15:17:00 | 1989-12-06 15:17:00 | 1989-12-10 03:07:00 | 1 ␣
↪  | 2     | 1     |
| 2          | 1989-12-08 16:22:00 | 1989-12-08 16:22:00 | 1989-12-10 03:07:00 | 0 ␣
↪  | 1     | 0     |
| 2          | 1989-12-10 03:07:00 | null                | null                | 0 ␣
↪  | 0     | 0     |
└────+───────+───────+───────+──+──+──+
>>> aggregate_event_bound_window(df, ToEventWindowBounds(False, "is_C", True, None))
shape: (8, 7)
+────+───────+───────+───────+──+──+──+
| subject_id | timestamp           | timestamp_at_start  | timestamp_at_end    | is_
↪A | is_B | is_C |
| ---        | ---                 | ---                 | ---                 | ---
↪  | ---   | ---   |
| i64        | datetime[μs]        | datetime[μs]        | datetime[μs]        |␣
↪i64   | i64   | i64   |
|────|───────|───────|───────|──|──|──|
```

```
| 1         | 1989-12-01 12:03:00 | 1989-12-01 12:03:00 | 1989-12-03 13:14:00 | 0 ␣
↪   | 1     | 1     |
| 1         | 1989-12-03 13:14:00 | 1989-12-03 13:14:00 | 1989-12-03 13:14:00 | 0 ␣
↪   | 0     | 0     |
| 1         | 1989-12-05 15:17:00 | null                | null                | 0 ␣
↪   | 0     | 0     |
| 2         | 1989-12-02 12:03:00 | 1989-12-02 12:03:00 | 1989-12-06 15:17:00 | 2 ␣
↪   | 1     | 1     |
| 2         | 1989-12-04 13:14:00 | 1989-12-04 13:14:00 | 1989-12-06 15:17:00 | 1 ␣
↪   | 1     | 1     |
| 2         | 1989-12-06 15:17:00 | 1989-12-06 15:17:00 | 1989-12-06 15:17:00 | 0 ␣
↪   | 0     | 0     |
| 2         | 1989-12-08 16:22:00 | 1989-12-08 16:22:00 | 1989-12-10 03:07:00 | 0 ␣
↪   | 1     | 1     |
| 2         | 1989-12-10 03:07:00 | 1989-12-10 03:07:00 | 1989-12-10 03:07:00 | 0 ␣
↪   | 0     | 0     |
└────+───────+───────+───────+──+──+──+
>>> aggregate_event_bound_window(df, ToEventWindowBounds(True, "is_C", True,␣
↪timedelta(days=3)))
shape: (8, 7)
+────+───────+───────+───────+──+──+──+
| subject_id | timestamp           | timestamp_at_start  | timestamp_at_end    | is_
↪A | is_B | is_C |
| ---        | ---                 | ---                 | ---                 | ---
↪   | ---   | ---   |
| i64        | datetime[μs]        | datetime[μs]        | datetime[μs]        |␣
↪i64   | i64   | i64   |
|────|───────|───────|───────|──|──|──|
| 1         | 1989-12-01 12:03:00 | null                | null                | 0 ␣
↪   | 0     | 0     |
| 1         | 1989-12-03 13:14:00 | null                | null                | 0 ␣
↪   | 0     | 0     |
| 1         | 1989-12-05 15:17:00 | null                | null                | 0 ␣
↪   | 0     | 0     |
| 2         | 1989-12-02 12:03:00 | 1989-12-05 12:03:00 | 1989-12-06 15:17:00 | 1 ␣
↪   | 1     | 1     |
| 2         | 1989-12-04 13:14:00 | 1989-12-07 13:14:00 | 1989-12-10 03:07:00 | 0 ␣
↪   | 2     | 1     |
| 2         | 1989-12-06 15:17:00 | 1989-12-09 15:17:00 | 1989-12-10 03:07:00 | 0 ␣
↪   | 1     | 1     |
| 2         | 1989-12-08 16:22:00 | null                | null                | 0 ␣
↪   | 0     | 0     |
| 2         | 1989-12-10 03:07:00 | null                | null                | 0 ␣
↪   | 0     | 0     |
└────+───────+───────+───────+──+──+──+
>>> aggregate_event_bound_window(df, (True, "is_C", True, timedelta(days=3)))
shape: (8, 7)
+────+───────+───────+───────+──+──+──+
| subject_id | timestamp           | timestamp_at_start  | timestamp_at_end    | is_
↪A | is_B | is_C |
| ---        | ---                 | ---                 | ---                 | ---
↪   | ---   | ---   |
```

```
| i64         | datetime[μs]       | datetime[μs]       | datetime[μs]       |␣
↪i64  | i64  | i64  |
|----|-------|-------|-------|--|--|--|
| 1          | 1989-12-01 12:03:00 | null                | null               | 0 ␣
↪ | 0   | 0    |
| 1          | 1989-12-03 13:14:00 | null                | null               | 0 ␣
↪ | 0   | 0    |
| 1          | 1989-12-05 15:17:00 | null                | null               | 0 ␣
↪ | 0   | 0    |
| 2          | 1989-12-02 12:03:00 | 1989-12-05 12:03:00 | 1989-12-06 15:17:00 | 1 ␣
↪ | 1   | 1    |
| 2          | 1989-12-04 13:14:00 | 1989-12-07 13:14:00 | 1989-12-10 03:07:00 | 0 ␣
↪ | 2   | 1    |
| 2          | 1989-12-06 15:17:00 | 1989-12-09 15:17:00 | 1989-12-10 03:07:00 | 0 ␣
↪ | 1   | 1    |
| 2          | 1989-12-08 16:22:00 | null                | null               | 0 ␣
↪ | 0   | 0    |
| 2          | 1989-12-10 03:07:00 | null                | null               | 0 ␣
↪ | 0   | 0    |
└----+-------+-------+-------+--+--+--+
```

aces.aggregate.**aggregate_temporal_window**(predicates_df: *DataFrame*, endpoint_expr:
TemporalWindowBounds | *tuple[bool, datetime.timedelta,
bool, datetime.timedelta | None]*) → DataFrame

Aggregates the predicates dataframe into the specified temporal buckets.

# TODO: Use https://hypothesis.readthedocs.io/en/latest/quickstart.html to add extra tests.

> **Parameters**
>
>> **predicates_df: DataFrame**
>>
>>> The dataframe containing the predicates. The input must be sorted in ascending order by timestamp within each subject group. It must contain the following columns:
>>>
>>> - A column `subject_id` which contains the subject ID.
>>>
>>> - A column `timestamp` which contains the timestamp at which the event contained in any given row occurred.
>>>
>>> - A set of "predicate" columns that contain counts of the number of times a given predicate is satisfied in the event contained in any given row.
>>
>> **endpoint_expr:** *TemporalWindowBounds* **| tuple[bool, datetime.timedelta, bool, datetime.timedelta | None]**
>>
>>> The expression defining the temporal window endpoints. Can be specified as a tuple or a TemporalWindowBounds object, which is just a named tuple of the expected form. Said expected form is as follows:
>>>
>>> - The first element is a boolean indicating whether the start of the window is inclusive.
>>>
>>> - The second element is a timedelta object indicating the size of the window.
>>>
>>> - The third element is a boolean indicating whether the end of the window is inclusive.
>>>
>>> - The fourth element is a timedelta object indicating the offset from the timestamp of the row to the start of the window.

**Returns**

The dataframe that has been aggregated temporally according to the specified `endpoint_expr`. This dataframe will contain the same number of rows and be in the same order (in terms of `subject_id` and `timestamp`) as the input dataframe. The column names will also be the same as the input dataframe, but the values in the predicate columns of the output dataframe will be the sum of the values in the predicate columns of the input dataframe from the timestamp of the event in the row of the output dataframe spanning the specified temporal window.

The dataframe that has been aggregated in a temporal manner according to the specified `endpoint_expr`. This aggregation means the following:

- The output dataframe will contain the same number of rows and be in the same order (in terms of `subject_id` and `timestamp`) as the input dataframe.

- The column names will also be the same as the input dataframe, plus there will be two additional column, `timestamp_at_end` that contains the timestamp of the end of the aggregation window for each row (or null if no such aggregation window exists) and `timestamp_at_start` that contains the timestamp at the start of the aggregation window corresponding to the row.

- The values in the predicate columns of the output dataframe will contain the sum of the values in the predicate columns of the input dataframe from the timestamp of the event in any given row plus the specified offset (`endpoint_expr[3]`) to the timestamp of the given row plus the offset plus the window size (`endpoint_expr[1]`), less either the start or end of the window pending the `left_inclusive` (`endpoint_expr[0]`) and `right_inclusive` (`endpoint_expr[2]`) values. The sum of the predicate columns over the rows in the original dataframe spanning each row's `timestamp + offset` to each row's `timestamp + offset + window_size` should be exactly equal to the values in the predicate columns of the output dataframe (again, less the left or right inclusive values as specified).

**Examples**

```
>>> import polars as pl
>>> _ = pl.Config.set_tbl_width_chars(150)
>>> from datetime import datetime, timedelta
>>> df = pl.DataFrame({
...     "subject_id": [1, 1, 1, 1, 2, 2],
...     "timestamp": [
...         # Subject 1
...         datetime(year=1989, month=12, day=1, hour=12, minute=3),
...         datetime(year=1989, month=12, day=2, hour=5,  minute=17),
...         datetime(year=1989, month=12, day=2, hour=12, minute=3),
...         datetime(year=1989, month=12, day=6, hour=11, minute=0),
...         # Subject 2
...         datetime(year=1989, month=12, day=1, hour=13, minute=14),
...         datetime(year=1989, month=12, day=3, hour=15, minute=17),
...     ],
...     "is_A": [1, 0, 1, 0, 0, 0],
...     "is_B": [0, 1, 0, 1, 1, 0],
...     "is_C": [1, 1, 0, 0, 1, 0],
... })
>>> aggregate_temporal_window(df, TemporalWindowBounds(
... True, timedelta(days=7), True, None))
shape: (6, 7)
```

```
+----+-------+-------+-------+--+--+--+
| subject_id | timestamp          | timestamp_at_start | timestamp_at_end   | is_
↪A | is_B | is_C |
| ---        | ---                | ---                | ---                | ---
↪  | --- | --- |
| i64        | datetime[μs]       | datetime[μs]       | datetime[μs]       |␣
↪i64  | i64 | i64 |
|----|-------|-------|-------|--|--|--|
| 1          | 1989-12-01 12:03:00 | 1989-12-01 12:03:00 | 1989-12-08 12:03:00 | 2 ␣
↪  | 2    | 2    |
| 1          | 1989-12-02 05:17:00 | 1989-12-02 05:17:00 | 1989-12-09 05:17:00 | 1 ␣
↪  | 2    | 1    |
| 1          | 1989-12-02 12:03:00 | 1989-12-02 12:03:00 | 1989-12-09 12:03:00 | 1 ␣
↪  | 1    | 0    |
| 1          | 1989-12-06 11:00:00 | 1989-12-06 11:00:00 | 1989-12-13 11:00:00 | 0 ␣
↪  | 1    | 0    |
| 2          | 1989-12-01 13:14:00 | 1989-12-01 13:14:00 | 1989-12-08 13:14:00 | 0 ␣
↪  | 1    | 1    |
| 2          | 1989-12-03 15:17:00 | 1989-12-03 15:17:00 | 1989-12-10 15:17:00 | 0 ␣
↪  | 0    | 0    |
└----+-------+-------+-------+--+--+--+
>>> aggregate_temporal_window(df, (
... True, timedelta(days=1), True, timedelta(days=0)))
shape: (6, 7)
+----+-------+-------+-------+--+--+--+
| subject_id | timestamp          | timestamp_at_start | timestamp_at_end   | is_
↪A | is_B | is_C |
| ---        | ---                | ---                | ---                | ---
↪  | --- | --- |
| i64        | datetime[μs]       | datetime[μs]       | datetime[μs]       |␣
↪i64  | i64 | i64 |
|----|-------|-------|-------|--|--|--|
| 1          | 1989-12-01 12:03:00 | 1989-12-01 12:03:00 | 1989-12-02 12:03:00 | 2 ␣
↪  | 1    | 2    |
| 1          | 1989-12-02 05:17:00 | 1989-12-02 05:17:00 | 1989-12-03 05:17:00 | 1 ␣
↪  | 1    | 1    |
| 1          | 1989-12-02 12:03:00 | 1989-12-02 12:03:00 | 1989-12-03 12:03:00 | 1 ␣
↪  | 0    | 0    |
| 1          | 1989-12-06 11:00:00 | 1989-12-06 11:00:00 | 1989-12-07 11:00:00 | 0 ␣
↪  | 1    | 0    |
| 2          | 1989-12-01 13:14:00 | 1989-12-01 13:14:00 | 1989-12-02 13:14:00 | 0 ␣
↪  | 1    | 1    |
| 2          | 1989-12-03 15:17:00 | 1989-12-03 15:17:00 | 1989-12-04 15:17:00 | 0 ␣
↪  | 0    | 0    |
└----+-------+-------+-------+--+--+--+
>>> aggregate_temporal_window(df, (
... True, timedelta(days=1), False, timedelta(days=0)))
shape: (6, 7)
+----+-------+-------+-------+--+--+--+
| subject_id | timestamp          | timestamp_at_start | timestamp_at_end   | is_
↪A | is_B | is_C |
| ---        | ---                | ---                | ---                | ---
```

```
→   | ---    | ---    |
| i64        | datetime[μs]        | datetime[μs]        | datetime[μs]        |␣
→i64  | i64  | i64  |
|----|-------|-------|-------|--|--|--|
| 1          | 1989-12-01 12:03:00 | 1989-12-01 12:03:00 | 1989-12-02 12:03:00 | 1 ␣
→   | 1     | 2     |
| 1          | 1989-12-02 05:17:00 | 1989-12-02 05:17:00 | 1989-12-03 05:17:00 | 1 ␣
→   | 1     | 1     |
| 1          | 1989-12-02 12:03:00 | 1989-12-02 12:03:00 | 1989-12-03 12:03:00 | 1 ␣
→   | 0     | 0     |
| 1          | 1989-12-06 11:00:00 | 1989-12-06 11:00:00 | 1989-12-07 11:00:00 | 0 ␣
→   | 1     | 0     |
| 2          | 1989-12-01 13:14:00 | 1989-12-01 13:14:00 | 1989-12-02 13:14:00 | 0 ␣
→   | 1     | 1     |
| 2          | 1989-12-03 15:17:00 | 1989-12-03 15:17:00 | 1989-12-04 15:17:00 | 0 ␣
→   | 0     | 0     |
└----+-------+-------+-------+--+--+--+
>>> aggregate_temporal_window(df, (
... False, timedelta(days=1), False, timedelta(days=0)))
shape: (6, 7)
+----+-------+-------+-------+--+--+--+
| subject_id | timestamp           | timestamp_at_start  | timestamp_at_end    | is_
→A | is_B | is_C |
| ---        | ---                 | ---                 | ---                 | ---
→   | ---   | ---   |
| i64        | datetime[μs]        | datetime[μs]        | datetime[μs]        |␣
→i64  | i64  | i64  |
|----|-------|-------|-------|--|--|--|
| 1          | 1989-12-01 12:03:00 | 1989-12-01 12:03:00 | 1989-12-02 12:03:00 | 0 ␣
→   | 1     | 1     |
| 1          | 1989-12-02 05:17:00 | 1989-12-02 05:17:00 | 1989-12-03 05:17:00 | 1 ␣
→   | 0     | 0     |
| 1          | 1989-12-02 12:03:00 | 1989-12-02 12:03:00 | 1989-12-03 12:03:00 | 0 ␣
→   | 0     | 0     |
| 1          | 1989-12-06 11:00:00 | 1989-12-06 11:00:00 | 1989-12-07 11:00:00 | 0 ␣
→   | 0     | 0     |
| 2          | 1989-12-01 13:14:00 | 1989-12-01 13:14:00 | 1989-12-02 13:14:00 | 0 ␣
→   | 0     | 0     |
| 2          | 1989-12-03 15:17:00 | 1989-12-03 15:17:00 | 1989-12-04 15:17:00 | 0 ␣
→   | 0     | 0     |
└----+-------+-------+-------+--+--+--+
>>> aggregate_temporal_window(df, (
... False, timedelta(days=-1), False, timedelta(days=0)))
shape: (6, 7)
+----+-------+-------+-------+--+--+--+
| subject_id | timestamp           | timestamp_at_start  | timestamp_at_end    | is_
→A | is_B | is_C |
| ---        | ---                 | ---                 | ---                 | ---
→   | ---   | ---   |
| i64        | datetime[μs]        | datetime[μs]        | datetime[μs]        |␣
→i64  | i64  | i64  |
|----|-------|-------|-------|--|--|--|
```

```
| 1         | 1989-12-01 12:03:00 | 1989-12-01 12:03:00 | 1989-11-30 12:03:00 | 0 ␣
↪ | 0     | 0     |
| 1         | 1989-12-02 05:17:00 | 1989-12-02 05:17:00 | 1989-12-01 05:17:00 | 1 ␣
↪ | 0     | 1     |
| 1         | 1989-12-02 12:03:00 | 1989-12-02 12:03:00 | 1989-12-01 12:03:00 | 0 ␣
↪ | 1     | 1     |
| 1         | 1989-12-06 11:00:00 | 1989-12-06 11:00:00 | 1989-12-05 11:00:00 | 0 ␣
↪ | 0     | 0     |
| 2         | 1989-12-01 13:14:00 | 1989-12-01 13:14:00 | 1989-11-30 13:14:00 | 0 ␣
↪ | 0     | 0     |
| 2         | 1989-12-03 15:17:00 | 1989-12-03 15:17:00 | 1989-12-02 15:17:00 | 0 ␣
↪ | 0     | 0     |
└────+───────+───────+───────+──+──+──+
>>> aggregate_temporal_window(df, (
... False, timedelta(hours=12), False, timedelta(hours=12)))
shape: (6, 7)
+────+───────+───────+───────+──+──+──+
| subject_id | timestamp           | timestamp_at_start  | timestamp_at_end    | is_
↪A | is_B | is_C |
| ---        | ---                 | ---                 | ---                 | ---
↪ | ---   | ---   |
| i64        | datetime[μs]        | datetime[μs]        | datetime[μs]        |␣
↪i64   | i64   | i64   |
|────|───────|───────|───────|──|──|──|
| 1         | 1989-12-01 12:03:00 | 1989-12-02 00:03:00 | 1989-12-02 12:03:00 | 0 ␣
↪ | 1     | 1     |
| 1         | 1989-12-02 05:17:00 | 1989-12-02 17:17:00 | 1989-12-03 05:17:00 | 0 ␣
↪ | 0     | 0     |
| 1         | 1989-12-02 12:03:00 | 1989-12-03 00:03:00 | 1989-12-03 12:03:00 | 0 ␣
↪ | 0     | 0     |
| 1         | 1989-12-06 11:00:00 | 1989-12-06 23:00:00 | 1989-12-07 11:00:00 | 0 ␣
↪ | 0     | 0     |
| 2         | 1989-12-01 13:14:00 | 1989-12-02 01:14:00 | 1989-12-02 13:14:00 | 0 ␣
↪ | 0     | 0     |
| 2         | 1989-12-03 15:17:00 | 1989-12-04 03:17:00 | 1989-12-04 15:17:00 | 0 ␣
↪ | 0     | 0     |
└────+───────+───────+───────+──+──+──+
>>> # Note that left_inclusive and right_inclusive are relative to the temporal␣
↪ordering of the window
>>> # and not the timestamp of the row. E.g., if left_inclusive is False, the␣
↪window will not include
>>> # the earliest event in the aggregation window, regardless of whether that is␣
↪earlier than the
>>> # timestamp of the row.
>>> aggregate_temporal_window(df, (
... False, timedelta(days=-1), True, timedelta(days=1)))
shape: (6, 7)
+────+───────+───────+───────+──+──+──+
| subject_id | timestamp           | timestamp_at_start  | timestamp_at_end    | is_
↪A | is_B | is_C |
| ---        | ---                 | ---                 | ---                 | ---
↪ | ---   | ---   |
```

```
| i64         | datetime[μs]        | datetime[μs]        | datetime[μs]        |␣
→i64  | i64  | i64  |
|----|-------|-------|-------|--|--|--|
| 1           | 1989-12-01 12:03:00 | 1989-12-02 12:03:00 | 1989-12-01 12:03:00 | 1 ␣
↪  | 1   | 1    |
| 1           | 1989-12-02 05:17:00 | 1989-12-03 05:17:00 | 1989-12-02 05:17:00 | 1 ␣
↪  | 0   | 0    |
| 1           | 1989-12-02 12:03:00 | 1989-12-03 12:03:00 | 1989-12-02 12:03:00 | 0 ␣
↪  | 0   | 0    |
| 1           | 1989-12-06 11:00:00 | 1989-12-07 11:00:00 | 1989-12-06 11:00:00 | 0 ␣
↪  | 0   | 0    |
| 2           | 1989-12-01 13:14:00 | 1989-12-02 13:14:00 | 1989-12-01 13:14:00 | 0 ␣
↪  | 0   | 0    |
| 2           | 1989-12-03 15:17:00 | 1989-12-04 15:17:00 | 1989-12-03 15:17:00 | 0 ␣
↪  | 0   | 0    |
└----+-------+-------+-------+--+--+--+
>>> aggregate_temporal_window(df, (
... True, timedelta(days=-1), False, timedelta(days=1)))
shape: (6, 7)
+----+-------+-------+-------+--+--+--+
| subject_id | timestamp           | timestamp_at_start  | timestamp_at_end    | is_
↪A | is_B | is_C |
| ---        | ---                 | ---                 | ---                 | ---
↪  | --- | ---  |
| i64         | datetime[μs]        | datetime[μs]        | datetime[μs]        |␣
→i64  | i64  | i64  |
|----|-------|-------|-------|--|--|--|
| 1           | 1989-12-01 12:03:00 | 1989-12-02 12:03:00 | 1989-12-01 12:03:00 | 1 ␣
↪  | 1   | 2    |
| 1           | 1989-12-02 05:17:00 | 1989-12-03 05:17:00 | 1989-12-02 05:17:00 | 1 ␣
↪  | 1   | 1    |
| 1           | 1989-12-02 12:03:00 | 1989-12-03 12:03:00 | 1989-12-02 12:03:00 | 1 ␣
↪  | 0   | 0    |
| 1           | 1989-12-06 11:00:00 | 1989-12-07 11:00:00 | 1989-12-06 11:00:00 | 0 ␣
↪  | 1   | 0    |
| 2           | 1989-12-01 13:14:00 | 1989-12-02 13:14:00 | 1989-12-01 13:14:00 | 0 ␣
↪  | 1   | 1    |
| 2           | 1989-12-03 15:17:00 | 1989-12-04 15:17:00 | 1989-12-03 15:17:00 | 0 ␣
↪  | 0   | 0    |
└----+-------+-------+-------+--+--+--+
```

aces.aggregate.**boolean_expr_bound_sum**(df: *DataFrame*, boundary_expr: *Expr*, mode: *str*, closed: *str*, offset: *timedelta* = `datetime.timedelta(0)`) → DataFrame

Sums all columns of `df` between each row plus an offset and the next per-subject satisfying event.

# TODO: Use https://hypothesis.readthedocs.io/en/latest/quickstart.html to test this function.

Performs a boolean-expression-bounded summation over the columns of `df`. The logic of this is as follows.

- If mode is `bound_to_row`, then that means that the _left_ endpoint of the window must correspond to an instance where `boundary_expr` is `True` and the _right_ endpoint will correspond to a row in the dataframe.

- If mode is `row_to_bound`, then that means that the _right_ endpoint of the window must correspond to an instance where `boundary_expr` is `True` and the _left_ endpoint will correspond to a row in the dataframe.

- If `closed` is `'none'`, then neither the associated left row endpoint nor the associated right row endpoint will be included in the summation.

- If `closed` is `'both'`, then both the associated left row endpoint and the associated right row endpoint will be included in the summation.

- If `closed` is `'left'`, then only the associated left row endpoint will be included in the summation.

- If `closed` is `'right'`, then only the associated right row endpoint will be included in the summation.

- The output dataframe will have the same number of rows and order as the input dataframe. Each row will correspond to the aggregation over the matching window that uses that row as the "row" side of the terminating aggregation (regardless of whether that corresponds to the left or the right endpoint, and regardless of whether that endpoint would actually be included in the calculation based on `closed`). The associated `boundary_expr` side of the endpoint will be the nearest possible `boundary_expr` endpoint that produces a non-empty set of rows over which to aggregate. Note that this may depend on the value of `closed` – e.g., a row with `boundary_expr` being True can produce a one-element set of rows to aggregate if `closed` is `'both'`, but this is not possible if `closed` is something else, and in that case that same row may instead rely on a different row to fill its `boundary_expr` endpoint when evaluated as a "row endpoint" during calculation.

- Offset further modifies this logic by applying a temporal offset of fixed size to the "row" endpoint. All other logic stays the same.

In particular, suppose that we have following rows and boolean boundary expression evaluations (for a single subject):

`markdown Rows: [0, 1, 2, 3, 4, 5, 6] Boundary Expression: [False, True, False, True, True, False, False] `

Then, we would aggregate the following rows under the following specified conditions:

`markdown mode | closed | aggregate_groups -------------|--------|----------------------------- bound_to_row | both | [], [1], [1, 2], [3], [4], [4, 5], [4, 5, 6] bound_to_row | left | [], [], [1], [1, 2], [3], [4], [4, 5] bound_to_row | right | [], [], [2], [2, 3], [4], [5], [5, 6] bound_to_row | none | [], [], [], [2], [], [], [5] row_to_bound | both | [0, 1], [1], [2, 3], [3], [4], [], [] row_to_bound | left | [0], [], [2], [], [], [], [] row_to_bound | right | [1], [2, 3], [3], [4], [], [], [] row_to_bound | none | [], [2], [], [], [], [], [] `

How to think about this? the `closed` parameter controls where we put the endpoints to merge for our bounds. Consider the case accented with ** above, where we have the row being row 1 (indexing from 0), we are in a `row_to_bound` setting, and we have `closed = "none"`. For this, as the left endpoint (here the row, as we are in `row_to_bound` is not included, so our boundary on the left is between row 2 and 3. Our boundary on the right is just to the left of the next row which has the True boolean value. In this case, that means between row 2 and 3 (as row 1 is to the left of our left endpoint). In contrast, if closed were `"left"`, then row 1 would be a possible bound row to use on the right, and thus by virtue of our right endpoint being open, we would include no rows.

**Parameters**

**df: DataFrame**

The dataframe to be aggregated. The input must be sorted in ascending order by timestamp within each subject group. It must contain the following columns:

- A column `subject_id` which contains the subject ID.

- A column `timestamp` which contains the timestamp at which the event contained in any given row occurred.

- A set of other columns that will be summed over the specified windows.

**boundary_expr: Expr**

> A boolean expression which can be evaluated over the passed dataframe, and defines rows that can serve as valid left or right boundaries for the aggregation step. The precise window over which the dataframe will be aggregated will depend on `mode` and `closed`. See above for an explanation of the various configurations possible.

**mode: str**

**closed: str**

**offset: timedelta** = `datetime.timedelta(0)`

**Returns**

> The dataframe that has been aggregated in an event bound manner according to the specified `endpoint_expr`. This aggregation means the following:
>
> - The output dataframe will contain the same number of rows and be in the same order (in terms of `subject_id` and `timestamp`) as the input dataframe.
>
> - The column names will also be the same as the input dataframe, plus there will be two additional column, `timestamp_at_end` that contains the timestamp of the end of the aggregation window for each row (or null if no such aggregation window exists) and `timestamp_at_start` that contains the timestamp at the start of the aggregation window corresponding to the row.
>
> - The values in the predicate columns of the output dataframe will contain the sum of the values over the permissible row ranges given the input parameters. See above for an explanation.

**Examples**

```
>>> import polars as pl
>>> _ = pl.Config.set_tbl_width_chars(150)
>>> from datetime import datetime
>>> df = pl.DataFrame({
...     "subject_id": [1, 1, 1, 2, 2, 2, 2, 2],
...     "timestamp": [
...         # Subject 1
...         datetime(year=1989, month=12, day=1,  hour=12, minute=3),
...         datetime(year=1989, month=12, day=3,  hour=13, minute=14), # HAS EVENT
→BOUND
...         datetime(year=1989, month=12, day=5,  hour=15, minute=17),
...         # Subject 2
...         datetime(year=1989, month=12, day=2,  hour=12, minute=3),
...         datetime(year=1989, month=12, day=4,  hour=13, minute=14),
...         datetime(year=1989, month=12, day=6,  hour=15, minute=17), # HAS EVENT
→BOUND
...         datetime(year=1989, month=12, day=8,  hour=16, minute=22),
...         datetime(year=1989, month=12, day=10, hour=3,  minute=7),  # HAS EVENT
→BOUND
...     ],
...     "idx":  [0, 1, 2, 3, 4, 5, 6, 7],
...     "is_A": [1, 0, 1, 1, 1, 1, 0, 0],
...     "is_B": [0, 1, 0, 1, 0, 1, 1, 1],
...     "is_C": [0, 1, 0, 0, 0, 1, 0, 1],
... })
```

```
>>> boolean_expr_bound_sum(
...     df,
...     pl.col("idx").is_in([1, 4, 7]),
...     "bound_to_row",
...     "both",
... ).drop("idx")
shape: (8, 7)
+------------+-------+-------+-------+--+--+--+
| subject_id | timestamp          | timestamp_at_start | timestamp_at_end   | is_
↪A | is_B | is_C |
| ---        | ---                | ---                | ---                | ---
↪   | ---  | ---  |
| i64        | datetime[μs]       | datetime[μs]       | datetime[μs]       |␣
↪i64  | i64  | i64  |
|----|-------|-------|-------|--|--|--|
| 1          | 1989-12-01 12:03:00 | null               | null               | 0 ␣
↪   | 0    | 0    |
| 1          | 1989-12-03 13:14:00 | 1989-12-03 13:14:00 | 1989-12-03 13:14:00 | 0 ␣
↪   | 1    | 1    |
| 1          | 1989-12-05 15:17:00 | 1989-12-03 13:14:00 | 1989-12-05 15:17:00 | 1 ␣
↪   | 1    | 1    |
| 2          | 1989-12-02 12:03:00 | null               | null               | 0 ␣
↪   | 0    | 0    |
| 2          | 1989-12-04 13:14:00 | 1989-12-04 13:14:00 | 1989-12-04 13:14:00 | 1 ␣
↪   | 0    | 0    |
| 2          | 1989-12-06 15:17:00 | 1989-12-04 13:14:00 | 1989-12-06 15:17:00 | 2 ␣
↪   | 1    | 1    |
| 2          | 1989-12-08 16:22:00 | 1989-12-04 13:14:00 | 1989-12-08 16:22:00 | 2 ␣
↪   | 2    | 1    |
| 2          | 1989-12-10 03:07:00 | 1989-12-10 03:07:00 | 1989-12-10 03:07:00 | 0 ␣
↪   | 1    | 1    |
└----+-------+-------+-------+--+--+--+
>>> boolean_expr_bound_sum(
...     df,
...     pl.col("idx").is_in([1, 4, 7]),
...     "bound_to_row",
...     "none",
... ).drop("idx")
shape: (8, 7)
+------------+-------+-------+-------+--+--+--+
| subject_id | timestamp          | timestamp_at_start | timestamp_at_end   | is_
↪A | is_B | is_C |
| ---        | ---                | ---                | ---                | ---
↪   | ---  | ---  |
| i64        | datetime[μs]       | datetime[μs]       | datetime[μs]       |␣
↪i64  | i64  | i64  |
|----|-------|-------|-------|--|--|--|
| 1          | 1989-12-01 12:03:00 | null               | null               | 0 ␣
↪   | 0    | 0    |
| 1          | 1989-12-03 13:14:00 | null               | null               | 0 ␣
↪   | 0    | 0    |
| 1          | 1989-12-05 15:17:00 | 1989-12-03 13:14:00 | 1989-12-05 15:17:00 | 0 ␣
```

```
→  | 0     | 0     |
| 2          | 1989-12-02 12:03:00 | null                | null                | 0 ␣
→  | 0     | 0     |
| 2          | 1989-12-04 13:14:00 | null                | null                | 0 ␣
→  | 0     | 0     |
| 2          | 1989-12-06 15:17:00 | 1989-12-04 13:14:00 | 1989-12-06 15:17:00 | 0 ␣
→  | 0     | 0     |
| 2          | 1989-12-08 16:22:00 | 1989-12-04 13:14:00 | 1989-12-08 16:22:00 | 1 ␣
→  | 1     | 1     |
| 2          | 1989-12-10 03:07:00 | 1989-12-04 13:14:00 | 1989-12-10 03:07:00 | 1 ␣
→  | 2     | 1     |
└────+───────+───────+───────+──+──+──+
>>> boolean_expr_bound_sum(
...     df,
...     pl.col("idx").is_in([1, 4, 7]),
...     "bound_to_row",
...     "left",
... ).drop("idx")
shape: (8, 7)
+────+───────+───────+───────+──+──+──+
| subject_id | timestamp           | timestamp_at_start  | timestamp_at_end    | is_
→A | is_B | is_C |
| ---        | ---                 | ---                 | ---                 | ---
→  | ---   | ---   |
| i64        | datetime[µs]        | datetime[µs]        | datetime[µs]        |␣
→i64   | i64   | i64   |
|────|───────|───────|───────|──|──|──|
| 1          | 1989-12-01 12:03:00 | null                | null                | 0 ␣
→  | 0     | 0     |
| 1          | 1989-12-03 13:14:00 | 1989-12-03 13:14:00 | 1989-12-03 13:14:00 | 0 ␣
→  | 0     | 0     |
| 1          | 1989-12-05 15:17:00 | 1989-12-03 13:14:00 | 1989-12-05 15:17:00 | 0 ␣
→  | 1     | 1     |
| 2          | 1989-12-02 12:03:00 | null                | null                | 0 ␣
→  | 0     | 0     |
| 2          | 1989-12-04 13:14:00 | 1989-12-04 13:14:00 | 1989-12-04 13:14:00 | 0 ␣
→  | 0     | 0     |
| 2          | 1989-12-06 15:17:00 | 1989-12-04 13:14:00 | 1989-12-06 15:17:00 | 1 ␣
→  | 0     | 0     |
| 2          | 1989-12-08 16:22:00 | 1989-12-04 13:14:00 | 1989-12-08 16:22:00 | 2 ␣
→  | 1     | 1     |
| 2          | 1989-12-10 03:07:00 | 1989-12-10 03:07:00 | 1989-12-10 03:07:00 | 0 ␣
→  | 0     | 0     |
└────+───────+───────+───────+──+──+──+
>>> boolean_expr_bound_sum(
...     df,
...     pl.col("idx").is_in([1, 4, 7]),
...     "bound_to_row",
...     "right",
... ).drop("idx")
shape: (8, 7)
+────+───────+───────+───────+──+──+──+
```

(continued from previous page)

```
| subject_id | timestamp           | timestamp_at_start  | timestamp_at_end    | is_
↪A | is_B | is_C |
| ---        | ---                 | ---                 | ---                 | ---
↪  | --- | --- |
| i64        | datetime[μs]        | datetime[μs]        | datetime[μs]        |␣
↪i64  | i64 | i64 |
|----|-------|-------|-------|--|--|--|
| 1          | 1989-12-01 12:03:00 | null                | null                | 0 ␣
↪  | 0     | 0    |
| 1          | 1989-12-03 13:14:00 | null                | null                | 0 ␣
↪  | 0     | 0    |
| 1          | 1989-12-05 15:17:00 | 1989-12-03 13:14:00 | 1989-12-05 15:17:00 | 1 ␣
↪  | 0     | 0    |
| 2          | 1989-12-02 12:03:00 | null                | null                | 0 ␣
↪  | 0     | 0    |
| 2          | 1989-12-04 13:14:00 | null                | null                | 0 ␣
↪  | 0     | 0    |
| 2          | 1989-12-06 15:17:00 | 1989-12-04 13:14:00 | 1989-12-06 15:17:00 | 1 ␣
↪  | 1     | 1    |
| 2          | 1989-12-08 16:22:00 | 1989-12-04 13:14:00 | 1989-12-08 16:22:00 | 1 ␣
↪  | 2     | 1    |
| 2          | 1989-12-10 03:07:00 | 1989-12-04 13:14:00 | 1989-12-10 03:07:00 | 1 ␣
↪  | 3     | 2    |
└────+───────+───────+───────+──+──+──+
>>> boolean_expr_bound_sum(
...     df,
...     pl.col("idx").is_in([1, 4, 7]),
...     "row_to_bound",
...     "both",
... ).drop("idx")
shape: (8, 7)
+----+-------+-------+-------+--+--+--+
| subject_id | timestamp           | timestamp_at_start  | timestamp_at_end    | is_
↪A | is_C |
| ---        | ---                 | ---                 | ---                 | ---
↪  | --- | --- |
| i64        | datetime[μs]        | datetime[μs]        | datetime[μs]        |␣
↪i64  | i64 | i64 |
|----|-------|-------|-------|--|--|--|
| 1          | 1989-12-01 12:03:00 | 1989-12-01 12:03:00 | 1989-12-03 13:14:00 | 1 ␣
↪  | 1     | 1    |
| 1          | 1989-12-03 13:14:00 | 1989-12-03 13:14:00 | 1989-12-03 13:14:00 | 0 ␣
↪  | 1     | 1    |
| 1          | 1989-12-05 15:17:00 | null                | null                | 0 ␣
↪  | 0     | 0    |
| 2          | 1989-12-02 12:03:00 | 1989-12-02 12:03:00 | 1989-12-04 13:14:00 | 2 ␣
↪  | 1     | 0    |
| 2          | 1989-12-04 13:14:00 | 1989-12-04 13:14:00 | 1989-12-04 13:14:00 | 1 ␣
↪  | 0     | 0    |
| 2          | 1989-12-06 15:17:00 | 1989-12-06 15:17:00 | 1989-12-10 03:07:00 | 1 ␣
↪  | 3     | 2    |
| 2          | 1989-12-08 16:22:00 | 1989-12-08 16:22:00 | 1989-12-10 03:07:00 | 0 ␣
```

(continues on next page)

```
→ | 2     | 1     |
| 2          | 1989-12-10 03:07:00 | 1989-12-10 03:07:00 | 1989-12-10 03:07:00 | 0 ␣
→ | 1     | 1     |
└────+───────+───────+───────+──+──+──+
>>> boolean_expr_bound_sum(
...     df,
...     pl.col("idx").is_in([1, 4, 7]),
...     "row_to_bound",
...     "none",
... ).drop("idx")
shape: (8, 7)
+────+───────+───────+───────+──+──+──+
| subject_id | timestamp           | timestamp_at_start  | timestamp_at_end    | is_
→A | is_B | is_C |
| ---        | ---                 | ---                 | ---                 | ---
→  | ---   | ---   |
| i64        | datetime[μs]        | datetime[μs]        | datetime[μs]        |␣
→i64   | i64   | i64   |
|────|───────|───────|───────|──|──|──|
| 1          | 1989-12-01 12:03:00 | 1989-12-01 12:03:00 | 1989-12-03 13:14:00 | 0 ␣
→  | 0     | 0     |
| 1          | 1989-12-03 13:14:00 | null                | null                | 0 ␣
→  | 0     | 0     |
| 1          | 1989-12-05 15:17:00 | null                | null                | 0 ␣
→  | 0     | 0     |
| 2          | 1989-12-02 12:03:00 | 1989-12-02 12:03:00 | 1989-12-04 13:14:00 | 0 ␣
→  | 0     | 0     |
| 2          | 1989-12-04 13:14:00 | 1989-12-04 13:14:00 | 1989-12-10 03:07:00 | 1 ␣
→  | 2     | 1     |
| 2          | 1989-12-06 15:17:00 | 1989-12-06 15:17:00 | 1989-12-10 03:07:00 | 0 ␣
→  | 1     | 0     |
| 2          | 1989-12-08 16:22:00 | 1989-12-08 16:22:00 | 1989-12-10 03:07:00 | 0 ␣
→  | 0     | 0     |
| 2          | 1989-12-10 03:07:00 | null                | null                | 0 ␣
→  | 0     | 0     |
└────+───────+───────+───────+──+──+──+
>>> boolean_expr_bound_sum(
...     df,
...     pl.col("idx").is_in([1, 4, 7]),
...     "row_to_bound",
...     "left",
... ).drop("idx")
shape: (8, 7)
+────+───────+───────+───────+──+──+──+
| subject_id | timestamp           | timestamp_at_start  | timestamp_at_end    | is_
→A | is_B | is_C |
| ---        | ---                 | ---                 | ---                 | ---
→  | ---   | ---   |
| i64        | datetime[μs]        | datetime[μs]        | datetime[μs]        |␣
→i64   | i64   | i64   |
|────|───────|───────|───────|──|──|──|
| 1          | 1989-12-01 12:03:00 | 1989-12-01 12:03:00 | 1989-12-03 13:14:00 | 1 ␣
```

```
→  | 0     | 0     |
| 1         | 1989-12-03 13:14:00 | null                | null                | 0 ␣
→  | 0     | 0     |
| 1         | 1989-12-05 15:17:00 | null                | null                | 0 ␣
→  | 0     | 0     |
| 2         | 1989-12-02 12:03:00 | 1989-12-02 12:03:00 | 1989-12-04 13:14:00 | 1 ␣
→  | 1     | 0     |
| 2         | 1989-12-04 13:14:00 | 1989-12-04 13:14:00 | 1989-12-10 03:07:00 | 2 ␣
→  | 2     | 1     |
| 2         | 1989-12-06 15:17:00 | 1989-12-06 15:17:00 | 1989-12-10 03:07:00 | 1 ␣
→  | 2     | 1     |
| 2         | 1989-12-08 16:22:00 | 1989-12-08 16:22:00 | 1989-12-10 03:07:00 | 0 ␣
→  | 1     | 0     |
| 2         | 1989-12-10 03:07:00 | null                | null                | 0 ␣
→  | 0     | 0     |
└────+───────+───────+───────+──+──+──+
>>> boolean_expr_bound_sum(
...     df,
...     pl.col("idx").is_in([1, 4, 7]),
...     "row_to_bound",
...     "right",
... ).drop("idx")
shape: (8, 7)
+────+───────+───────+───────+──+──+──+
| subject_id | timestamp           | timestamp_at_start  | timestamp_at_end    | is_
→A | is_B | is_C |
| ---        | ---                 | ---                 | ---                 | ---
→  | --- | --- |
| i64        | datetime[μs]        | datetime[μs]        | datetime[μs]        |␣
→i64   | i64 | i64 |
|────|───────|───────|───────|──|──|──|
| 1         | 1989-12-01 12:03:00 | 1989-12-01 12:03:00 | 1989-12-03 13:14:00 | 0 ␣
→  | 1     | 1     |
| 1         | 1989-12-03 13:14:00 | 1989-12-03 13:14:00 | 1989-12-03 13:14:00 | 0 ␣
→  | 0     | 0     |
| 1         | 1989-12-05 15:17:00 | null                | null                | 0 ␣
→  | 0     | 0     |
| 2         | 1989-12-02 12:03:00 | 1989-12-02 12:03:00 | 1989-12-04 13:14:00 | 1 ␣
→  | 0     | 0     |
| 2         | 1989-12-04 13:14:00 | 1989-12-04 13:14:00 | 1989-12-04 13:14:00 | 0 ␣
→  | 0     | 0     |
| 2         | 1989-12-06 15:17:00 | 1989-12-06 15:17:00 | 1989-12-10 03:07:00 | 0 ␣
→  | 2     | 1     |
| 2         | 1989-12-08 16:22:00 | 1989-12-08 16:22:00 | 1989-12-10 03:07:00 | 0 ␣
→  | 1     | 1     |
| 2         | 1989-12-10 03:07:00 | 1989-12-10 03:07:00 | 1989-12-10 03:07:00 | 0 ␣
→  | 0     | 0     |
└────+───────+───────+───────+──+──+──+
>>> #### WITH OFFSET ####
>>> boolean_expr_bound_sum(
...     df,
...     pl.col("idx").is_in([1, 4, 7]),
```

```
...     "bound_to_row",
...     "both",
...     offset = timedelta(days=3),
... ).drop("idx")
shape: (8, 7)
+----+-------+-------+-------+--+--+--+
| subject_id | timestamp           | timestamp_at_start  | timestamp_at_end    | is_
↪A | is_B | is_C |
| ---        | ---                 | ---                 | ---                 | ---
↪ | --- | --- |
| i64        | datetime[μs]        | datetime[μs]        | datetime[μs]        |␣
↪i64 | i64 | i64 |
|----|-------|-------|-------|--|--|--|
| 1          | 1989-12-01 12:03:00 | 1989-12-03 13:14:00 | 1989-12-04 12:03:00 | 0 ␣
↪ | 1   | 1     |
| 1          | 1989-12-03 13:14:00 | 1989-12-03 13:14:00 | 1989-12-06 13:14:00 | 1 ␣
↪ | 1   | 1     |
| 1          | 1989-12-05 15:17:00 | 1989-12-03 13:14:00 | 1989-12-08 15:17:00 | 1 ␣
↪ | 1   | 1     |
| 2          | 1989-12-02 12:03:00 | 1989-12-04 13:14:00 | 1989-12-05 12:03:00 | 1 ␣
↪ | 0   | 0     |
| 2          | 1989-12-04 13:14:00 | 1989-12-04 13:14:00 | 1989-12-07 13:14:00 | 2 ␣
↪ | 1   | 1     |
| 2          | 1989-12-06 15:17:00 | 1989-12-04 13:14:00 | 1989-12-09 15:17:00 | 2 ␣
↪ | 2   | 1     |
| 2          | 1989-12-08 16:22:00 | 1989-12-10 03:07:00 | 1989-12-11 16:22:00 | 0 ␣
↪ | 1   | 1     |
| 2          | 1989-12-10 03:07:00 | 1989-12-10 03:07:00 | 1989-12-13 03:07:00 | 0 ␣
↪ | 1   | 1     |
└----+-------+-------+-------+--+--+--+
>>> boolean_expr_bound_sum(
...     df,
...     pl.col("idx").is_in([1, 4, 7]),
...     "bound_to_row",
...     "left",
...     offset = timedelta(days=3),
... ).drop("idx")
shape: (8, 7)
+----+-------+-------+-------+--+--+--+
| subject_id | timestamp           | timestamp_at_start  | timestamp_at_end    | is_
↪A | is_B | is_C |
| ---        | ---                 | ---                 | ---                 | ---
↪ | --- | --- |
| i64        | datetime[μs]        | datetime[μs]        | datetime[μs]        |␣
↪i64 | i64 | i64 |
|----|-------|-------|-------|--|--|--|
| 1          | 1989-12-01 12:03:00 | 1989-12-03 13:14:00 | 1989-12-04 12:03:00 | 0 ␣
↪ | 1   | 1     |
| 1          | 1989-12-03 13:14:00 | 1989-12-03 13:14:00 | 1989-12-06 13:14:00 | 1 ␣
↪ | 1   | 1     |
| 1          | 1989-12-05 15:17:00 | 1989-12-03 13:14:00 | 1989-12-08 15:17:00 | 1 ␣
↪ | 1   | 1     |
```

```
| 2           | 1989-12-02 12:03:00 | 1989-12-04 13:14:00 | 1989-12-05 12:03:00 | 1 ␣
↳  | 0      | 0      |
| 2           | 1989-12-04 13:14:00 | 1989-12-04 13:14:00 | 1989-12-07 13:14:00 | 2 ␣
↳  | 1      | 1      |
| 2           | 1989-12-06 15:17:00 | 1989-12-04 13:14:00 | 1989-12-09 15:17:00 | 2 ␣
↳  | 2      | 1      |
| 2           | 1989-12-08 16:22:00 | 1989-12-10 03:07:00 | 1989-12-11 16:22:00 | 0 ␣
↳  | 1      | 1      |
| 2           | 1989-12-10 03:07:00 | 1989-12-10 03:07:00 | 1989-12-13 03:07:00 | 0 ␣
↳  | 1      | 1      |
└────+───────+───────+───────+──+──+──+
>>> boolean_expr_bound_sum(
...     df,
...     pl.col("idx").is_in([1, 4, 7]),
...     "bound_to_row",
...     "none",
...     timedelta(days=-3),
... ).drop("idx")
shape: (8, 7)
+────+───────+───────+───────+──+──+──+
| subject_id | timestamp           | timestamp_at_start  | timestamp_at_end    | is_
↳A | is_B | is_C |
| ---        | ---                 | ---                 | ---                 | ---
↳  | ---  | ---  |
| i64        | datetime[μs]        | datetime[μs]        | datetime[μs]        |␣
↳i64  | i64  | i64  |
|────|───────|───────|───────|──|──|──|
| 1           | 1989-12-01 12:03:00 | null                | null                | 0 ␣
↳  | 0      | 0      |
| 1           | 1989-12-03 13:14:00 | null                | null                | 0 ␣
↳  | 0      | 0      |
| 1           | 1989-12-05 15:17:00 | null                | null                | 0 ␣
↳  | 0      | 0      |
| 2           | 1989-12-02 12:03:00 | null                | null                | 0 ␣
↳  | 0      | 0      |
| 2           | 1989-12-04 13:14:00 | null                | null                | 0 ␣
↳  | 0      | 0      |
| 2           | 1989-12-06 15:17:00 | null                | null                | 0 ␣
↳  | 0      | 0      |
| 2           | 1989-12-08 16:22:00 | 1989-12-04 13:14:00 | 1989-12-05 16:22:00 | 0 ␣
↳  | 0      | 0      |
| 2           | 1989-12-10 03:07:00 | 1989-12-04 13:14:00 | 1989-12-07 03:07:00 | 1 ␣
↳  | 1      | 1      |
└────+───────+───────+───────+──+──+──+
>>> boolean_expr_bound_sum(
...     df,
...     pl.col("idx").is_in([1, 4, 7]),
...     "bound_to_row",
...     "right",
...     offset = timedelta(days=-3),
... ).drop("idx")
shape: (8, 7)
```

(continued from previous page)

```
+----+-------+-------+-------+--+--+--+
| subject_id | timestamp           | timestamp_at_start | timestamp_at_end   | is_
↪A | is_B | is_C |
| ---        | ---                 | ---                | ---                | ---
↪   | --- | ---  |
| i64        | datetime[μs]        | datetime[μs]       | datetime[μs]       |␣
↪i64  | i64 | i64  |
|----|-------|-------|-------|--|--|--|
| 1          | 1989-12-01 12:03:00 | null               | null               | 0 ␣
↪   | 0    | 0    |
| 1          | 1989-12-03 13:14:00 | null               | null               | 0 ␣
↪   | 0    | 0    |
| 1          | 1989-12-05 15:17:00 | null               | null               | 0 ␣
↪   | 0    | 0    |
| 2          | 1989-12-02 12:03:00 | null               | null               | 0 ␣
↪   | 0    | 0    |
| 2          | 1989-12-04 13:14:00 | null               | null               | 0 ␣
↪   | 0    | 0    |
| 2          | 1989-12-06 15:17:00 | null               | null               | 0 ␣
↪   | 0    | 0    |
| 2          | 1989-12-08 16:22:00 | 1989-12-04 13:14:00 | 1989-12-05 16:22:00 | 0 ␣
↪   | 0    | 0    |
| 2          | 1989-12-10 03:07:00 | 1989-12-04 13:14:00 | 1989-12-07 03:07:00 | 1 ␣
↪   | 1    | 1    |
└----+-------+-------+-------+--+--+--+
>>> boolean_expr_bound_sum(
...     df,
...     pl.col("idx").is_in([1, 4, 7]),
...     "row_to_bound",
...     "both",
...     offset = timedelta(days=3),
... ).drop("idx")
shape: (8, 7)
+----+-------+-------+-------+--+--+--+
| subject_id | timestamp           | timestamp_at_start | timestamp_at_end   | is_
↪A | is_B | is_C |
| ---        | ---                 | ---                | ---                | ---
↪   | --- | ---  |
| i64        | datetime[μs]        | datetime[μs]       | datetime[μs]       |␣
↪i64  | i64 | i64  |
|----|-------|-------|-------|--|--|--|
| 1          | 1989-12-01 12:03:00 | null               | null               | 0 ␣
↪   | 0    | 0    |
| 1          | 1989-12-03 13:14:00 | null               | null               | 0 ␣
↪   | 0    | 0    |
| 1          | 1989-12-05 15:17:00 | null               | null               | 0 ␣
↪   | 0    | 0    |
| 2          | 1989-12-02 12:03:00 | 1989-12-05 12:03:00 | 1989-12-10 03:07:00 | 1 ␣
↪   | 3    | 2    |
| 2          | 1989-12-04 13:14:00 | 1989-12-07 13:14:00 | 1989-12-10 03:07:00 | 0 ␣
↪   | 2    | 1    |
| 2          | 1989-12-06 15:17:00 | 1989-12-09 15:17:00 | 1989-12-10 03:07:00 | 0 ␣
```

(continues on next page)

```
→  | 1      | 1      |
| 2          | 1989-12-08 16:22:00 | null                | null                | 0 ␣
→  | 0      | 0      |
| 2          | 1989-12-10 03:07:00 | null                | null                | 0 ␣
→  | 0      | 0      |
└────+───────+───────+───────+──+──+──+
>>> boolean_expr_bound_sum(
...     df,
...     pl.col("idx").is_in([1, 4, 7]),
...     "row_to_bound",
...     "left",
...     offset = timedelta(days=3),
... ).drop("idx")
shape: (8, 7)
+────+───────+───────+───────+──+──+──+
| subject_id | timestamp           | timestamp_at_start  | timestamp_at_end    | is_
→A | is_B | is_C |
| ---        | ---                 | ---                 | ---                 | ---
→  | ---    | ---    |
| i64        | datetime[μs]        | datetime[μs]        | datetime[μs]        |␣
→i64   | i64    | i64    |
|────|───────|───────|───────|──|──|──|
| 1          | 1989-12-01 12:03:00 | null                | null                | 0 ␣
→  | 0      | 0      |
| 1          | 1989-12-03 13:14:00 | null                | null                | 0 ␣
→  | 0      | 0      |
| 1          | 1989-12-05 15:17:00 | null                | null                | 0 ␣
→  | 0      | 0      |
| 2          | 1989-12-02 12:03:00 | 1989-12-05 12:03:00 | 1989-12-10 03:07:00 | 1 ␣
→  | 2      | 1      |
| 2          | 1989-12-04 13:14:00 | 1989-12-07 13:14:00 | 1989-12-10 03:07:00 | 0 ␣
→  | 1      | 0      |
| 2          | 1989-12-06 15:17:00 | 1989-12-09 15:17:00 | 1989-12-10 03:07:00 | 0 ␣
→  | 0      | 0      |
| 2          | 1989-12-08 16:22:00 | null                | null                | 0 ␣
→  | 0      | 0      |
| 2          | 1989-12-10 03:07:00 | null                | null                | 0 ␣
→  | 0      | 0      |
└────+───────+───────+───────+──+──+──+
>>> boolean_expr_bound_sum(
...     df,
...     pl.col("idx").is_in([1, 4, 7]),
...     "row_to_bound",
...     "none",
...     offset = timedelta(days=-3),
... ).drop("idx")
shape: (8, 7)
+────+───────+───────+───────+──+──+──+
| subject_id | timestamp           | timestamp_at_start  | timestamp_at_end    | is_
→A | is_B | is_C |
| ---        | ---                 | ---                 | ---                 | ---
→  | ---    | ---    |
```

```
| i64        | datetime[µs]        | datetime[µs]        | datetime[µs]         |␣
↪i64  | i64  | i64  |
|----|-------|-------|-------|--|--|--|
| 1          | 1989-12-01 12:03:00 | 1989-11-28 12:03:00 | 1989-12-03 13:14:00 | 1 ␣
↪    | 0     | 0     |
| 1          | 1989-12-03 13:14:00 | 1989-11-30 13:14:00 | 1989-12-03 13:14:00 | 1 ␣
↪    | 0     | 0     |
| 1          | 1989-12-05 15:17:00 | 1989-12-02 15:17:00 | 1989-12-03 13:14:00 | 0 ␣
↪    | 0     | 0     |
| 2          | 1989-12-02 12:03:00 | 1989-11-29 12:03:00 | 1989-12-04 13:14:00 | 1 ␣
↪    | 1     | 0     |
| 2          | 1989-12-04 13:14:00 | 1989-12-01 13:14:00 | 1989-12-04 13:14:00 | 1 ␣
↪    | 1     | 0     |
| 2          | 1989-12-06 15:17:00 | 1989-12-03 15:17:00 | 1989-12-04 13:14:00 | 0 ␣
↪    | 0     | 0     |
| 2          | 1989-12-08 16:22:00 | 1989-12-05 16:22:00 | 1989-12-10 03:07:00 | 1 ␣
↪    | 2     | 1     |
| 2          | 1989-12-10 03:07:00 | 1989-12-07 03:07:00 | 1989-12-10 03:07:00 | 0 ␣
↪    | 1     | 0     |
└----+-------+-------+-------+--+--+--+
>>> boolean_expr_bound_sum(
...     df,
...     pl.col("idx").is_in([1, 4, 7]),
...     "row_to_bound",
...     "right",
...     offset = timedelta(days=-3),
... ).drop("idx")
shape: (8, 7)
+----+-------+-------+-------+--+--+--+
| subject_id | timestamp           | timestamp_at_start  | timestamp_at_end    | is_␣
↪A | is_B | is_C |
| ---        | ---                 | ---                 | ---                 | ---␣
↪   | ---  | ---  |
| i64        | datetime[µs]        | datetime[µs]        | datetime[µs]         |␣
↪i64  | i64  | i64  |
|----|-------|-------|-------|--|--|--|
| 1          | 1989-12-01 12:03:00 | 1989-11-28 12:03:00 | 1989-12-03 13:14:00 | 1 ␣
↪    | 1     | 1     |
| 1          | 1989-12-03 13:14:00 | 1989-11-30 13:14:00 | 1989-12-03 13:14:00 | 1 ␣
↪    | 1     | 1     |
| 1          | 1989-12-05 15:17:00 | 1989-12-02 15:17:00 | 1989-12-03 13:14:00 | 0 ␣
↪    | 1     | 1     |
| 2          | 1989-12-02 12:03:00 | 1989-11-29 12:03:00 | 1989-12-04 13:14:00 | 2 ␣
↪    | 1     | 0     |
| 2          | 1989-12-04 13:14:00 | 1989-12-01 13:14:00 | 1989-12-04 13:14:00 | 2 ␣
↪    | 1     | 0     |
| 2          | 1989-12-06 15:17:00 | 1989-12-03 15:17:00 | 1989-12-04 13:14:00 | 1 ␣
↪    | 0     | 0     |
| 2          | 1989-12-08 16:22:00 | 1989-12-05 16:22:00 | 1989-12-10 03:07:00 | 1 ␣
↪    | 3     | 2     |
| 2          | 1989-12-10 03:07:00 | 1989-12-07 03:07:00 | 1989-12-10 03:07:00 | 0 ␣
↪    | 2     | 1     |
```

```
└────+───────+───────+───────+──+──+──+
```

```
>>> boolean_expr_bound_sum(df, pl.col("idx").is_in([1, 4, 7]), "invalid_mode",
→"right",
...     offset = timedelta(days=-3))
Traceback (most recent call last):
    ...
ValueError: Mode 'invalid_mode' invalid!
>>> boolean_expr_bound_sum(df, pl.col("idx").is_in([1, 4, 7]), "row_to_bound",
→"invalid_closed",
...     offset = timedelta(days=-3))
Traceback (most recent call last):
    ...
ValueError: Closed 'invalid_closed' invalid!
```

```
>>> boolean_expr_bound_sum(df, pl.col("idx").is_in([1, 4, 7]), mode="row_to_bound",
...         closed="right", offset=timedelta(days=1)).columns
['subject_id', 'timestamp', 'timestamp_at_start', 'timestamp_at_end', 'idx', 'is_A', 'is_B',
→'is_C']
>>> boolean_expr_bound_sum(df, pl.col("idx").is_in([1, 4, 7]), mode="row_to_bound",
...         closed="left", offset=timedelta(days=-1)).columns
['subject_id', 'timestamp', 'timestamp_at_start', 'timestamp_at_end', 'idx', 'is_A', 'is_B',
→'is_C']
```

### aces.config module

This module contains functions for loading and parsing the configuration file and subsequently building a tree structure from the configuration.

**class** aces.config.**DerivedPredicateConfig**(expr: *str*, static: *bool* = *False*)

> Bases: object
>
> A configuration object for derived predicates, which are composed of multiple input predicates.
>
> > **Parameters**
> >
> > > **expr: str**
> > > The expression defining the derived predicate in terms of other predicates.
> >
> > **Raises**
> > > **ValueError** – If the expression is empty, does not start with 'and(' or 'or(', or does not contain at least two input predicates.

### Examples

```
>>> pred = DerivedPredicateConfig("and(P1, P2, P3)")
>>> pred = DerivedPredicateConfig("and()")
Traceback (most recent call last):
    ...
ValueError: Derived predicate expression must have at least two input predicates␣
↪(comma separated).
Got: 'and()'
>>> pred = DerivedPredicateConfig("or(PA, PB)")
>>> pred = DerivedPredicateConfig("PA + PB")
Traceback (most recent call last):
    ...
ValueError: Derived predicate expression must start with 'and(' or 'or('. Got: 'PA␣
↪+ PB'
>>> pred = DerivedPredicateConfig("")
Traceback (most recent call last):
    ...
ValueError: Derived predicates must have a non-empty expression field.
```

**eval_expr**() → Expr

> Returns a Polars expression that evaluates this predicate against necessary dependent predicates.

> Note: The output syntax for the following examples is dependent on the polars version used. The expected outputs have been validated on polars version 0.20.30.

### Examples

```
>>> expr = DerivedPredicateConfig("and(P1, P2, P3)").eval_expr()
>>> print(expr)
[(col("P1")) > (dyn int: 0)].all_horizontal([[(col("P2")) >
    (dyn int: 0)], [(col("P3")) > (dyn int: 0)]])
>>> expr = DerivedPredicateConfig("or(PA, PB)").eval_expr()
>>> print(expr)
[(col("PA")) > (dyn int: 0)].any_horizontal([[(col("PB")) > (dyn int: 0)]])
```

**expr** : str

**property is_plain** : bool

**static** : bool = **False**

**class** aces.config.**EventConfig**(predicate*: str*)

> Bases: object

A configuration object for defining the event that triggers the task extraction process.

This is defined by all events that match a simple predicate. This event serves as the root of the window tree, and its form is dictated by the fact that we must be able to localize the tree to identify valid realizations of the tree.

### Examples

```
>>> event = EventConfig("event_type//ADMISSION")
>>> event.predicate
'event_type//ADMISSION'
```

**predicate** : str

**class** aces.config.**PlainPredicateConfig**(code: 'str | dict[str, Any]', value_min: 'float | None' = None, value_max: 'float | None' = None, value_min_inclusive: 'bool | None' = None, value_max_inclusive: 'bool | None' = None, static: 'bool' = False, other_cols: 'dict[str, str]' = <factory>)

Bases: object

**ESGPT_eval_expr**(values_column: *str | None = **None**) → Expr

Returns a Polars expression that evaluates this predicate for a ESGPT formatted dataset.

Note: The output syntax for the following examples is dependent on the polars version used. The expected outputs have been validated on polars version 0.20.30.

### Examples

```
>>> # Should handle univariate regression values
>>> expr = PlainPredicateConfig("HR", value_min=120, value_min_inclusive=False)
>>> expr = expr.ESGPT_eval_expr("HR")
>>> print(expr)
[(col("HR")) > (dyn int: 120)]
>>> expr = PlainPredicateConfig("BP//systolic", 120, 140, True, False).ESGPT_
→eval_expr("BP_value")
>>> print(expr)
[(col("BP")) == (String(systolic))].all_horizontal([[(col("BP_value")) >=
    (dyn int: 120)], [(col("BP_value")) < (dyn int: 140)]])
>>> cfg = PlainPredicateConfig("BP//systolic", value_min=120, value_min_
→inclusive=False)
>>> expr = cfg.ESGPT_eval_expr("blood_pressure_value")
>>> print(expr)
[(col("BP")) == (String(systolic))].all_horizontal([[(col("blood_pressure_value
→")) >
    (dyn int: 120)]])
>>> cfg = PlainPredicateConfig("BP//systolic", value_max=140, value_max_
→inclusive=True)
>>> expr = cfg.ESGPT_eval_expr("blood_pressure_value")
>>> print(expr)
[(col("BP")) == (String(systolic))].all_horizontal([[(col("blood_pressure_value
→")) <=
    (dyn int: 140)]])
>>> expr = PlainPredicateConfig("BP//diastolic").ESGPT_eval_expr()
>>> print(expr)
[(col("BP")) == (String(diastolic))]
>>> expr = PlainPredicateConfig("event_type//ADMISSION").ESGPT_eval_expr()
>>> print(expr)
col("event_type").strict_cast(String).str.split([String(&)]).list.
→contains([String(ADMISSION)])
```

(continues on next page)

```
>>> expr = PlainPredicateConfig("BP//diastolic//atrial").ESGPT_eval_expr()
>>> print(expr)
[(col("BP")) == (String(diastolic//atrial))]
>>> expr = PlainPredicateConfig("BP//diastolic", None, None).ESGPT_eval_expr()
>>> print(expr)
[(col("BP")) == (String(diastolic))]
>>> expr = PlainPredicateConfig("BP").ESGPT_eval_expr()
>>> print(expr)
col("BP").is_not_null()
>>> expr = PlainPredicateConfig("BP//systole", other_cols={"chamber": "atrial"}
↪).ESGPT_eval_expr()
>>> print(expr)
[(col("BP")) == (String(systole))].all_horizontal([[(col("chamber")) ==␣
↪(String(atrial))]])
```

```
>>> expr = PlainPredicateConfig("BP//systolic", value_min=120).ESGPT_eval_expr()
Traceback (most recent call last):
    ...
ValueError: Must specify a values column for ESGPT predicates with a value_min␣
↪= 120
>>> expr = PlainPredicateConfig("BP//systolic", value_max=140).ESGPT_eval_expr()
Traceback (most recent call last):
    ...
ValueError: Must specify a values column for ESGPT predicates with a value_max␣
↪= 140
```

**MEDS_eval_expr**() → Expr

Returns a Polars expression that evaluates this predicate for a MEDS formatted dataset.

Note: The output syntax for the following examples is dependent on the polars version used. The expected outputs have been validated on polars version 0.20.30.

### Examples

```
>>> expr = PlainPredicateConfig("BP//systolic", 120, 140, True, False).MEDS_
↪eval_expr()
>>> print(expr)
[(col("code")) == (String(BP//systolic))].all_horizontal([[(col("numeric_value
↪")) >=
   (dyn int: 120)], [(col("numeric_value")) < (dyn int: 140)]])
>>> cfg = PlainPredicateConfig("BP//systolic", value_min=120, value_min_
↪inclusive=False)
>>> expr = cfg.MEDS_eval_expr()
>>> print(expr)
[(col("code")) == (String(BP//systolic))].all_horizontal([[(col("numeric_value
↪")) >
   (dyn int: 120)]])
>>> cfg = PlainPredicateConfig("BP//systolic", value_max=140, value_max_
↪inclusive=True)
>>> expr = cfg.MEDS_eval_expr()
>>> print(expr)
```

```
[(col("code")) == (String(BP//systolic))].all_horizontal([[(col("numeric_value
→")) <=
   (dyn int: 140)]])
>>> cfg = PlainPredicateConfig("BP//diastolic")
>>> expr = cfg.MEDS_eval_expr()
>>> print(expr)
[(col("code")) == (String(BP//diastolic))]
>>> cfg = PlainPredicateConfig("BP//diastolic", other_cols={"chamber": "atrial"}
→)
>>> expr = cfg.MEDS_eval_expr()
>>> print(expr)
[(col("code")) == (String(BP//diastolic))].all_horizontal([[(col("chamber")) ==
   (String(atrial))]])
```

```
>>> cfg = PlainPredicateConfig(code={'regex': None, 'any': None})
>>> expr = cfg.MEDS_eval_expr()
Traceback (most recent call last):
    ...
ValueError: Only one of 'regex' or 'any' can be specified in the code field!
Got: ['regex', 'any'].
>>> cfg = PlainPredicateConfig(code={'foo': None})
>>> expr = cfg.MEDS_eval_expr()
Traceback (most recent call last):
    ...
ValueError: Invalid specification in the code field! Got: {'foo': None}.
Expected one of 'regex', 'any'.
>>> cfg = PlainPredicateConfig(code={'regex': ''})
>>> expr = cfg.MEDS_eval_expr()
Traceback (most recent call last):
    ...
ValueError: Invalid specification in the code field! Got: {'regex': ''}.
Expected a non-empty string for 'regex'.
>>> cfg = PlainPredicateConfig(code={'any': []})
>>> expr = cfg.MEDS_eval_expr()
Traceback (most recent call last):
    ...
ValueError: Invalid specification in the code field! Got: {'any': []}.
Expected a list of strings for 'any'.
```

```
>>> cfg = PlainPredicateConfig(code={'regex': '^foo.*'})
>>> expr = cfg.MEDS_eval_expr()
>>> print(expr)
col("code").str.contains([String(^foo.*)])
>>> cfg = PlainPredicateConfig(code={'regex': '^foo.*'}, value_min=120)
>>> expr = cfg.MEDS_eval_expr()
>>> print(expr)
col("code").str.contains([String(^foo.*)]).all_horizontal([[(col("numeric_value
→")) >
(dyn int: 120)]])
>>> cfg = PlainPredicateConfig(code={'any': ['foo', 'bar']})
>>> expr = cfg.MEDS_eval_expr()
```

```
>>> print(expr)
col("code").is_in([Series])
```

**code** : str | dict[str, Any]

**property is_plain** : bool

**other_cols** : dict[str, str]

**static** : bool = **False**

**value_max** : float | None = **None**

**value_max_inclusive** : bool | None = **None**

**value_min** : float | None = **None**

**value_min_inclusive** : bool | None = **None**

**class** aces.config.**TaskExtractorConfig**(predicates*: dict[str,* aces.config.PlainPredicateConfig |
aces.config.DerivedPredicateConfig*]*, trigger*:* EventConfig,
windows*: dict[str,* aces.config.WindowConfig*] | None*,
label_window*: str | None* = **None**, index_timestamp_window*: str |*
*None* = **None**)

Bases: object

A configuration object for parsing the plain-data stored in a task extractor config.

This class can be serialized to and deserialized from a YAML file, and is largely a collection of utilities to parse, validate, and leverage task extraction configuration data in practice. There is no state stored in this class that is not present or recoverable from the source YAML file on disk. It also can be read from a simplified, "user-friendly" language, which can also be stored on or read from disk, which is ultimately parsed into the expansive, full specification contained in the YAML file referenced above.

### Parameters

**predicates: dict[str,** *aces.config.PlainPredicateConfig* **|** *aces.config.DerivedPredicateConfig***]**
A dictionary of predicate configurations, stored as either plain or derived predicate configuration objects (which are simple dataclasses with utility functions over plain dictionaries).

**trigger:** *EventConfig*
The event configuration that triggers the task extraction process. This is a simple dataclass with a single field, the name of the predicate that triggers the task extraction and serves as the root of the window tree.

**windows: dict[str,** *aces.config.WindowConfig***] | None**
A dictionary of window configurations. Each window configuration is a simple dataclass with that can be materialized to/from a simple, POD dictionary.

### Raises
**ValueError** – If any window or predicate names are not composed of alphanumeric or "_" characters.

**Examples**

```
>>> from bigtree import print_tree
>>> predicates = {
...     "admission": PlainPredicateConfig("admission"),
...     "discharge": PlainPredicateConfig("discharge"),
...     "death": PlainPredicateConfig("death"),
...     "death_or_discharge": DerivedPredicateConfig("or(death, discharge)"),
...     "diabetes_icd9": PlainPredicateConfig("ICD9CM//250.02"),
...     "diabetes_icd10": PlainPredicateConfig("ICD10CM//E11.65"),
...     "diabetes": DerivedPredicateConfig("or(diabetes_icd9, diabetes_icd10)"),
...     "diabetes_and_discharge": DerivedPredicateConfig("and(diabetes, discharge)
↪"),
... }
>>> trigger = EventConfig("admission")
>>> windows = {
...     "input": WindowConfig(
...         start=None,
...         end="trigger + 24h",
...         start_inclusive=True,
...         end_inclusive=True,
...         has={"_ANY_EVENT": "(32, None)"},
...         index_timestamp="end",
...     ),
...     "gap": WindowConfig(
...         start="input.end",
...         end="start + 24h",
...         start_inclusive=False,
...         end_inclusive=True,
...         has={"death_or_discharge": "(None, 0)", "admission": "(None, 0)"},
...     ),
...     "target": WindowConfig(
...         start="gap.end",
...         end="start -> death_or_discharge",
...         start_inclusive=False,
...         end_inclusive=True,
...         has={},
...         label="death",
...     ),
... }
>>> config = TaskExtractorConfig(predicates=predicates, trigger=trigger,
↪windows=windows)
>>> print(config.plain_predicates)
{'admission': PlainPredicateConfig(code='admission',
                    value_min=None,
                    value_max=None,
                    value_min_inclusive=None,
                    value_max_inclusive=None,
                    static=False,
                    other_cols={}),
 'discharge': PlainPredicateConfig(code='discharge',
                    value_min=None,
                    value_max=None,
```

(continues on next page)

```
                            value_min_inclusive=None,
                            value_max_inclusive=None,
                            static=False,
                            other_cols={}),
  'death': PlainPredicateConfig(code='death',
                            value_min=None,
                            value_max=None,
                            value_min_inclusive=None,
                            value_max_inclusive=None,
                            static=False,
                            other_cols={}),
  'diabetes_icd9': PlainPredicateConfig(code='ICD9CM//250.02',
                            value_min=None,
                            value_max=None,
                            value_min_inclusive=None,
                            value_max_inclusive=None,
                            static=False,
                            other_cols={}),
  'diabetes_icd10': PlainPredicateConfig(code='ICD10CM//E11.65',
                            value_min=None,
                            value_max=None,
                            value_min_inclusive=None,
                            value_max_inclusive=None,
                            static=False,
                            other_cols={})}
>>> print(config.label_window)
target
>>> print(config.index_timestamp_window)
input
>>> print(config.derived_predicates)
{'death_or_discharge': DerivedPredicateConfig(expr='or(death, discharge)',␣
↪static=False),
 'diabetes': DerivedPredicateConfig(expr='or(diabetes_icd9, diabetes_icd10)',␣
↪static=False),
 'diabetes_and_discharge': DerivedPredicateConfig(expr='and(diabetes, discharge)',␣
↪static=False)}
>>> print(nx.write_network_text(config.predicates_DAG))
+- death
|   └- death_or_discharge - discharge
+- discharge
|   ├- diabetes_and_discharge - diabetes
|   └-  ...
+- diabetes_icd9
|   └- diabetes - diabetes_icd10
|         └-  ...
+- diabetes_icd10
    └-  ...
>>> print_tree(config.window_tree)
trigger
└- input.end
    ├- input.start
    └- gap.end
```

```
       └─ target.end
```

**Configs will error out in various ways when passed inappropriate arguments:**

```
>>> config_path = "/foo/non_existent_file.yaml"
>>> cfg = TaskExtractorConfig.load(config_path)
Traceback (most recent call last):
    ...
FileNotFoundError: Cannot load missing configuration file /foo/non_existent_
↪file.yaml!
>>> import tempfile
>>> with tempfile.NamedTemporaryFile(mode="w", suffix=".txt") as f:
...     config_path = Path(f.name)
...     cfg = TaskExtractorConfig.load(config_path)
Traceback (most recent call last):
    ...
ValueError: Only supports reading from '.yaml'. Got: '.txt' in ....txt'.
>>> predicates_path = "/foo/non_existent_predicates.yaml"
>>> with tempfile.NamedTemporaryFile(mode="w", suffix=".yaml") as f:
...     config_path = Path(f.name)
...     cfg = TaskExtractorConfig.load(config_path, predicates_path)
Traceback (most recent call last):
    ...
FileNotFoundError: Cannot load missing predicates file /foo/non_existent_
↪predicates.yaml!
>>> with tempfile.NamedTemporaryFile(mode="w", suffix=".txt") as f:
...     predicates_path = Path(f.name)
...     with tempfile.NamedTemporaryFile(mode="w", suffix=".yaml") as f2:
...         config_path = Path(f2.name)
...         cfg = TaskExtractorConfig.load(config_path, predicates_path)
Traceback (most recent call last):
    ...
ValueError: Only supports reading from '.yaml'. Got: '.txt' in ....txt'.
>>> import yaml
>>> data = {
...     'predicates': {},
...     'trigger': {},
...     'foo': {}
... }
>>> with tempfile.NamedTemporaryFile(mode="w", suffix=".yaml") as f:
...     config_path = Path(f.name)
...     yaml.dump(data, f)
...     cfg = TaskExtractorConfig.load(config_path)
Traceback (most recent call last):
    ...
ValueError: Unrecognized keys in configuration file: 'foo'
>>> with tempfile.NamedTemporaryFile(mode="w", suffix=".yaml") as f:
...     predicates_path = Path(f.name)
...     yaml.dump(data, f)
...     with tempfile.NamedTemporaryFile(mode="w", suffix=".yaml") as f2:
...         config_path = Path(f2.name)
...         cfg = TaskExtractorConfig.load(config_path, predicates_path)
```

```
Traceback (most recent call last):
    ...
ValueError: Unrecognized keys in configuration file: 'foo, trigger'
>>> predicates = {"foo bar": PlainPredicateConfig("foo")}
>>> trigger = EventConfig("foo")
>>> config = TaskExtractorConfig(predicates=predicates, trigger=trigger,
↪windows={})
Traceback (most recent call last):
    ...
ValueError: Predicate name 'foo bar' is invalid; must be composed of
↪alphanumeric or '_' characters.
>>> predicates = {"foo": str("foo")}
>>> trigger = EventConfig("foo")
>>> config = TaskExtractorConfig(predicates=predicates, trigger=trigger,
↪windows={})
...
Traceback (most recent call last):
    ...
ValueError: Invalid predicate configuration for 'foo': foo. Must be either a
↪PlainPredicateConfig or
DerivedPredicateConfig object. Got: <class 'str'>
>>> predicates = {
...     "foo": PlainPredicateConfig("foo"),
...     "foobar": DerivedPredicateConfig("or(foo, bar)"),
... }
>>> trigger = EventConfig("foo")
>>> config = TaskExtractorConfig(predicates=predicates, trigger=trigger,
↪windows={})
Traceback (most recent call last):
    ...
KeyError: "Missing 1 relationships: Derived predicate 'foobar' references
↪undefined predicate 'bar'"
>>> predicates = {"foo": PlainPredicateConfig("foo")}
>>> trigger = EventConfig("foo")
>>> windows = {"foo bar": WindowConfig("gap.end", "start + 24h", True, True)}
>>> config = TaskExtractorConfig(predicates=predicates, trigger=trigger,
↪windows=windows)
Traceback (most recent call last):
    ...
ValueError: Window name 'foo bar' is invalid; must be composed of alphanumeric
↪or '_' characters.
>>> windows = {"foo": WindowConfig("gap.end", "start + 24h", True, True, {},
↪"bar")}
>>> config = TaskExtractorConfig(predicates=predicates, trigger=trigger,
↪windows=windows)
Traceback (most recent call last):
    ...
ValueError: Label must be one of the defined predicates. Got: bar for window
↪'foo'
>>> windows = {"foo": WindowConfig("gap.end", "start + 24h", True, True, {},
↪"foo", "bar")}
>>> config = TaskExtractorConfig(predicates=predicates, trigger=trigger,
```

```
→windows=windows)
Traceback (most recent call last):
    ...
ValueError: Index timestamp must be either 'start' or 'end'. Got: bar for␣
→window 'foo'
>>> windows = {
...     "foo": WindowConfig("gap.end", "start + 24h", True, True, {}, "foo"),
...     "bar": WindowConfig("gap.end", "start + 24h", True, True, {}, "foo")
... }
>>> config = TaskExtractorConfig(predicates=predicates, trigger=trigger,␣
→windows=windows)
Traceback (most recent call last):
    ...
ValueError: Only one window can be labeled, found 2 labeled windows: foo, bar
>>> windows = {
...     "foo": WindowConfig("gap.end", "start + 24h", True, True, {}, "foo",
→"start"),
...     "bar": WindowConfig("gap.end", "start + 24h", True, True, {}, index_
→timestamp="start")
... }
>>> config = TaskExtractorConfig(predicates=predicates, trigger=trigger,␣
→windows=windows)
...
Traceback (most recent call last):
    ...
ValueError: Only the 'start'/'end' of one window can be used as the index␣
→timestamp, found
2 windows with index_timestamp: foo, bar
>>> predicates = {"foo": PlainPredicateConfig("foo")}
>>> trigger = EventConfig("bar")
>>> config = TaskExtractorConfig(predicates=predicates, trigger=trigger,␣
→windows={})
Traceback (most recent call last):
    ...
KeyError: "Trigger event predicate 'bar' not found in predicates: foo"
```

property **derived_predicates** : OrderedDict[str, *DerivedPredicateConfig*]

Returns an ordered dictionary mapping derived predicates to their configs in a proper order.

**index_timestamp_window** : str | None = **None**

**label_window** : str | None = **None**

classmethod **load**(config_path: *str* | *Path*, predicates_path: *str* | *Path* | *None* = **None**) →
                *TaskExtractorConfig*

Load a configuration file from the given path and return it as a dict.

> **Parameters**
>
> > **config_path: str | Path**
> > The path to which a configuration object will be read from in YAML form.
>
> **Raises**
>
> > • **FileNotFoundError** – If the file does not exist.

---

- **ValueError** – If the file is not a ".yaml" file.

**Examples**

```
>>> import tempfile
>>> yaml = ruamel.yaml.YAML(typ="safe", pure=True)
>>> config_dict = {
...     "metadata": {'description': 'A test configuration file'},
...     "description": 'this is a test',
...     "predicates": {"admission": {"code": "admission"}},
...     "trigger": "admission",
...     "windows": {
...         "start": {
...             "start": None, "end": "trigger + 24h", "start_inclusive": True,
...             "end_inclusive": True,
...         }
...     },
... }
>>> with tempfile.NamedTemporaryFile(mode="w", suffix=".yaml") as f:
...     config_path = Path(f.name)
...     yaml.dump(config_dict, f)
...     cfg = TaskExtractorConfig.load(config_path)
>>> cfg
TaskExtractorConfig(predicates={'admission': PlainPredicateConfig(code='admission',
                                            value_min=None, value_max=None,
                                            value_min_inclusive=None, value_
→max_inclusive=None,
                                            static=False, other_cols={})},
                    trigger=EventConfig(predicate='admission'),
                    windows={'start': WindowConfig(start=None, end='trigger + 24h',
                                            start_inclusive=True, end_
→inclusive=True, has={},
                                            label=None, index_timestamp=None)},
                    label_window=None, index_timestamp_window=None)
```

```
>>> predicates_dict = {
...     "metadata": {'description': 'A test predicates file'},
...     "description": 'this is a test',
...     "patient_demographics": {"brown_eyes": {"code": "eye_color//BR"}},
...     "predicates": {"admission": {"code": "admission"}},
... }
>>> no_predicates_config = {k: v for k, v in config_dict.items() if k !=
→"predicates"}
>>> with (tempfile.NamedTemporaryFile(mode="w", suffix=".yaml") as config_fp,
...       tempfile.NamedTemporaryFile(mode="w", suffix=".yaml") as pred_fp):
...     config_path = Path(config_fp.name)
...     pred_path = Path(pred_fp.name)
...     yaml.dump(no_predicates_config, config_fp)
...     yaml.dump(predicates_dict, pred_fp)
...     cfg = TaskExtractorConfig.load(config_path, pred_path)
>>> cfg
TaskExtractorConfig(predicates={'admission': PlainPredicateConfig(code='admission',
```

(continues on next page)

```
                                          value_min=None, value_max=None,
                                          value_min_inclusive=None, value_
→max_inclusive=None,
                                          static=False, other_cols={}),
                          'brown_eyes': PlainPredicateConfig(code='eye_
→color//BR',
                                          value_min=None, value_max=None,
                                          value_min_inclusive=None,
                                          value_max_inclusive=None,
→static=True,
                                          other_cols={})},
                  trigger=EventConfig(predicate='admission'),
                  windows={'start': WindowConfig(start=None, end='trigger + 24h',
                                          start_inclusive=True, end_
→inclusive=True, has={},
                                          label=None, index_timestamp=None)},
                  label_window=None, index_timestamp_window=None)
```

```
>>> config_dict = {
...      "metadata": {'description': 'A test configuration file'},
...      "description": 'this is a test for joining static and plain predicates',
...      "patient_demographics": {"male": {"code": "MALE"}, "female": {"code":
→"FEMALE"}},
...      "predicates": {"normal_male_lab_range": {"code": "LAB", "value_min": 0,
→"value_max": 100,
...                     "value_min_inclusive": True, "value_max_inclusive": True},
...                     "normal_female_lab_range": {"code": "LAB", "value_min": 0,
→"value_max": 90,
...                     "value_min_inclusive": True, "value_max_inclusive": True},
...                     "normal_lab_male": {"expr": "and(normal_male_lab_range,
→male)"},
...                     "normal_lab_female": {"expr": "and(normal_female_lab_range,
→ female)"}},
...      "trigger": "_ANY_EVENT",
...      "windows": {
...          "start": {
...              "start": None, "end": "trigger + 24h", "start_inclusive": True,
...              "end_inclusive": True, "has": {"normal_lab_male": "(1, None)"},
...          }
...      },
... }
>>> with tempfile.NamedTemporaryFile(mode="w", suffix=".yaml") as f:
...      config_path = Path(f.name)
...      yaml.dump(config_dict, f)
...      cfg = TaskExtractorConfig.load(config_path)
>>> cfg.predicates.keys()
dict_keys(['normal_lab_male', 'normal_male_lab_range', 'female', 'male'])
```

```
>>> config_dict = {
...      "metadata": {'description': 'A test configuration file'},
...      "description": 'this is a test for nested derived predicates',
```

---

```
...        "patient_demographics": {"male": {"code": "MALE"}, "female": {"code":
→"FEMALE"}},
...        "predicates": {"abnormally_low_male_lab_range": {"code": "LAB", "value_
→max": 90,
...                    "value_max_inclusive": False},
...                    "abnormally_low_female_lab_range": {"code": "LAB", "value_
→max": 80,
...                    "value_max_inclusive": False},
...                    "abnormally_high_lab_range": {"code": "LAB", "value_min":␣
→120,
...                    "value_min_inclusive": False},
...                    "abnormal_lab_male_range": {"expr":
...                            "or(abnormally_low_male_lab_range, abnormally_
→high_lab_range)"},
...                    "abnormal_lab_female_range": {"expr":
...                            "or(abnormally_low_female_lab_range, abnormally_
→high_lab_range)"},
...                    "abnormal_lab_male": {"expr": "and(abnormal_lab_male_range,
→ male)"},
...                    "abnormal_lab_female": {"expr": "and(abnormal_lab_female_
→range, female)"},
...                    "abnormal_labs": {"expr": "or(abnormal_lab_male, abnormal_
→lab_female)"}},
...        "trigger": "_ANY_EVENT",
...        "windows": {
...            "start": {
...                "start": None, "end": "trigger + 24h", "start_inclusive": True,
...                "end_inclusive": True, "label": "abnormal_labs",
...                "has": {"abnormal_labs": "(1, None)"},
...            }
...        },
... }
>>> with tempfile.NamedTemporaryFile(mode="w", suffix=".yaml") as f:
...        config_path = Path(f.name)
...        yaml.dump(config_dict, f)
...        cfg = TaskExtractorConfig.load(config_path)
>>> cfg.predicates.keys()
dict_keys(['abnormal_lab_female', 'abnormal_lab_female_range', 'abnormal_lab_male',
'abnormal_lab_male_range', 'abnormal_labs', 'abnormally_high_lab_range',
'abnormally_low_female_lab_range', 'abnormally_low_male_lab_range', 'female', 'male
→'])
```

```
>>> predicates_dict = {
...        "metadata": {'description': 'A test predicates file'},
...        "description": 'this is a test',
...        "patient_demographics": {"brown_eyes": {"code": "eye_color//BR"}},
...        "predicates": {'admission': "invalid"},
... }
>>> with (tempfile.NamedTemporaryFile(mode="w", suffix=".yaml") as config_fp,
...        tempfile.NamedTemporaryFile(mode="w", suffix=".yaml") as pred_fp):
...        config_path = Path(config_fp.name)
...        pred_path = Path(pred_fp.name)
```

```
...         yaml.dump(no_predicates_config, config_fp)
...         yaml.dump(predicates_dict, pred_fp)
...         cfg = TaskExtractorConfig.load(config_path, pred_path)
Traceback (most recent call last):
    ...
ValueError: Predicate 'admission' is not defined correctly in the configuration␣
↪file. Currently
defined as the string: invalid. Please refer to the documentation for the␣
↪supported formats.
>>> predicates_dict = {
...         "predicates": {'adm': {"code": "admission"}},
... }
>>> with (tempfile.NamedTemporaryFile(mode="w", suffix=".yaml") as config_fp,
...         tempfile.NamedTemporaryFile(mode="w", suffix=".yaml") as pred_fp):
...         config_path = Path(config_fp.name)
...         pred_path = Path(pred_fp.name)
...         yaml.dump(no_predicates_config, config_fp)
...         yaml.dump(predicates_dict, pred_fp)
...         cfg = TaskExtractorConfig.load(config_path, pred_path)
Traceback (most recent call last):
    ...
KeyError: "Something referenced predicate 'admission' that wasn't defined in␣
↪the configuration."
>>> config_dict = {
...         "predicates": {"A": {"code": "A"}, "B": {"code": "B"}, "A_or_B": {"expr
↪": "or(A, B)"},
...                        "A_or_B_and_C": {"expr": "and(A_or_B, C)"}},
...         "trigger": "_ANY_EVENT",
...         "windows": {"start": {"start": None, "end": "trigger + 24h", "start_
↪inclusive": True,
...                 "end_inclusive": True, "has": {"A_or_B_and_C": "(1, None)"}}},
... }
>>> with tempfile.NamedTemporaryFile(mode="w", suffix=".yaml") as f:
...         config_path = Path(f.name)
...         yaml.dump(config_dict, f)
...         cfg = TaskExtractorConfig.load(config_path)
Traceback (most recent call last):
    ...
KeyError: "Predicate 'C' referenced in 'A_or_B_and_C' is not defined in the␣
↪configuration."
```

property **plain_predicates** : dict[str, *aces.config.PlainPredicateConfig*]

    code} format.

        **Type**

            Returns a dictionary of plain predicates in {name

**predicates** : dict[str, *aces.config.PlainPredicateConfig* | *aces.config.DerivedPredicateConfig*]

property **predicates_DAG** : DiGraph

**trigger** : *EventConfig*

property **window_tree** : Node

---

**windows** : dict[str, *aces.config.WindowConfig*] | None

**class** aces.config.**WindowConfig**(start: str | None, end: str | None, start_inclusive: bool, end_inclusive: bool, has: dict[str, str] = <factory>, label: str | None = None, index_timestamp: str | None = None)

Bases: object

A configuration object for defining a window in the task extraction process.

This defines the boundary points and constraints for a window in the patient record in the task extraction process.

> **Parameters**
>
> **start**
>> The boundary conditions for the start of the window. This (like end) can either be None, in which case the window starts at the beginning of the patient record, or is expressed through a string language that expresses a relative startpoint to this window either in reference to (a) another window's start or end event, (b) this window's *end* event. In case (a), this window's end event must either be None or reference this window's start event, and in case (b), this window's end event must reference a different window's start or end event. The string language is as follows:
>>
>> • None: The window starts at the beginning of the patient record.
>>
>> • $REFERENCED <- $PREDICATE or $REFERENCED -> $PREDICATE: The window starts at the closest event satisfying the predicate $PREDICATE relative to the $REFERENCED event. Form $REFERENCED <- $PREDICATE means that the window starts at the closest event _prior **to_** the $REFERENCED event that satisfies the predicate $PREDICATE, and the other form is analogous but with the closest event _after_ the $REFERENCED event.
>>
>> • $REFERENCED +- timedelta: The window starts at the $REFERENCED event plus or minus the specified timedelta. The timedelta is expressed through the string language specified in the utils.parse_timedelta function.
>>
>> • $REFERENCED: The window starts at the $REFERENCED event.
>>
>> **In all cases, the $REFERENCED event must be either**
>>
>>> • The name of another window's start or end event, as specified by $WINDOW_NAME.start or $WINDOW_NAME.end.
>>>
>>> • This window's end event, as specified by end.
>>
>> In the case that $REFERENCED is this window's end event, the window must be defined such that start would precede end in the order of the patient record (e.g., $PREDICATE -> end is invalid, and end - timedelta is invalid).
>
> **end**
>> The name of the event that ends the window. See the documentation for start for more details on the specification language.
>
> **start_inclusive**
>> Whether or not the start event is included in the window. Note that this term can not only dictate whether an event's counts are included in the summarization of the window, but also whether or not an event satisfying $PREDICATE can be used as the boundary of an event. E.g., if we have that start_inclusive=False and the *end* field is equal to start -> $PREDICATE, and it so happens that the *start* event itself satisfies $PREDICATE, the fact that start_inclusive=False will mean that we do not consider the *start* event itself to

be a valid start to any window that ends at the same *start* event, as its timestamp when considered as the prospective "window start timestamp" occurs "after" the effective timestamp of itself when considered as the $PREDICATE event that marks the window end given that `start_inclusive=False` and thus we will think of the window as truly starting an iota after the timestamp of the *start* event itself.

**end_inclusive**
Whether or not the end event is included in the window.

**has**
A dictionary of predicates that must be present in the window, mapped to tuples of the form (`min_valid`, `max_valid`) that define the valid range the count of observations of the named predicate that must be found in a window for it to be considered valid. Either `min_valid` or `max_valid` constraints can be `None`, in which case those endpoints are left unconstrained. Likewise, unreferenced predicates are also left unconstrained. Note that as predicate counts are always integral, this specification does not need an additional inclusive/exclusive endpoint field, as one can simply increment the bound by one in the appropriate direction to achieve the result. Instead, this bound is always interpreted to be inclusive, so a window would satisfy the constraint for predicate name with constraint `name: (1, 2)` if the count of observations of predicate name in a window was either 1 or 2. All constraints in the dictionary must be satisfied on a window for it to be included.

**label**
A string that specifies the name of a predicate to be used as the label for the task. The predicate count of the window this field is specified in will be extracted as a column in the final result. Hence, there can only be one 'label' per TaskExtractorConfig. If more than one 'label' is specified, an error is raised. If the specified 'label' is not a defined predicate, an error is also raised. If no 'label' is specified, there will be not be a 'label' column.

**index_timestamp**
A string that is either 'start' or 'end' and is used to index result rows. If it is defined, there will be an 'index_timestamp' column in the result with its values equal to the 'start' or 'end' timestamp of the window in which it was specified. Usually, this will be specified to indicate the time of prediction for the task, which is often the 'end' of the input window. There can only be one 'index_timestamp' per TaskExtractorConfig. If more than one 'index_timestamp' is specified, an error is raised. If the specified 'index_timestamp' is not 'start' or 'end', an error is also raised. If no 'index_timestamp' is defined, there will be no 'index_timestamp' column.

Raises
> **ValueError** – If the window is misconfigured in any of a variety of ways; see below for examples.

**Examples**

```
>>> input_window = WindowConfig(
...     start=None,
...     end="trigger + 2 days",
...     start_inclusive=True,
...     end_inclusive=True,
...     has={"_ANY_EVENT": "(5, None)"},
...     index_timestamp="end",
... )
>>> input_window.referenced_event
('trigger',)
>>> # This window does not reference any "true" external predicates, only implicit␣
```

```
→predicates like
>>> # start, end, and * events, so this list should be empty.
>>> sorted(input_window.referenced_predicates)
['_ANY_EVENT']
>>> input_window.start_endpoint_expr
ToEventWindowBounds(left_inclusive=True,
                    end_event='-_RECORD_START',
                    right_inclusive=True,
                    offset=datetime.timedelta(0))
>>> input_window.end_endpoint_expr
TemporalWindowBounds(left_inclusive=False,
                     window_size=datetime.timedelta(days=2),
                     right_inclusive=False,
                     offset=datetime.timedelta(0))
>>> input_window.root_node
'end'
>>> gap_window = WindowConfig(
...     start="input.end",
...     end="start + 24h",
...     start_inclusive=False,
...     end_inclusive=True,
...     has={"discharge": "(None, 0)", "death": "(None, 0)"}
... )
>>> gap_window.referenced_event
('input', 'end')
>>> sorted(gap_window.referenced_predicates)
['death', 'discharge']
>>> gap_window.start_endpoint_expr is None
True
>>> gap_window.end_endpoint_expr
TemporalWindowBounds(left_inclusive=False,
                     window_size=datetime.timedelta(days=1),
                     right_inclusive=True,
                     offset=datetime.timedelta(0))
>>> gap_window.root_node
'start'
>>> gap_window = WindowConfig(
...     start="input.end",
...     end="start + 0h",
...     start_inclusive=False,
...     end_inclusive=True,
...     has={"discharge": "(None, 0)", "death": "(None, 0)"}
... )
>>> gap_window.referenced_event
('input', 'end')
>>> sorted(gap_window.referenced_predicates)
['death', 'discharge']
>>> gap_window.start_endpoint_expr is None
True
>>> gap_window.end_endpoint_expr is None
True
>>> gap_window.root_node
```

```
'start'
>>> target_window = WindowConfig(
...     start="gap.end",
...     end="start -> discharge_or_death",
...     start_inclusive=False,
...     end_inclusive=True,
...     has={}
... )
>>> target_window.referenced_event
('gap', 'end')
>>> sorted(target_window.referenced_predicates)
['discharge_or_death']
>>> target_window.start_endpoint_expr is None
True
>>> target_window.end_endpoint_expr
ToEventWindowBounds(left_inclusive=False,
                    end_event='discharge_or_death',
                    right_inclusive=True,
                    offset=datetime.timedelta(0))
>>> target_window.root_node
'start'
>>> target_window = WindowConfig(
...     start="end",
...     end="gap.end <- discharge_or_death",
...     start_inclusive=False,
...     end_inclusive=True,
...     has={}
... )
>>> target_window.referenced_event
('gap', 'end')
>>> sorted(target_window.referenced_predicates)
['discharge_or_death']
>>> target_window.start_endpoint_expr is None
True
>>> target_window.end_endpoint_expr
ToEventWindowBounds(left_inclusive=False,
                    end_event='-discharge_or_death',
                    right_inclusive=False,
                    offset=datetime.timedelta(0))
>>> target_window.root_node
'end'
```

```
>>> invalid_window = WindowConfig(
...     start="gap.end gap.start",
...     end="start -> discharge_or_death",
...     start_inclusive=False,
...     end_inclusive=True,
...     has={}
... )
Traceback (most recent call last):
    ...
ValueError: Window boundary reference must be either a valid alphanumeric/'_'␣
```

```
↪string or a reference to
    another window's start or end event, formatted as a valid alphanumeric/'_'␣
↪string, followed by
    '.start' or '.end'.
    Got: 'gap.end gap.start'
>>> invalid_window = WindowConfig(
...     start="input",
...     end="start window -> discharge_or_death",
...     start_inclusive=False,
...     end_inclusive=True,
...     has={"discharge": "(None, 0)", "death": "(None, 0)"}
... )
Traceback (most recent call last):
    ...
ValueError: Window boundary reference must be either a valid alphanumeric/'_'␣
↪string or a reference
to another window's start or end event, formatted as a valid alphanumeric/'_'␣
↪string, followed by
'.start' or '.end'. Got: 'start window'
>>> invalid_window = WindowConfig(
...     start="input",
...     end="window.foo -> discharge_or_death",
...     start_inclusive=False,
...     end_inclusive=True,
...     has={"discharge": "(None, 0)", "death": "(None, 0)"}
... )
Traceback (most recent call last):
    ...
ValueError: Window boundary reference must be either a valid alphanumeric/'_'␣
↪string or a reference
to another window's start or end event, formatted as a valid alphanumeric/'_'␣
↪string, followed by
'.start' or '.end'. Got: 'window.foo'
>>> invalid_window = WindowConfig(
...     start=None, end=None, start_inclusive=True, end_inclusive=True, has={}
... )
Traceback (most recent call last):
    ...
ValueError: Window cannot progress from the start of the record to the end of the␣
↪record.
>>> invalid_window = WindowConfig(
...     start="input.end",
...     end="start - 2d",
...     start_inclusive=False,
...     end_inclusive=True,
...     has={"discharge": "(None, 0)", "death": "(None, 0)"}
... )
Traceback (most recent call last):
    ...
ValueError: Window start will not occur before window end! Got: input.end -> start -
↪ 2d
>>> invalid_window = WindowConfig(
```

```
...     start="end -> predicate",
...     end="input.end",
...     start_inclusive=False,
...     end_inclusive=True,
...     has={"discharge": "(None, 0)", "death": "(None, 0)"}
... )
Traceback (most recent call last):
    ...
ValueError: Window start will not occur before window end! Got: end -> predicate ->␣
→input.end
>>> invalid_window = WindowConfig(
...     start="end - 24h", end="start + 1d", start_inclusive=True, end_
→inclusive=True, has={}
... )
Traceback (most recent call last):
    ...
ValueError: Exactly one of the start or end of the window must reference the other.
Got: end - 24h -> start + 1d
>>> invalid_window = WindowConfig(
...     start="input.end",
...     end="input.end + 2d",
...     start_inclusive=False,
...     end_inclusive=True,
...     has={"discharge": "(None, 0)", "death": "(None, 0)"}
... )
Traceback (most recent call last):
    ...
ValueError: Exactly one of the start or end of the window must reference the other.
Got: input.end -> input.end + 2d
>>> invalid_window = WindowConfig(
...     start="input.end",
...     end="start + -24h",
...     start_inclusive=False,
...     end_inclusive=True,
...     has={"discharge": "(None, 0)", "death": "(None, 0)"}
... )
Traceback (most recent call last):
    ...
ValueError: Window boundary cannot contain both '+' and '-' operators.
>>> invalid_window = WindowConfig(
...     start="input.end",
...     end="start + invalid time string.",
...     start_inclusive=False,
...     end_inclusive=True,
...     has={"discharge": "(None, 0)", "death": "(None, 0)"}
... )
Traceback (most recent call last):
    ...
ValueError: Failed to parse timedelta from window offset for 'invalid time string.'
>>> target_window = WindowConfig(
...     start="gap.end",
...     end="start <-> discharge_or_death",
```

```
...        start_inclusive=False,
...        end_inclusive=True,
...        has={}
... )
Traceback (most recent call last):
    ...
ValueError: Window boundary cannot contain both '->' and '<-' operators.
>>> invalid_window = WindowConfig(
...        start="input.end",
...        end="input.end + 2d",
...        start_inclusive=False,
...        end_inclusive=True,
...        has={"discharge": "(0)", "death": "(None, 0)"}
... )
Traceback (most recent call last):
    ...
ValueError: Invalid constraint format: discharge.
Expected format: '(min, max)'. Got: '(0)'
```

**property constraint_predicates** : set[str]

**end** : str | None

**property end_endpoint_expr** : None | *ToEventWindowBounds* | *TemporalWindowBounds*

**end_inclusive** : bool

**has** : dict[str, str]

**index_timestamp** : str | None = **None**

**label** : str | None = **None**

**property referenced_event** : tuple[str]

**property referenced_predicates** : set[str]

**property root_node** : str

    Returns 'start' if the end of the window is defined relative to the start and 'end' otherwise.

**start** : str | None

**property start_endpoint_expr** : None | *ToEventWindowBounds* | *TemporalWindowBounds*

**start_inclusive** : bool

## aces.constraints module

Contains utilities for validating that windows satisfy a set of constraints.

aces.constraints.**check_constraints**(window_constraints: *dict[str, tuple[int | None, int | None]]*,
summary_df: *DataFrame*) → DataFrame

> Checks the constraints on the counts of predicates in the summary dataframe.

> > **Parameters**

> > > **window_constraints: dict[str, tuple[int | None, int | None]]**
> > > constraints on counts of predicates that must be satsified, organized as a dictionary from
> > > predicate column name to the lowerbound and upper bound range required for that constraint
> > > to be satisfied.

> > > **summary_df: DataFrame**
> > > A dataframe containing a row for every possible prospective window to be analyzed. The
> > > only columns expected are predicate columns within the `window_constraints` dictionary.

> Returns: A filtered dataframe containing only the rows that satisfy the constraints.

> > **Raises**
> > **ValueError** – If the constraint for a column is empty.

### Examples

```
>>> from datetime import datetime
>>> df = pl.DataFrame({
...     "subject_id": [1, 1, 1, 1, 2, 2],
...     "timestamp": [
...         # Subject 1
...         datetime(year=1989, month=12, day=1, hour=12, minute=3),
...         datetime(year=1989, month=12, day=2, hour=5,  minute=17),
...         datetime(year=1989, month=12, day=2, hour=12, minute=3),
...         datetime(year=1989, month=12, day=6, hour=11, minute=0),
...         # Subject 2
...         datetime(year=1989, month=12, day=1, hour=13, minute=14),
...         datetime(year=1989, month=12, day=3, hour=15, minute=17),
...     ],
...     "is_A": [1, 4, 1, 3, 3,  3],
...     "is_B": [0, 2, 0, 2, 10, 2],
...     "is_C": [1, 1, 1, 0, 1,  1],
... })
>>> check_constraints({"is_A": (None, None), "is_B": (2, 6), "is_C": (1, 1)}, df)
Traceback (most recent call last):
    ...
ValueError: Invalid constraint for 'is_A': None - None
>>> check_constraints({"is_A": (2, 1), "is_B": (2, 6), "is_C": (1, 1)}, df)
Traceback (most recent call last):
    ...
ValueError: Invalid constraint for 'is_A': 2 - 1
>>> check_constraints({"is_A": (3, 4), "is_B": (2, 6), "is_C": (1, 1)}, df)
shape: (2, 5)
+----+-------+--+--+--+
| subject_id | timestamp          | is_A | is_B | is_C |
```

```
| ---        | ---                 | --- | --- | --- |
| i64        | datetime[μs]        | i64 | i64 | i64 |
|----|-------|--|--|--|
| 1          | 1989-12-02 05:17:00 | 4   | 2   | 1   |
| 2          | 1989-12-03 15:17:00 | 3   | 2   | 1   |
└----+-------+--+--+--+
>>> check_constraints({"is_A": (3, 4), "is_B": (2, None), "is_C": (None, 1)}, df)
shape: (4, 5)
+----+-------+--+--+--+
| subject_id | timestamp           | is_A | is_B | is_C |
| ---        | ---                 | ---  | ---  | ---  |
| i64        | datetime[μs]        | i64  | i64  | i64  |
|----|-------|--|--|--|
| 1          | 1989-12-02 05:17:00 | 4    | 2    | 1    |
| 1          | 1989-12-06 11:00:00 | 3    | 2    | 0    |
| 2          | 1989-12-01 13:14:00 | 3    | 10   | 1    |
| 2          | 1989-12-03 15:17:00 | 3    | 2    | 1    |
└----+-------+--+--+--+
>>> predicates_df = pl.DataFrame({
...     "subject_id": [1, 1, 3],
...     "timestamp": [datetime(1980, 12, 28), datetime(2010, 6, 20), datetime(2010,
↪5, 11)],
...     "A": [False, False, False],
...     "_ANY_EVENT": [True, True, True],
... })
>>> check_constraints({"_ANY_EVENT": (1, None)}, predicates_df)
shape: (3, 4)
+----+-------+---+----+
| subject_id | timestamp           | A     | _ANY_EVENT |
| ---        | ---                 | ---   | ---        |
| i64        | datetime[μs]        | bool  | bool       |
|----|-------|---|----|
| 1          | 1980-12-28 00:00:00 | false | true       |
| 1          | 2010-06-20 00:00:00 | false | true       |
| 3          | 2010-05-11 00:00:00 | false | true       |
└----+-------+---+----+
```

aces.constraints.**check_static_variables**(patient_demographics: *list[str]*, predicates_df: *DataFrame*) → DataFrame

Checks the constraints on the counts of predicates in the summary dataframe.

> **Parameters**
>
> > **patient_demographics: list[str]**
> > List of columns representing static patient demographics.
> >
> > **predicates_df: DataFrame**
> > Dataframe containing a row for each event with patient demographics and timestamps.
>
> Returns: A filtered dataframe containing only the rows that satisfy the patient demographics.
>
> > **Raises**
> > **ValueError** – If the static predicate used by constraint is not in the predicates dataframe.

### Examples

```
>>> from datetime import datetime
>>> predicates_df = pl.DataFrame({
...     "subject_id": [1, 1, 1, 1, 1, 2, 2, 2],
...     "timestamp": [
...         # Subject 1
...         None,
...         datetime(year=1989, month=12, day=1, hour=12, minute=3),
...         datetime(year=1989, month=12, day=2, hour=5,  minute=17),
...         datetime(year=1989, month=12, day=2, hour=12, minute=3),
...         datetime(year=1989, month=12, day=6, hour=11, minute=0),
...         # Subject 2
...         None,
...         datetime(year=1989, month=12, day=1, hour=13, minute=14),
...         datetime(year=1989, month=12, day=3, hour=15, minute=17),
...     ],
...     "is_A": [0, 1, 4, 1, 0, 3, 3,  3],
...     "is_B": [0, 0, 2, 0, 0, 2, 10, 2],
...     "is_C": [0, 1, 1, 1, 0, 0, 1,  1],
...     "male": [1, 0, 0, 0, 0, 0, 0,  0]
... })
```

```
>>> check_static_variables(['male'], predicates_df)
shape: (4, 5)
+----+-------+--+--+--+
| subject_id | timestamp           | is_A | is_B | is_C |
| ---        | ---                 | ---  | ---  | ---  |
| i64        | datetime[µs]        | i64  | i64  | i64  |
|----|-------|--|--|--|
| 1          | 1989-12-01 12:03:00 | 1    | 0    | 1    |
| 1          | 1989-12-02 05:17:00 | 4    | 2    | 1    |
| 1          | 1989-12-02 12:03:00 | 1    | 0    | 1    |
| 1          | 1989-12-06 11:00:00 | 0    | 0    | 0    |
└----+-------+--+--+--+
>>> check_static_variables(['female'], predicates_df)
Traceback (most recent call last):
    ...
ValueError: Static predicate 'female' not found in the predicates dataframe.
```

### aces.expand_shards module

aces.expand_shards.**expand_shards**(*shards*: *str*) → str

This function expands a set of shard prefixes and number of shards into a list of all shards or expands a directory into a list of all files within it.

This can be useful with Hydra applications where you wish to expand a list of options for the sweeper to sweep over but can't use an OmegaConf resolver as those are evaluated after the sweep has been initialized.

> **Parameters**
>
> > ***shards*: str**
> > A list of shard prefixes and number of shards to expand, or a directory to list all files.

> **Returns:** **A comma-separated list of all shards, expanded to the specified number, or all files in the**
> directory.

**Examples**

```
>>> import polars as pl
>>> import tempfile
```

```
>>> expand_shards("train/4", "val/IID/1", "val/prospective/1")
'train/0,train/1,train/2,train/3,val/IID/0,val/prospective/0'
>>> expand_shards("data/data_4", "data/test_4")
'data/data_0,data/data_1,data/data_2,data/data_3,data/test_0,data/test_1,data/test_2,
↪data/test_3'
```

```
>>> parquet_data = pl.DataFrame({
...     "subject_id": [1, 1, 1, 2, 3],
...     "time": ["1/1/1989 00:00", "1/1/1989 01:00", "1/1/1989 01:00", "1/1/1989 02:
↪00", None],
...     "code": ['admission', 'discharge', 'discharge', 'admission', "gender"],
... }).with_columns(pl.col("time").str.strptime(pl.Datetime, format="%m/%d/%Y %H:%M
↪"))
```

```
>>> with tempfile.TemporaryDirectory() as tmpdirname:
...     for i in range(4):
...         if i in (0, 2):
...             data_path = Path(tmpdirname) / f"evens/0/file_{i}.parquet"
...             data_path.parent.mkdir(parents=True, exist_ok=True)
...         else:
...             data_path = Path(tmpdirname) / f"{i}.parquet"
...         parquet_data.write_parquet(data_path)
...     json_fp = Path(tmpdirname) / "4.json"
...     _ = json_fp.write_text('["foo"]')
...     result = expand_shards(tmpdirname)
...     sorted(result.split(","))
['1', '3', 'evens/0/file_0', 'evens/0/file_2']
```

```
>>> expand_shards("train.invalid")
Traceback (most recent call last):
    ...
ValueError: Invalid shard format: train.invalid
```

aces.expand_shards.**main**() → None

## aces.extract_subtree module

This module contains the functions for extracting constraint hierarchy subtrees.

aces.extract_subtree.**extract_subtree**(subtree*: Node*, subtree_anchor_realizations*: DataFrame*,
        predicates_df*: DataFrame*, subtree_root_offset*: timedelta* =
        *datetime*.*timedelta*(**0**)) → DataFrame

The main algorithmic recursive call to identify valid realizations of a subtree.

This function takes in a global `predicates_df`, a subtree of constraints, and the temporal offset that any real-ization the root timestamp of the subtree would have relative to the corresponding subtree anchor. It will use this information to recurse through the subtree and identify any valid realizations of this subtree, returning them in a dataframe keyed by the subtree anchor event timestamps and with a series of columns containing subtree edge start and end timestamps and contained predicate counts.

**Parameters**

**subtree: Node**

The subtree to extract realizations from. This is specified through a `BigTree.Node` object. This `Node` object can have zero or more children, each of which must have the following:

- `name`: The name of the subtree root.

- `constraints`: The constraints associated with the subtree root, structured as a dictio-nary from predicate column name to a tuple containing the valid (inclusive) minimum and maximum values the predicate counts can take on (use `None` for no constraint).

- `endpoint_expr`: A tuple containing the endpoint expression for the subtree root. This should be either a `ToEventWindowBounds` or a `TemporalWindowBounds` formatted tu-ple object, less the offset parameter, as that is something determined by the structure of the tree, not pre-set in the configuration.

**subtree_anchor_realizations: DataFrame**

The dataframe containing the anchor to subtree root mapping. This dataframe will have the following columns:

- `"subject_id"`: The ID of the subject. All analyses will be performed within `subject_id` groups.

- `subtree_anchor_timestamp`: The timestamp of all possible prospective subtree anchor realizations. These will all correspond to extant events (`subject_id`, `timestamp` pairs in `predicates_df`).

**predicates_df: DataFrame**
The dataframe containing the predicates to summarize. This dataframe will have the follow-ing mandatory columns:

**subtree_root_offset: timedelta = datetime.timedelta(0)**
The temporal offset of the subtree root relative to the subtree anchor.

**Returns**
The result of the subtree extraction, containing subjects who satisfy the conditions defined in the subtree. Timestamps for the start/end boundaries of each window specified in the subtree configuration, as well as predicate counts for each window, are provided.

**Return type**
pl.DataFrame

**Examples**

```
>>> from bigtree import Node
>>> from datetime import datetime
>>> from .types import ToEventWindowBounds, TemporalWindowBounds
>>> # We'll use an example for in-hospital mortality prediction. Our root event of
→the tree will be
>>> # an admission event.
>>> root = Node("admission")
>>> #
>>> #### BRANCH 1 ####
>>> # Our first branch off of admission will be checking a gap window, then our
→target window.
>>> # Node 1 will represent our gap window. We say that in the 24 hours after the
→admission, there
>>> # should be no discharges, deaths, or covid events.
>>> gap_node = Node("gap") # This sets the node's name.
>>> gap_node.endpoint_expr = TemporalWindowBounds(True, timedelta(days=2), True)
>>> gap_node.constraints = {
...     "is_discharge": (None, 0), "is_death": (None, 0), "is_covid_dx": (None, 0)
... }
>>> gap_node.parent = root
>>> # Node 2 will start our target window and span until the next discharge or
→death event.
>>> # There should be no covid events.
>>> target_node = Node("target") # This sets the node's name.
>>> target_node.endpoint_expr = ToEventWindowBounds(True, "is_discharge", True)
>>> target_node.constraints = {"is_covid_dx": (None, 0)}
>>> target_node.parent = gap_node
>>> #
>>> #### BRANCH 2 ####
>>> # Finally, for our second branch, we will impose no constraints but track the
→input time range,
>>> # which will span from the beginning of the record to 24 hours after admission.
>>> input_end_node = Node("input_end")
>>> input_end_node.endpoint_expr = TemporalWindowBounds(True, timedelta(days=1),
→True)
>>> input_end_node.constraints = {}
>>> input_end_node.parent = root
>>> input_start_node = Node("input_start")
>>> input_start_node.endpoint_expr = ToEventWindowBounds(True, "-_RECORD_START",
→True)
>>> input_start_node.constraints = {}
>>> input_start_node.parent = root
>>> #
>>> #### BRANCH 3 ####
>>> # For our last branch, we will validate that the patient has sufficient
→historical data, asserting
>>> # that they should have at least 1 event of any kind at least 1 year prior to
→the trigger event.
>>> # This will be expressed through two windows, one spanning back a year, and the
→other looking
>>> # prior to that year.
```

(continues on next page)

```
>>> pre_node_1yr = Node("pre_node_1yr")
>>> pre_node_1yr.endpoint_expr = TemporalWindowBounds(False, timedelta(days=-365),␣
↪False)
>>> pre_node_1yr.constraints = {}
>>> pre_node_1yr.parent = root
>>> pre_node_total = Node("pre_node_total")
>>> pre_node_total.endpoint_expr = ToEventWindowBounds(False, "-_RECORD_START",␣
↪False)
>>> pre_node_total.constraints = {"*": (1, None)}
>>> pre_node_total.parent = pre_node_1yr
>>> #
>>> #### PREDICATES_DF ####
>>> # We'll have the following patient data:
>>> #  - subject 1 will have an admission that won't count because they'll have a␣
↪covid diagnosis,
>>> #    then an admission that won't count because there will be no associated␣
↪discharge.
>>> #  - subject 2 will have an admission that won't count because they'll have too␣
↪little data before
>>> #    it, then a second admission that will count.
>>> #  - subject 3 will have an admission that will be too short.
>>> #
>>> predicates_df = pl.DataFrame({
...     "subject_id": [
...         1, 1, 1, 1, 1, # Pre-event, Admission, Covid, Discharge, Admission.
...         2, 2, 2, 2, 2, # Pre-event-too-close, Admission, Discharge, Admission,␣
↪Death & Discharge.
...         3, 3, 3,       # Pre-event, Admission, Death
...     ],
...     "timestamp": [
...         # Subject 1
...         datetime(year=1980, month=12, day=1,  hour=12, minute=3),  # Pre-event
...         datetime(year=1989, month=12, day=3,  hour=13, minute=14), # Admission
...         datetime(year=1989, month=12, day=5,  hour=15, minute=17), # Covid
...         datetime(year=1989, month=12, day=7,  hour=11, minute=4),  # Discharge
...         datetime(year=1989, month=12, day=23, hour=3,  minute=12), # Admission
...         # Subject 2
...         datetime(year=1983, month=12, day=1,  hour=22, minute=2),  # Pre-event-
↪too-close
...         datetime(year=1983, month=12, day=2,  hour=12, minute=3),  # Admission
...         datetime(year=1983, month=12, day=8,  hour=13, minute=14), # Discharge
...         datetime(year=1989, month=12, day=6,  hour=15, minute=17), # Valid␣
↪Admission
...         datetime(year=1989, month=12, day=10, hour=16, minute=22), # Death &␣
↪Discharge
...         # Subject 3
...         datetime(year=1982, month=2,  day=13, hour=10, minute=44), # Pre-event
...         datetime(year=1999, month=12, day=6,  hour=15, minute=17), # Admission
...         datetime(year=1999, month=12, day=6,  hour=16, minute=22), # Discharge
...     ],
...     "is_admission": [0, 1, 0, 0, 1,   0, 1, 0, 1, 0,   0, 1, 0],
...     "is_discharge": [0, 0, 0, 1, 0,   0, 0, 1, 0, 1,   0, 0, 1],
```

```
...         "is_death":     [0, 0, 0, 0, 0,    0, 0, 0, 0, 1,    0, 0, 0],
...         "is_covid_dx":  [0, 0, 1, 0, 0,    0, 0, 0, 0, 0,    0, 0, 0],
...         "_ANY_EVENT":   [1, 1, 1, 1, 1,    1, 1, 1, 1, 1,    1, 1, 1],
... })
>>> subtreee_anchor_realizations = (
...      predicates_df
...      .filter(pl.col("is_admission") > 0)
...      .rename({"timestamp": "subtree_anchor_timestamp"})
...      .select("subject_id", "subtree_anchor_timestamp")
... )
>>> print(subtreee_anchor_realizations)
shape: (5, 2)
+----+---------+
| subject_id | subtree_anchor_timestamp |
| ---        | ---                      |
| i64        | datetime[μs]             |
|----|---------|
| 1          | 1989-12-03 13:14:00      |
| 1          | 1989-12-23 03:12:00      |
| 2          | 1983-12-02 12:03:00      |
| 2          | 1989-12-06 15:17:00      |
| 3          | 1999-12-06 15:17:00      |
└----+---------+
>>> out = extract_subtree(root, subtreee_anchor_realizations, predicates_df,
→timedelta(0))
>>> out.select("subject_id", "subtree_anchor_timestamp")
shape: (1, 2)
+----+---------+
| subject_id | subtree_anchor_timestamp |
| ---        | ---                      |
| i64        | datetime[μs]             |
|----|---------|
| 2          | 1989-12-06 15:17:00      |
└----+---------+
>>> out.columns
['subject_id',
 'target_summary',
 'subtree_anchor_timestamp',
 'gap_summary',
 'input_end_summary',
 'input_start_summary',
 'pre_node_total_summary',
 'pre_node_1yr_summary']
>>> def print_window(name: str, do_drop_any_events: bool = True):
...      drop_cols = ["window_name", "subject_id", "subtree_anchor_timestamp"]
...      if do_drop_any_events:
...          drop_cols.append("_ANY_EVENT")
...      return (
...          out.select("subject_id", "subtree_anchor_timestamp", name)
...          .unnest(name)
...          .drop(*drop_cols)
...      )
```

```
>>> print_window("gap_summary")
shape: (1, 6)
+-------+-------+-----+-----+----+-----+
| timestamp_at_start  | timestamp_at_end   | is_admission | is_discharge | is_
↪death | is_covid_dx |
| ---                 | ---                | ---          | ---          | ---     ↵
↪ | ---         |
| datetime[µs]        | datetime[µs]       | i64          | i64          | i64     ↵
↪ | i64         |
|-------|-------|-----|-----|----|-----|
| 1989-12-06 15:17:00 | 1989-12-08 15:17:00 | 1           | 0            | 0       ↵
↪ | 0           |
└-------+-------+-----+-----+----+-----+
>>> print_window("target_summary")
shape: (1, 6)
+-------+-------+-----+-----+----+-----+
| timestamp_at_start  | timestamp_at_end   | is_admission | is_discharge | is_
↪death | is_covid_dx |
| ---                 | ---                | ---          | ---          | ---     ↵
↪ | ---         |
| datetime[µs]        | datetime[µs]       | i64          | i64          | i64     ↵
↪ | i64         |
|-------|-------|-----|-----|----|-----|
| 1989-12-08 15:17:00 | 1989-12-10 16:22:00 | 0           | 1            | 1       ↵
↪ | 0           |
└-------+-------+-----+-----+----+-----+
>>> print_window("input_start_summary")
shape: (1, 6)
+-------+-------+-----+-----+----+-----+
| timestamp_at_start  | timestamp_at_end   | is_admission | is_discharge | is_
↪death | is_covid_dx |
| ---                 | ---                | ---          | ---          | ---     ↵
↪ | ---         |
| datetime[µs]        | datetime[µs]       | i64          | i64          | i64     ↵
↪ | i64         |
|-------|-------|-----|-----|----|-----|
| 1983-12-01 22:02:00 | 1989-12-06 15:17:00 | 2           | 1            | 0       ↵
↪ | 0           |
└-------+-------+-----+-----+----+-----+
>>> print_window("input_end_summary")
shape: (1, 6)
+-------+-------+-----+-----+----+-----+
| timestamp_at_start  | timestamp_at_end   | is_admission | is_discharge | is_
↪death | is_covid_dx |
| ---                 | ---                | ---          | ---          | ---     ↵
↪ | ---         |
| datetime[µs]        | datetime[µs]       | i64          | i64          | i64     ↵
↪ | i64         |
|-------|-------|-----|-----|----|-----|
| 1989-12-06 15:17:00 | 1989-12-07 15:17:00 | 1           | 0            | 0       ↵
↪ | 0           |
└-------+-------+-----+-----+----+-----+
```

```
>>> print_window("pre_node_1yr_summary")
shape: (1, 6)
+-------+-------+-----+-----+----+-----+
| timestamp_at_start   | timestamp_at_end    | is_admission | is_discharge | is_
↪death | is_covid_dx |
| ---                  | ---                 | ---          | ---          | ---    ␣
↪ | ---         |
| datetime[µs]         | datetime[µs]        | i64          | i64          | i64    ␣
↪ | i64         |
|-------|-------|-----|-----|----|-----|
| 1989-12-06 15:17:00 | 1988-12-06 15:17:00 | 0            | 0            | 0      ␣
↪ | 0           |
└-------+-------+-----+-----+----+-----+
>>> print_window("pre_node_total_summary")
shape: (1, 6)
+-------+-------+-----+-----+----+-----+
| timestamp_at_start   | timestamp_at_end    | is_admission | is_discharge | is_
↪death | is_covid_dx |
| ---                  | ---                 | ---          | ---          | ---    ␣
↪ | ---         |
| datetime[µs]         | datetime[µs]        | i64          | i64          | i64    ␣
↪ | i64         |
|-------|-------|-----|-----|----|-----|
| 1983-12-01 22:02:00 | 1988-12-06 15:17:00 | 1            | 1            | 0      ␣
↪ | 0           |
└-------+-------+-----+-----+----+-----+
```

```
>>> root = Node("root")
>>> child = Node("child")
>>> child.endpoint_expr = (True, timedelta(days=3))
>>> child.constraints = {}
>>> child.parent = root
>>> predicates_df = pl.DataFrame({
...     "subject_id": [1],
...     "timestamp": [datetime(2020, 1, 1)]
... })
>>> subtree_anchor_realizations = pl.DataFrame({
...     "subject_id": [1],
...     "subtree_anchor_timestamp": [datetime(2020, 1, 1)]
... })
>>> print(child.endpoint_expr)
(True, datetime.timedelta(days=3))
>>> extract_subtree(root, subtree_anchor_realizations, predicates_df, timedelta(0))
shape: (1, 3)
+----+--------+-----------+
| subject_id | subtree_anchor_timestamp | child_summary                      |
| ---        | ---                      | ---                                |
| i64        | datetime[µs]             | struct[3]                          |
|----|--------|-----------|
| 1          | 2020-01-01 00:00:00      | {"child",2020-01-01 00:00:00,2... |
└----+--------+-----------+
>>> print(child.endpoint_expr)
```

```
(True, datetime.timedelta(days=3))
```

```
>>> child.endpoint_expr = (True, 42)
>>> extract_subtree(root, subtree_anchor_realizations, predicates_df, timedelta(0))
Traceback (most recent call last):
    ...
ValueError: Invalid endpoint expression: '(True, 42, datetime.timedelta(0))'
```

## aces.predicates module

This module contains functions for generating predicate columns for event sequences.

aces.predicates.**direct_load_plain_predicates**(data_path: *Path*, predicates: *list[str]*, ts_format: *str |
None*) → DataFrame

Loads a CSV file from disk and verifies that the necessary plain predicate columns are present.

**This CSV file must have the following columns:**

- subject_id: The subject identifier.
- timestamp: The timestamp of the event, in the format "MM/DD/YYYY HH:MM".
- Any additional columns specified in the set of desired plain predicates.

### Parameters

**data_path: Path**
The path to the CSV file.

**predicates: list[str]**
The list of columns to read from the CSV file.

### Returns
The Polars DataFrame containing the specified columns.

### Example

```
>>> import tempfile
>>> CSV_data = pl.DataFrame({
...     "subject_id": [1, 1, 1, 1, 2, 2],
...     "timestamp": [None, "01/01/2021 00:00", None, "01/01/2021 12:00", "01/02/
→2021 00:00", None],
...     "is_admission": [0, 1, 0, 0, 1, 0],
...     "is_discharge": [0, 0, 0, 1, 0, 0],
...     "is_male": [1, 0, 0, 0, 0, 0],
...     "is_female": [0, 0, 0, 0, 0, 1],
...     "brown_eyes": [0, 0, 1, 0, 0, 0],
... })
>>> with tempfile.NamedTemporaryFile(mode="w", suffix=".parquet") as f:
...     data_path = Path(f.name)
...     CSV_data.write_parquet(data_path)
...     direct_load_plain_predicates(data_path, ["is_admission", "is_discharge",
→"is_male",
```

```
...           "is_female", "brown_eyes"], "%m/%d/%Y %H:%M")
shape: (5, 7)
+----+-------+-----+-----+---+----+----+
| subject_id | timestamp          | is_admission | is_discharge | is_male | is_
↪female | brown_eyes |
| ---        | ---                | ---          | ---          | ---     | ---   ␣
↪    | ---        |
| i64        | datetime[μs]       | i64          | i64          | i64     | i64   ␣
↪    | i64        |
|----|-------|-----|-----|---|----|----|
| 1          | null               | 0            | 0            | 1       | 0     ␣
↪    | 1          |
| 1          | 2021-01-01 00:00:00 | 1           | 0            | 0       | 0     ␣
↪    | 0          |
| 1          | 2021-01-01 12:00:00 | 0           | 1            | 0       | 0     ␣
↪    | 0          |
| 2          | 2021-01-02 00:00:00 | 1           | 0            | 0       | 0     ␣
↪    | 0          |
| 2          | null               | 0            | 0            | 0       | 1     ␣
↪    | 0          |
└----+-------+-----+-----+---+----+----+
```

If the timestamp column is already a timestamp, then the `ts_format` argument id not needed, but can be used without an error.

```
>>> with tempfile.NamedTemporaryFile(mode="w", suffix=".parquet") as f:
...     data_path = Path(f.name)
...     (
...         CSV_data
...         .with_columns(pl.col("timestamp").str.strptime(pl.Datetime, format="%m/
↪%d/%Y %H:%M"))
...         .write_parquet(data_path)
...     )
...     direct_load_plain_predicates(data_path, ["is_admission", "is_discharge",
↪"is_male",
...         "is_female", "brown_eyes"], "%m/%d/%Y %H:%M")
shape: (5, 7)
+----+-------+-----+-----+---+----+----+
| subject_id | timestamp          | is_admission | is_discharge | is_male | is_
↪female | brown_eyes |
| ---        | ---                | ---          | ---          | ---     | ---   ␣
↪    | ---        |
| i64        | datetime[μs]       | i64          | i64          | i64     | i64   ␣
↪    | i64        |
|----|-------|-----|-----|---|----|----|
| 1          | null               | 0            | 0            | 1       | 0     ␣
↪    | 1          |
| 1          | 2021-01-01 00:00:00 | 1           | 0            | 0       | 0     ␣
↪    | 0          |
| 1          | 2021-01-01 12:00:00 | 0           | 1            | 0       | 0     ␣
↪    | 0          |
| 2          | 2021-01-02 00:00:00 | 1           | 0            | 0       | 0     ␣
```

```
→    | 0          |
| 2            | null                | 0            | 0            | 0        | 1        ␣
→    | 0          |
└────+───────+─────+─────+───+────+────+
>>> with tempfile.NamedTemporaryFile(mode="w", suffix=".parquet") as f:
...     data_path = Path(f.name)
...     (
...         CSV_data
...         .with_columns(pl.col("timestamp").str.strptime(pl.Datetime, format="%m/
→%d/%Y %H:%M"))
...         .write_parquet(data_path)
...     )
...     direct_load_plain_predicates(data_path, ["is_admission", "is_discharge",
→"is_male",
...         "is_female", "brown_eyes"], None)
shape: (5, 7)
+────+───────+─────+─────+───+────+────+
| subject_id | timestamp           | is_admission | is_discharge | is_male | is_
→female | brown_eyes |
| ---        | ---                 | ---          | ---          | ---     | ---      ␣
→    | ---        |
| i64        | datetime[µs]        | i64          | i64          | i64     | i64      ␣
→    | i64        |
|────|───────|─────|─────|───|────|────|
| 1          | null                | 0            | 0            | 1       | 0        ␣
→    | 1          |
| 1          | 2021-01-01 00:00:00 | 1            | 0            | 0       | 0        ␣
→    | 0          |
| 1          | 2021-01-01 12:00:00 | 0            | 1            | 0       | 0        ␣
→    | 0          |
| 2          | 2021-01-02 00:00:00 | 1            | 0            | 0       | 0        ␣
→    | 0          |
| 2          | null                | 0            | 0            | 0       | 1        ␣
→    | 0          |
└────+───────+─────+─────+───+────+────+
>>> with tempfile.NamedTemporaryFile(mode="w", suffix=".csv") as f:
...     data_path = Path(f.name)
...     CSV_data.write_csv(data_path)
...     direct_load_plain_predicates(data_path, ["is_admission", "is_discharge",
→"is_male",
...         "is_female", "brown_eyes"], "%m/%d/%Y %H:%M")
shape: (5, 7)
+────+───────+─────+─────+───+────+────+
| subject_id | timestamp           | is_admission | is_discharge | is_male | is_
→female | brown_eyes |
| ---        | ---                 | ---          | ---          | ---     | ---      ␣
→    | ---        |
| i64        | datetime[µs]        | i64          | i64          | i64     | i64      ␣
→    | i64        |
|────|───────|─────|─────|───|────|────|
| 1          | null                | 0            | 0            | 1       | 0        ␣
→    | 1          |
```

```
| 1            | 2021-01-01 00:00:00 | 1               | 0               | 0            | 0     ␣
↪   | 0          |
| 1            | 2021-01-01 12:00:00 | 0               | 1               | 0            | 0     ␣
↪   | 0          |
| 2            | 2021-01-02 00:00:00 | 1               | 0               | 0            | 0     ␣
↪   | 0          |
| 2            | null                | 0               | 0               | 0            | 1     ␣
↪   | 0          |
└────+───────+─────+─────+───+────+────+
>>> with tempfile.NamedTemporaryFile(mode="w", suffix=".csv") as f:
...     data_path = Path(f.name)
...     CSV_data.write_csv(data_path)
...     direct_load_plain_predicates(data_path, ["is_discharge", "brown_eyes"], "%m/
↪%d/%Y %H:%M")
shape: (5, 4)
+----+-------+-----+----+
| subject_id | timestamp           | is_discharge | brown_eyes |
| ---        | ---                 | ---          | ---        |
| i64        | datetime[μs]        | i64          | i64        |
|----|-------|-----|----|
| 1          | null                | 0            | 1          |
| 1          | 2021-01-01 00:00:00 | 0            | 0          |
| 1          | 2021-01-01 12:00:00 | 1            | 0          |
| 2          | 2021-01-02 00:00:00 | 0            | 0          |
| 2          | null                | 0            | 0          |
└────+───────+─────+────+
>>> with tempfile.NamedTemporaryFile(mode="w", suffix=".csv") as f:
...     data_path = Path(f.name)
...     CSV_data.write_csv(data_path)
...     direct_load_plain_predicates(data_path, ["is_foobar"], "%m/%d/%Y %H:%M")
Traceback (most recent call last):
    ...
polars.exceptions.ColumnNotFoundError: ['is_foobar']
>>> with tempfile.NamedTemporaryFile(mode="w", suffix=".foo") as f:
...     data_path = Path(f.name)
...     CSV_data.write_csv(data_path)
...     direct_load_plain_predicates(data_path, ["is_discharge"], "%m/%d/%Y %H:%M")
Traceback (most recent call last):
    ...
ValueError: Unsupported file format: .foo
>>> with tempfile.TemporaryDirectory() as d:
...     data_path = Path(d) / "data.csv"
...     assert not data_path.exists()
...     direct_load_plain_predicates(data_path, ["is_admission", "is_discharge"], "
↪%m/%d/%Y %H:%M")
Traceback (most recent call last):
    ...
FileNotFoundError: Direct predicates file ... does not exist!
>>> with tempfile.NamedTemporaryFile(mode="w", suffix=".parquet") as f:
...     data_path = Path(f.name)
...     CSV_data.write_parquet(data_path)
...     direct_load_plain_predicates(data_path, ["is_admission", "is_discharge"],␣
```

```
→None)
Traceback (most recent call last):
    ...
ValueError: Must provide a timestamp format for direct predicates with str␣
→timestamps.
>>> with tempfile.NamedTemporaryFile(mode="w", suffix=".parquet") as f:
...     data_path = Path(f.name)
...     (
...         CSV_data
...         .with_columns(
...             pl.col("timestamp").str.strptime(pl.Datetime, format="%m/%d/%Y %H:%M
→")
...             .dt.timestamp()
...         )
...         .write_parquet(data_path)
...     )
...     direct_load_plain_predicates(data_path, ["is_admission", "is_discharge"],␣
→None)
Traceback (most recent call last):
    ...
TypeError: Passed predicates have timestamps of invalid type Int64.
```

aces.predicates.**generate_plain_predicates_from_esgpt**(data_path: *Path*, predicates: *dict*) →
DataFrame

Generate plain predicate columns from an ESGPT dataset.

To learn more about the ESGPT format, please visit https://eventstreamml.readthedocs.io/en/latest/

> **Parameters**
>
>> **data_path: Path**
>> The path to the ESGPT dataset directory.
>>
>> **predicates: dict**
>> The dictionary of plain predicate configurations.
>
> **Returns**
>
>> The Polars DataFrame containing the extracted predicates per subject per timestamp across the
>> entire ESGPT dataset.
>>
>> ```
>> >>> import pytest
>> >>> import sys
>> >>> from unittest.mock import patch
>> >>> from pathlib import Path
>> >>> with patch.dict(sys.modules, {"EventStream.data.dataset_polars":␣
>> →None}):
>> ...     generate_plain_predicates_from_esgpt(Path("/fake/path"), {})
>> Traceback (most recent call last):
>>     ...
>> ImportError: The 'EventStream' package is required to load ESGPT␣
>> →datasets. If you mean to use a
>> MEDS dataset, please specify the 'MEDS' standard. Otherwise, please␣
>> →install the package from
>> ```

```
https://github.com/mmcdermott/EventStreamGPT and add the package to␣
→your PYTHONPATH.
```

aces.predicates.**generate_plain_predicates_from_meds**(data_path: *Path*, predicates: *dict*) → DataFrame

> Generate plain predicate columns from a MEDS dataset.
>
> To learn more about the MEDS format, please visit https://github.com/Medical-Event-Data-Standard/meds
>
> > **Parameters**
> >
> > > **data_path: Path**
> > > > The path to the MEDS dataset file.
> > >
> > > **predicates: dict**
> > > > The dictionary of plain predicate configurations.
> >
> > **Returns**
> > > The Polars DataFrame containing the extracted predicates per subject per timestamp across the entire MEDS dataset.

**Example**

```
>>> import tempfile
>>> from .config import PlainPredicateConfig
>>> parquet_data = pl.DataFrame({
...     "subject_id": [1, 1, 1, 2, 3],
...     "time": ["1/1/1989 00:00", "1/1/1989 01:00", "1/1/1989 01:00", "1/1/1989 02:
→00", None],
...     "code": ['admission', 'discharge', 'discharge', 'admission', "gender//male
→"],
... }).with_columns(pl.col("time").str.strptime(pl.Datetime, format="%m/%d/%Y %H:%M
→"))
>>> with tempfile.NamedTemporaryFile(mode="w", suffix=".parquet") as f:
...     data_path = Path(f.name)
...     parquet_data.write_parquet(data_path)
...     generate_plain_predicates_from_meds(
...         data_path,
...         {"discharge": PlainPredicateConfig("discharge"),
...             "male": PlainPredicateConfig("gender//male", static=True)}
...     )
shape: (4, 4)
+----+-------+----+--+
| subject_id | timestamp           | discharge | male |
| ---        | ---                 | ---       | ---  |
| i64        | datetime[μs]        | i64       | i64  |
|----|-------|----|--|
| 1          | 1989-01-01 00:00:00 | 0         | 0    |
| 1          | 1989-01-01 01:00:00 | 2         | 0    |
| 2          | 1989-01-01 02:00:00 | 0         | 0    |
| 3          | null                | 0         | 1    |
└----+-------+----+--+
```

aces.predicates.**get_predicates_df**(cfg: *TaskExtractorConfig*, data_config: *DictConfig*) → DataFrame

> Generate predicate columns based on the configuration.

> Parameters

>> **cfg:** *TaskExtractorConfig*
>>> The TaskExtractorConfig object containing the predicates information.

>> `data_path`
>>> Path to external data (file path to .csv or .parquet, or ESGPT directory) as string or Path.

>> `standard`
>>> The data standard, either 'CSV, 'MEDS' or 'ESGPT'.

> **Returns**
>> The Polars DataFrame with the added predicate columns.

> **Return type**
>> pl.DataFrame

> **Raises**
>> `ValueError` – If an invalid predicate type is specified in the configuration.

**Example**

```
>>> import tempfile
>>> from .config import PlainPredicateConfig, DerivedPredicateConfig, EventConfig,
→WindowConfig
>>> data = pl.DataFrame({
...     "subject_id": [1, 1, 1, 2, 2, 2],
...     "timestamp": [
...         None,
...         "01/01/2021 00:00",
...         "01/01/2021 12:00",
...         None,
...         "01/02/2021 00:00",
...         "01/02/2021 12:00"],
...     "adm":      [0, 1, 0, 0, 1, 0],
...     "dis":      [0, 0, 1, 0, 0, 0],
...     "death":    [0, 0, 0, 0, 0, 1],
...     "male":     [1, 0, 0, 0, 0, 0],
...     "female":   [0, 0, 0, 1, 0, 0],
... })
>>> predicates = {
...     "adm": PlainPredicateConfig("adm"),
...     "dis": PlainPredicateConfig("dis"),
...     "death": PlainPredicateConfig("death"),
...     "male": PlainPredicateConfig("male", static=True), # predicate match based
→on name for direct
...     "death_or_dis": DerivedPredicateConfig("or(death, dis)"),
... }
>>> trigger = EventConfig("adm")
>>> windows = {
...     "input": WindowConfig(
...         start=None,
...         end="trigger + 24h",
...         start_inclusive=True,
...         end_inclusive=True,
```

(continues on next page)

```
...                 has={"_ANY_EVENT": "(32, None)"},
...         ),
...         "gap": WindowConfig(
...             start="input.end",
...             end="start + 24h",
...             start_inclusive=False,
...             end_inclusive=True,
...             has={
...                 "death_or_dis": "(None, 0)",
...                 "adm": "(None, 0)",
...             },
...         ),
...         "target": WindowConfig(
...             start="gap.end",
...             end="start -> death_or_dis",
...             start_inclusive=False,
...             end_inclusive=True,
...             has={},
...         ),
... }
>>> config = TaskExtractorConfig(predicates=predicates, trigger=trigger,
↪windows=windows)
>>> with tempfile.NamedTemporaryFile(mode="w", suffix=".csv") as f:
...     data_path = Path(f.name)
...     data.write_csv(data_path)
...     data_config = DictConfig({
...         "path": str(data_path), "standard": "direct", "ts_format": "%m/%d/%Y %H:
↪%M"
...     })
...     get_predicates_df(config, data_config)
shape: (6, 8)
+----+-------+--+--+---+--+-----+----+
| subject_id | timestamp          | adm | dis | death | male | death_or_dis | _ANY_
↪EVENT |
| ---        | ---                | --- | --- | ---   | --- | ---          | ---
↪       |
| i64        | datetime[µs]       | i64 | i64 | i64   | i64 | i64          | i64
↪       |
|----|-------|--|--|---|--|-----|----|
| 1          | null               | 0   | 0   | 0     | 1   | 0            | null
↪       |
| 1          | 2021-01-01 00:00:00 | 1   | 0   | 0     | 0   | 0            | 1
↪       |
| 1          | 2021-01-01 12:00:00 | 0   | 1   | 0     | 0   | 1            | 1
↪       |
| 2          | null               | 0   | 0   | 0     | 0   | 0            | null
↪       |
| 2          | 2021-01-02 00:00:00 | 1   | 0   | 0     | 0   | 0            | 1
↪       |
| 2          | 2021-01-02 12:00:00 | 0   | 0   | 1     | 0   | 1            | 1
↪       |
└----+-------+--+--+---+--+-----+----+
```

```python
>>> with tempfile.NamedTemporaryFile(mode="w", suffix=".parquet") as f:
...     data_path = Path(f.name)
...     (
...         data
...         .with_columns(pl.col("timestamp").str.strptime(pl.Datetime, format="%m/
↪%d/%Y %H:%M"))
...         .write_parquet(data_path)
...     )
...     data_config = DictConfig({"path": str(data_path), "standard": "direct", "ts_
↪format": None})
...     get_predicates_df(config, data_config)
shape: (6, 8)
```

```
+----+-------+--+--+---+--+-----+----+
| subject_id | timestamp           | adm | dis | death | male | death_or_dis | _ANY_
↪EVENT |
| ---        | ---                 | --- | --- | ---   | --- | ---          | ---  ␣
↪      |
| i64        | datetime[μs]        | i64 | i64 | i64   | i64 | i64          | i64  ␣
↪      |
|----|-------|--|--|---|--|-----|----|
| 1          | null                | 0   | 0   | 0     | 1   | 0            | null␣
↪      |
| 1          | 2021-01-01 00:00:00 | 1   | 0   | 0     | 0   | 0            | 1    ␣
↪      |
| 1          | 2021-01-01 12:00:00 | 0   | 1   | 0     | 0   | 1            | 1    ␣
↪      |
| 2          | null                | 0   | 0   | 0     | 0   | 0            | null␣
↪      |
| 2          | 2021-01-02 00:00:00 | 1   | 0   | 0     | 0   | 0            | 1    ␣
↪      |
| 2          | 2021-01-02 12:00:00 | 0   | 0   | 1     | 0   | 1            | 1    ␣
↪      |
└----+-------+--+--+---+--+-----+----+
```

```python
>>> any_event_trigger = EventConfig("_ANY_EVENT")
>>> adm_only_predicates = {"adm": PlainPredicateConfig("adm"), "male":␣
↪PlainPredicateConfig("male")}
>>> st_end_windows = {
...     "input": WindowConfig(
...         start="end - 365d",
...         end="trigger + 24h",
...         start_inclusive=True,
...         end_inclusive=True,
...         has={
...             "_RECORD_END": "(None, 0)",   # These are added just to show start/
↪end predicates
...             "_RECORD_START": "(None, 0)", # These are added just to show start/
↪end predicates
...         },
...     ),
... }
>>> st_end_config = TaskExtractorConfig(
...     predicates=adm_only_predicates, trigger=any_event_trigger, windows=st_end_
```

```
↪windows
... )
>>> with tempfile.NamedTemporaryFile(mode="w", suffix=".csv") as f:
...     data_path = Path(f.name)
...     data.write_csv(data_path)
...     data_config = DictConfig({
...         "path": str(data_path), "standard": "direct", "ts_format": "%m/%d/%Y %H:
↪%M"
...     })
...     get_predicates_df(st_end_config, data_config)
shape: (6, 7)
┌──────┬─────────┬────┬────┬─────┬─────┬─────┐
│ subject_id │ timestamp           │ adm │ male │ _ANY_EVENT │ _RECORD_START │ _
↪RECORD_END │
│ ---        │ ---                 │ --- │ ---  │ ---        │ ---           │ --- ␣
↪          │
│ i64        │ datetime[μs]        │ i64 │ i64  │ i64        │ i64           │ i64 ␣
↪          │
╞══════╪═════════╪════╪════╪═════╪═════╪═════╡
│ 1          │ null                │ 0   │ 1    │ null       │ null          │ null␣
↪          │
│ 1          │ 2021-01-01 00:00:00 │ 1   │ 0    │ 1          │ 1             │ 0   ␣
↪          │
│ 1          │ 2021-01-01 12:00:00 │ 0   │ 0    │ 1          │ 0             │ 1   ␣
↪          │
│ 2          │ null                │ 0   │ 0    │ null       │ null          │ null␣
↪          │
│ 2          │ 2021-01-02 00:00:00 │ 1   │ 0    │ 1          │ 1             │ 0   ␣
↪          │
│ 2          │ 2021-01-02 12:00:00 │ 0   │ 0    │ 1          │ 0             │ 1   ␣
↪          │
└──────┴─────────┴────┴────┴─────┴─────┴─────┘
```

```
>>> data = pl.DataFrame({
...     "subject_id": [1, 1, 1, 2, 2],
...     "timestamp": [
...         None,
...         "01/01/2021 00:00",
...         "01/01/2021 12:00",
...         "01/02/2021 00:00",
...         "01/02/2021 12:00"],
...     "adm":     [0, 1, 0, 1, 0],
...     "male":    [1, 0, 0, 0, 0],
... })
>>> predicates = {
...     "adm": PlainPredicateConfig("adm"),
...     "male": PlainPredicateConfig("male", static=True), # predicate match based␣
↪on name for direct
...     "male_adm": DerivedPredicateConfig("and(male, adm)", static=['male']),
... }
>>> trigger = EventConfig("adm")
>>> windows = {
```

```
...         "input": WindowConfig(
...             start=None,
...             end="trigger + 24h",
...             start_inclusive=True,
...             end_inclusive=True,
...             has={"_ANY_EVENT": "(32, None)"},
...         ),
...         "gap": WindowConfig(
...             start="input.end",
...             end="start + 24h",
...             start_inclusive=False,
...             end_inclusive=True,
...             has={
...                 "adm": "(None, 0)",
...                 "male_adm": "(None, 0)",
...             },
...         ),
... }
>>> config = TaskExtractorConfig(predicates=predicates, trigger=trigger,
↪windows=windows)
>>> with tempfile.NamedTemporaryFile(mode="w", suffix=".csv") as f:
...     data_path = Path(f.name)
...     data.write_csv(data_path)
...     data_config = DictConfig({
...         "path": str(data_path), "standard": "direct", "ts_format": "%m/%d/%Y %H:
↪%M"
...     })
...     get_predicates_df(config, data_config)
shape: (5, 6)
+----+-------+--+--+----+----+
| subject_id | timestamp           | adm | male | male_adm | _ANY_EVENT |
| ---        | ---                 | --- | ---  | ---      | ---        |
| i64        | datetime[μs]        | i64 | i64  | i64      | i64        |
|----|-------|--|--|----|----|
| 1          | null                | 0   | 1    | 0        | null       |
| 1          | 2021-01-01 00:00:00 | 1   | 1    | 1        | 1          |
| 1          | 2021-01-01 12:00:00 | 0   | 1    | 0        | 1          |
| 2          | 2021-01-02 00:00:00 | 1   | 0    | 0        | 1          |
| 2          | 2021-01-02 12:00:00 | 0   | 0    | 0        | 1          |
└----+-------+--+--+----+----+
```

```
>>> with tempfile.NamedTemporaryFile(mode="w", suffix=".csv") as f:
...     data_path = Path(f.name)
...     data.write_csv(data_path)
...     data_config = DictConfig({
...         "path": str(data_path), "standard": "buzz", "ts_format": "%m/%d/%Y %H:%M
↪"
...     })
...     get_predicates_df(config, data_config)
Traceback (most recent call last):
    ...
ValueError: Invalid data standard: buzz. Options are 'direct', 'MEDS', 'ESGPT'.
```

aces.predicates.**process_esgpt_data**(subjects_df: *DataFrame*, events_df: *DataFrame*,
dynamic_measurements_df: *DataFrame*, value_columns: *dict[str, str]*,
predicates: *dict*) → DataFrame

> Process ESGPT data to generate plain predicate columns.

> > **Parameters**

> > > **events_df: DataFrame**
> > > > The Polars DataFrame containing the events data.

> > > **dynamic_measurements_df: DataFrame**
> > > > The Polars DataFrame containing the dynamic measurements data.

> > **Returns**
> > > The Polars DataFrame containing the extracted predicates per subject per timestamp across the
> > > entire ESGPT dataset.

**Examples**

```
>>> from datetime import datetime
>>> from .config import PlainPredicateConfig
>>> subjects_df = pl.DataFrame({
...     "subject_id": [1, 2],
...     "MRN": ["A123", "B456"],
...     "eye_colour": ["brown", "blue"],
...     "dob": [datetime(1980, 1, 1), datetime(1990, 1, 1)],
... })
>>> events_df = pl.DataFrame({
...     "event_id": [1, 2, 3, 4],
...     "subject_id": [1, 1, 2, 2],
...     "timestamp": [
...             datetime(2021, 1, 1, 0, 0),
...             datetime(2021, 1, 1, 12, 0),
...             datetime(2021, 1, 2, 0, 0),
...             datetime(2021, 1, 2, 12, 0),
...     ],
...     "event_type": ["adm", "dis", "adm", "obs"],
...     "age": [30, 30, 40, 40],
... })
>>> dynamic_measurements_df = pl.DataFrame({
...     "event_id": [1,     1,    1,    2,    2,    2,    3,     4,     5],
...     "adm_loc":  ["foo", None, None, None, None, None, "bar", None, None],
...     "dis_loc":  [None,  None, None, None, None, "H",  None,  None, None],
...     "HR":       [None,  150,  None, 120,  None, None, None,  177,  89],
...     "lab":      [None,  None, "K",  None, "K",  None, None,  None, "SpO2"],
...     "lab_val":  [None,  None, 5.1,  None, 3.8,  None, None,  None, 99],
... })
>>> value_columns = {
...     "is_admission": None,
...     "is_discharge": None,
...     "high_HR": "HR",
...     "high_Potassium": "lab_val",
... }
>>> predicates = {
```

(continues on next page)

```
...     "is_adm": PlainPredicateConfig(code="event_type//adm"),
...     "is_dis": PlainPredicateConfig(code="event_type//dis"),
...     "high_HR": PlainPredicateConfig(code="HR", value_min=140),
...     "high_Potassium": PlainPredicateConfig(code="lab//K", value_min=5.0),
...     "eye_colour": PlainPredicateConfig(code="eye_colour//brown", static=True),
... }
>>> process_esgpt_data(subjects_df, events_df, dynamic_measurements_df, value_
↪columns, predicates)
shape: (6, 7)
+----+-------+---+---+---+------+----+
| subject_id | timestamp           | is_adm | is_dis | high_HR | high_Potassium |␣
↪eye_colour |
| ---        | ---                 | ---    | ---    | ---     | ---            | --
↪-         |
| i64        | datetime[μs]        | i64    | i64    | i64     | i64            |␣
↪i64         |
|----|-------|---|---|---|------|----|
| 1          | null                | 0      | 0      | 0       | 0              | 1␣
↪           |
| 2          | null                | 0      | 0      | 0       | 0              | 0␣
↪           |
| 1          | 2021-01-01 00:00:00 | 1      | 0      | 1       | 1              | 0␣
↪           |
| 1          | 2021-01-01 12:00:00 | 0      | 1      | 0       | 0              | 0␣
↪           |
| 2          | 2021-01-02 00:00:00 | 1      | 0      | 0       | 0              | 0␣
↪           |
| 2          | 2021-01-02 12:00:00 | 0      | 0      | 1       | 0              | 0␣
↪           |
└----+-------+---+---+---+------+----+
```

### aces.query module

This module contains the main function for querying a task.

It accepts the configuration file and predicate columns, builds the tree, and recursively queries the tree.

aces.query.**query**(*cfg:* TaskExtractorConfig, *predicates_df: DataFrame*) → DataFrame

Query a task using the provided configuration file and predicates dataframe.

> **Parameters**
>
> > **cfg:** *TaskExtractorConfig*
> > TaskExtractorConfig object of the configuration file.
> >
> > **predicates_df: DataFrame**
> > Polars predicates dataframe.
>
> **Returns**
>
> > **The result of the task query, containing subjects who satisfy the conditions**
> > defined in cfg. Timestamps for the start/end boundaries of each window specified in the task
> > configuration, as well as predicate counts for each window, are provided.

> **Return type**
>> polars.DataFrame

> **Raises**
>> - **TypeError** – If predicates_df is not a polars.DataFrame.
>>
>> - **ValueError** – If the (subject_id, timestamp) columns are not unique.

Examples: These examples are limited for now; see the `tests` directory for full examples.

```
>>> import logging
>>> from io import StringIO
>>> log_stream = StringIO()
>>> logger.addHandler(logging.StreamHandler(log_stream))
>>> logger.setLevel(logging.INFO)
>>> from datetime import datetime
>>> from .config import PlainPredicateConfig, WindowConfig, EventConfig
```

```
>>> cfg = None # This is obviously invalid, but we're just testing the error case.
>>> predicates_df = {"subject_id": [1, 1], "timestamp": [1, 1]}
>>> query(cfg, predicates_df)
Traceback (most recent call last):
    ...
TypeError: Predicates dataframe type must be a polars.DataFrame. Got: <class 'dict'>
↪.
>>> query(cfg, pl.DataFrame(predicates_df))
Traceback (most recent call last):
    ...
ValueError: The (subject_id, timestamp) columns must be unique.
>>> cfg = TaskExtractorConfig(
...     predicates={"A": PlainPredicateConfig("A")},
...     trigger=EventConfig("_ANY_EVENT"),
...     windows={
...         "pre": WindowConfig(None, "trigger", True, False, index_timestamp="start
↪"),
...         "post": WindowConfig("pre.end", None, True, True, label="A"),
...     },
...     index_timestamp_window="pre",
...     label_window="post",
... )
>>> predicates_df = pl.DataFrame({
...     "subject_id": [1, 1, 3],
...     "timestamp": [datetime(1980, 12, 28), datetime(2010, 6, 20), datetime(2010,
↪5, 11)],
...     "A": [False, False, False],
...     "_ANY_EVENT": [True, True, True],
... })
>>> result = query(cfg, predicates_df)
>>> result.select("subject_id", "trigger")
shape: (3, 2)
+----+-------+
| subject_id | trigger           |
| ---        | ---               |
| i64        | datetime[μs]      |
```

---

```
|----|-------|
| 1          | 1980-12-28 00:00:00 |
| 1          | 2010-06-20 00:00:00 |
| 3          | 2010-05-11 00:00:00 |
└----+-------+
>>> "index_timestamp" in result.columns
True
>>> "label" in result.columns
True
>>> cfg = TaskExtractorConfig(
...     predicates={"A": PlainPredicateConfig("A", static=True)},
...     trigger=EventConfig("_ANY_EVENT"),
...     windows={},
... )
>>> query(cfg, predicates_df)
shape: (0, 0)
++
||
└+
>>> log_output = log_stream.getvalue()
>>> "Static variable criteria specified, filtering patient demographics..." in log_
↪output
True
>>> "No static variable criteria specified, removing all rows with null timestamps..
↪." in log_output
True
>>> predicates_df = pl.DataFrame({
...     "subject_id": [1, 1, 3],
...     "timestamp": [None, datetime(2010, 6, 20), datetime(2010, 5, 11)],
...     "A": [True, False, False],
...     "_ANY_EVENT": [False, False, False],
... })
>>> result = query(cfg, predicates_df)
>>> log_output = log_stream.getvalue()
>>> "No valid rows found for the trigger event" in log_output
True
```

## aces.run module

Main script for end-to-end cohort extraction.

aces.run.**get_and_validate_label_schema**(df: *DataFrame*) → Table

> Validates the schema of a MEDS data DataFrame.
>
> This function validates the schema of a MEDS label DataFrame, ensuring that it has the correct columns and that the columns are of the correct type. This function will:
>
> 1. Re-type any of the mandator MEDS column to the appropriate type.
>
> 2. Attempt to add the numeric_value or time columns if either are missing, and set it to None. It will not attempt to add any other missing columns even if do_retype is True as the other columns cannot be set to None.

> **Parameters**
>> **df: DataFrame**
>>> The MEDS label DataFrame to validate.
>
> **Returns**
>> The validated MEDS data DataFrame, with columns re-typed as needed.
>
> **Return type**
>> pa.Table
>
> **Raises**
>> **ValueError** – if do_retype is False and the MEDS data DataFrame is not schema compliant.

### Examples

```
>>> df = pl.DataFrame({})
>>> get_and_validate_label_schema(df)
Traceback (most recent call last):
    ...
ValueError: MEDS Data DataFrame must have a 'subject_id' column of type Int64.
>>> from datetime import datetime
>>> df = pl.DataFrame({
...     subject_id_field: pl.Series([1, 3, 2], dtype=pl.UInt32),
...     "time": [datetime(2021, 1, 1), datetime(2021, 1, 2), datetime(2021, 1, 3)],
...     "boolean_value": [1, 0, 100],
... })
>>> get_and_validate_label_schema(df)
pyarrow.Table
subject_id: int64
prediction_time: timestamp[us]
boolean_value: bool
integer_value: int64
float_value: double
categorical_value: string
----
subject_id: [[1,3,2]]
prediction_time: [[null,null,null]]
boolean_value: [[true,false,true]]
integer_value: [[null,null,null]]
float_value: [[null,null,null]]
categorical_value: [[null,null,null]]
```

aces.run.**main**(*cfg: DictConfig*) → None

## aces.types module

This module contains types defined by this package.

These are all simple types using named tuples so can be safely ignored by downstream users provided data fields are passed in the correct order.

**class** aces.types.**TemporalWindowBounds**(left_inclusive*: bool*, window_size*: timedelta*, right_inclusive*: bool*, offset*: timedelta | None* = **None**)

> Bases: `object`
>
> Named tuple to represent temporal window bounds.
>
> **left_inclusive**
> > The start of the window, inclusive.
> >
> > > **Type**
> > > > bool
>
> **window_size**
> > The size of the window.
> >
> > > **Type**
> > > > datetime.timedelta
>
> **right_inclusive**
> > The end of the window, inclusive.
> >
> > > **Type**
> > > > bool
>
> **offset**
> > The offset from the start of the window to the end of the window.
> >
> > > **Type**
> > > > datetime.timedelta | None
>
> **Example**
>
> ```
> >>> bounds = TemporalWindowBounds(
> ...     left_inclusive=True,
> ...     window_size=timedelta(days=1),
> ...     right_inclusive=False,
> ...     offset=timedelta(hours=1)
> ... )
> >>> bounds
> TemporalWindowBounds(left_inclusive=True,
>                      window_size=datetime.timedelta(days=1),
>                      right_inclusive=False,
>                      offset=datetime.timedelta(seconds=3600))
> >>> left_inclusive, window_size, right_inclusive, offset = bounds
> >>> bounds.left_inclusive
> True
> >>> window_size
> datetime.timedelta(days=1)
> >>> right_inclusive
> ```

```
False
>>> offset
datetime.timedelta(seconds=3600)
```

**left_inclusive** : bool

**offset** : timedelta | None = **None**

**property polars_gp_rolling_kwargs** : dict[str, str | datetime.timedelta]

Return the parameters for a group_by rolling operation in Polars.

### Examples

```
>>> TemporalWindowBounds(
...     left_inclusive=True,
...     window_size=timedelta(days=1),
...     right_inclusive=True,
...     offset=None
... ).polars_gp_rolling_kwargs
{'period': datetime.timedelta(days=1),
 'offset': datetime.timedelta(0),
 'closed': 'both'}
>>> TemporalWindowBounds(
...     left_inclusive=True,
...     window_size=timedelta(days=1),
...     right_inclusive=True,
...     offset=timedelta(hours=1)
... ).polars_gp_rolling_kwargs
{'period': datetime.timedelta(days=1),
 'offset': datetime.timedelta(seconds=3600),
 'closed': 'both'}
>>> TemporalWindowBounds(
...     left_inclusive=False,
...     window_size=timedelta(days=2),
...     right_inclusive=False,
...     offset=timedelta(minutes=1)
... ).polars_gp_rolling_kwargs
{'period': datetime.timedelta(days=2),
 'offset': datetime.timedelta(seconds=60),
 'closed': 'none'}
>>> TemporalWindowBounds(
...     left_inclusive=True,
...     window_size=timedelta(days=2),
...     right_inclusive=False,
...     offset=timedelta(minutes=1)
... ).polars_gp_rolling_kwargs
{'period': datetime.timedelta(days=2),
 'offset': datetime.timedelta(seconds=60),
 'closed': 'left'}
>>> TemporalWindowBounds(
...     left_inclusive=False,
```

```
...     window_size=timedelta(days=2),
...     right_inclusive=True,
...     offset=timedelta(minutes=1)
... ).polars_gp_rolling_kwargs
{'period': datetime.timedelta(days=2),
 'offset': datetime.timedelta(seconds=60),
 'closed': 'right'}
```

**right_inclusive** : bool

**window_size** : timedelta

**class** aces.types.**ToEventWindowBounds**(left_inclusive: *bool*, end_event: *str*, right_inclusive: *bool*, offset: *timedelta | None* = **None**)

Bases: object

Named tuple to represent temporal window bounds.

**left_inclusive**

The start of the window, inclusive.

> **Type**
> bool

**end_event**

The string name of the event that bounds the end of this window. Operationally, this is interpreted as the string name of the column which contains a positive value if the row corresponds to the end event of this window and a zero otherwise.

> **Type**
> str

**right_inclusive**

The end of the window, inclusive.

> **Type**
> bool

**offset**

The offset from the start of the window to the end of the window.

> **Type**
> datetime.timedelta | None

> **Raises**
> - **ValueError** – If *end_event* is an empty string.
> - **ValueError** – If *offset* is negative.

**Example**

```
>>> bounds = ToEventWindowBounds(
...     left_inclusive=True,
...     end_event="foo",
...     right_inclusive=False,
...     offset=timedelta(hours=1)
... )
>>> bounds
ToEventWindowBounds(left_inclusive=True,
                    end_event='foo',
                    right_inclusive=False,
                    offset=datetime.timedelta(seconds=3600))
>>> left_inclusive, end_event, right_inclusive, offset = bounds
>>> left_inclusive
True
>>> end_event
'foo'
>>> right_inclusive
False
>>> offset
datetime.timedelta(seconds=3600)
>>> bounds = ToEventWindowBounds(
...     left_inclusive=True,
...     end_event="",
...     right_inclusive=False,
...     offset=timedelta(hours=1)
... )
Traceback (most recent call last):
    ...
ValueError: The 'end_event' must be a non-empty string.
>>> bounds = ToEventWindowBounds(
...     left_inclusive=True,
...     end_event="_RECORD_START",
...     right_inclusive=False,
...     offset=timedelta(hours=1)
... )
Traceback (most recent call last):
    ...
ValueError: It doesn't make sense to have the start of the record _RECORD_START be
↪an end event. Did
you mean to make that be the start event (which should result in the `end_event`
↪parameter being
'-_RECORD_START')?
>>> bounds = ToEventWindowBounds(
...     left_inclusive=True,
...     end_event="-_RECORD_END",
...     right_inclusive=False,
...     offset=timedelta(hours=1)
... )
Traceback (most recent call last):
    ...
ValueError: It doesn't make sense to have the end of the record _RECORD_END be a
```

```
→start event. Did
you mean to make that be the end event (which should result in the `end_event`␣
→parameter being
'_RECORD_END')?
```

**property** `boolean_expr_bound_sum_kwargs` : dict[str, str | datetime.timedelta | polars.expr.expr.Expr]

    Return the parameters for a group_by rolling operation in Polars.

### Examples

```
>>> def print_kwargs(kwargs: dict):
...     for key, value in kwargs.items():
...         print(f"{key}: {value}")
>>> print_kwargs(ToEventWindowBounds(
...     left_inclusive=True, end_event="is_A", right_inclusive=False,␣
→offset=None
... ).boolean_expr_bound_sum_kwargs)
boundary_expr: [(col("is_A")) > (dyn int: 0)]
mode: row_to_bound
closed: left
offset: 0:00:00
>>> print_kwargs(ToEventWindowBounds(
...     left_inclusive=False, end_event="-is_B", right_inclusive=True,␣
→offset=None
... ).boolean_expr_bound_sum_kwargs)
boundary_expr: [(col("is_B")) > (dyn int: 0)]
mode: bound_to_row
closed: right
offset: 0:00:00
>>> print_kwargs(ToEventWindowBounds(
...     left_inclusive=False, end_event="is_B", right_inclusive=False,␣
→offset=timedelta(hours=-3)
... ).boolean_expr_bound_sum_kwargs)
boundary_expr: [(col("is_B")) > (dyn int: 0)]
mode: row_to_bound
closed: none
offset: -1 day, 21:00:00
>>> print_kwargs(ToEventWindowBounds(
...     left_inclusive=True,
...     end_event="-_RECORD_START",
...     right_inclusive=True,
...     offset=timedelta(days=2),
... ).boolean_expr_bound_sum_kwargs)
boundary_expr: [(col("timestamp")) == (col("timestamp").min().over([col(
→"subject_id")]))]
mode: bound_to_row
closed: both
offset: 2 days, 0:00:00
>>> print_kwargs(ToEventWindowBounds(
...     left_inclusive=False,
...     end_event="_RECORD_END",
```

```
...        right_inclusive=True,
...        offset=timedelta(days=1),
... ).boolean_expr_bound_sum_kwargs)
boundary_expr: [(col("timestamp")) == (col("timestamp").max().over([col(
↪"subject_id")]))]
mode: row_to_bound
closed: right
offset: 1 day, 0:00:00
```

**end_event** : str

**left_inclusive** : bool

**offset** : timedelta | None = **None**

**right_inclusive** : bool

## aces.utils module

aces.utils.**capture_output**() → Generator[StringIO, None, None]

A context manager to capture stdout output.

This can eventually be eliminated if https://github.com/kayjan/bigtree/issues/285 is resolved.

### Examples

```
>>> with capture_output() as captured:
...     print("Hello, world!")
>>> captured.getvalue().strip()
'Hello, world!'
```

aces.utils.**log_tree**(node: *Node*) → None

Logs the tree structure using logging.info.

aces.utils.**parse_timedelta**(time_str: *str | None = **None***) → timedelta

Parse a time string and return a timedelta object.

Using time expression parser: https://github.com/wroberts/pytimeparse

> **Parameters**
>
> > **time_str: str | None = None**
> > The time string to parse.
>
> **Returns**
> The parsed timedelta object.
>
> **Return type**
> datetime.timedelta

**Examples**

```
>>> parse_timedelta("1 days")
datetime.timedelta(days=1)
>>> parse_timedelta("1 day")
datetime.timedelta(days=1)
>>> parse_timedelta("1 days 2 hours 3 minutes 4 seconds")
datetime.timedelta(days=1, seconds=7384)
>>> parse_timedelta('1 day, 14:20:16')
datetime.timedelta(days=1, seconds=51616)
>>> parse_timedelta('365 days')
datetime.timedelta(days=365)
>>> parse_timedelta()
datetime.timedelta(0)
>>> parse_timedelta("")
datetime.timedelta(0)
>>> parse_timedelta(None)
datetime.timedelta(0)
```

### 8.1.3 Module contents

This is the main module of the ACES package.

It contains the main functions and classes needed to extract cohorts.

# LICENSE

# WHY ACES?

If you have a dataset and want to leverage it for machine learning tasks, the ACES ecosystem offers a streamlined and user-friendly approach. Here's how you can easily transform, prepare, and utilize your dataset with MEDS and ACES for efficient and effective machine learning:

## 10.1 I. Transform to MEDS

- Simplicity: Converting your dataset to the Medical Event Data Standard (MEDS) is straightforward and user-friendly compared to other Common Data Models (CDMs).

- Minimal Bias: This conversion process ensures that your data remains as close to its raw form as possible, minimizing the introduction of biases.

- MEDS-ETL: Follow this link for detailed instructions and ETLs to transform your dataset into the MEDS format!

## 10.2 II. Identify Predicates

- Task-Specific Concepts: Identify the predicates (data concepts) required for your specific machine learning tasks.

- Pre-Defined Criteria: Utilize our pre-defined criteria across various tasks and clinical areas to expedite this process.

- MEDS-DEV: Access our benchmark of tasks to find relevant predicates!

## 10.3 III. Set Dataset-Agnostic Criteria

- Standardization: Combine the identified predicates with standardized, dataset-agnostic criteria files.

- Examples: Refer to the MEDS-DEV examples for guidance on how to structure your criteria files for your private datasets!

## 10.4  IV. Run ACES

- Run the ACES Command-Line Interface tool (`aces-cli`) to extract cohorts based on your task - check out the Usage Guide for more information!

## 10.5  V. Run MEDS-Tab

- Painless Reproducibility: Use MEDS-Tab to obtain comparable, reproducible, and well-tuned XGBoost results tailored to your dataset-specific feature space!

By following these steps, you can seamlessly transform your dataset, define necessary criteria, and leverage powerful machine learning tools within the ACES and MEDS ecosystem. This approach not only simplifies the process but also ensures high-quality, reproducible results for your machine learning for health projects. It can reliably take no more than a week of full-time human effort to perform Steps I-V on new datasets in reasonable raw formulations!

# PYTHON MODULE INDEX

## a

# W