# OpenReview forum: "ACES: Automatic Cohort Extraction System for Event-Stream Datasets"
_ICLR.cc/2025/Conference — ICLR 2025 Poster_

### Official Review · Reviewer_z6xZ · 2024-10-28

**Soundness:** 2
**Presentation:** 2
**Contribution:** 1
**Rating:** 3
**Confidence:** 4

**Summary:**

This paper concentrates on enhancing the reproducibility of machine learning for healthcare research and introduces the Automatic Cohort Extraction System (ACES), a library/tool specifically developed for handling healthcare data formatted as event streams. ACES functions as a configuration language that simplifies task definitions by abstracting them into two main components: dataset-specific event predicates and dataset-agnostic inclusion/exclusion criteria. This structured approach allows ACES to support a diverse array of representative predictive tasks essential for healthcare analytics.

**Strengths:**

S1. The promotion of standardization and simplification in automatic cohort extraction is of considerable importance. This effort can enhance the reproducibility of machine learning applications in healthcare research. Furthermore, it has the potential to advance the digitalization of healthcare data, thereby providing substantial benefits to both clinicians and patients.

S2. The paper provides a relatively thorough overview of the current landscape of machine learning within healthcare research, including common data models, existing benchmarks, and representative applications/tasks.

S3. The ACES system is implemented as an open-source project, accompanied by comprehensive documentation.

**Weaknesses:**

W1. ACES primarily functions in healthcare analytics by performing preprocessing on broadly collected EHR data tailored to specific target applications, such as extracting particular features within a time window or performing aggregations. This process is ubiquitous in most EHR-related data mining tasks. ACES abstracts common dataframe transformation patterns across diverse healthcare applications, providing high-level encapsulation. Under the assumptions set by ACES, this approach simplifies the preprocessing to some extent. However, it is essential to note that this work can be seen as largely an engineering solution rooted in low-code programming concepts, with limited conceptual innovation. Additionally, it is challenging to discern any substantial transformative impact this encapsulation introduces to EHR data processing.

W2. Continuing from the previous point, ACES aims to implement a standard data extraction process for healthcare data in an event-stream format. However, the paper does not adequately articulate the specific challenges inherent to this process. This omission leaves readers uncertain of the unique problems ACES addresses, thereby diminishing the perceived contributions and impact of ACES.

W3. The paper lacks objective and quantitative evidence to substantiate the superiority of ACES over existing work aimed at solving the same research problem. This hence weakens the argument for ACES as a preferable solution in the competitive landscape of healthcare data processing and management technologies.

**Questions:**

Please refer to points W1 through W3 for detailed concerns and recommendations.

---

> ### Author Response · Authors · 2024-11-23
> **Individual Response to Reviewer z6xZ - Thank you!**
>
> Thank you for your service in reviewing our paper and for recognizing the importance of standardized cohort extraction! We respond to each of your concerns/questions below:
>
> ### W1: Innovation
>
> Thank you for your feedback!
>
> ***"...largely an engineering solution...limited conceptual innovation...***
>
>
> Per the ICLR 2025 Call for Papers ([iclr.cc](https://iclr.cc/Conferences/2025/CallForPapers)), we understand that engineering solutions, and in particular "infrastructure, software libraries, hardware, etc." are in-scope. As such, we aimed to deliver a high-impact software tool that is much needed in the machine learning for health community.
>
> While we acknowledge that software contributions differ from traditional ML innovation, ACES does introduce a novel, non-standard recursive framework that leverages a hierarchical, tree-based structure to extract temporally and contextually defined data windows. Unlike traditional flat cohort extraction methods, ACES processes a configuration file as a constraint tree. The algorithm's innovation lies in its ability to dynamically aggregate predicate counts and validate constraints at each recursive step, using a combination of temporal rolling operations and cumulative sum-based event-bound computations. In essence, ACES is **defining a new domain-specific language (DSL)** specifically for cohort extraction that is **simple, transparent, and highly expressive**.
>
> In our updated paper, we've now specifically highlighted these innovations in the algorithm overview section (outlined in red) and also added a new overview figure. We hope you can re-evaluate our contribution based on this!
>
>
>
> ***"...challenging to discern any substantial transformative impact..."***
>
> Regarding impact, we believe ACES has significant potential to shape EHR data processing and ML benchmarking. The benchmarking effort referenced in Section 4.3 and the Appendix (MEDS-DEV, not linked for anonymity) is a prime example of this, **involving collaborative efforts by many researchers across leading institutions worldwide**. This initiative uses ACES configuration files to standardize cohort definitions for reproducible ML pipelines. Further, we are committed to building a community around ACES to introduce new features, expand predicate definitions, and enhance the configuration language for greater flexibility. Along with other novel open-source tools that make use of and depend on ACES, we believe ACES can not only complement existing tools, but also be preferred in the reproducibility context.

---

> ### Author Response · Authors · 2024-11-23
>
> ### W2: Cohort Extraction Challenges
>
> ***"...does not adequately articulate the specific challenges inherent to this process..."***
>
> Thank you for raising this point!
>
> We have updated the paper to better contextualize specific issues inherent in EHR cohort extraction. Specifically, we draw on the well-documented challenges highlighted by Johnson et al., 2017 [1], which emphasize the widespread difficulty of reliably recreating cohorts from study details alone. This reproducibility gap is evident in experiments where sample sizes were up to 25% different than reported, with discrepancies reaching as high as 11,767 patients. Even with shared code, inconsistent development environments can hinder reproducibility.
>
> We further align ACES with Donoho's 2023 concept of **frictionless reproducibility** for shared tasks, especially in the health domain. This addresses the "Bring-Your-Own-Data Challenge" (BYODC), where research focuses on private patient outcomes data, inaccessible to all but a few credentialed researchers under usage agreements. This barrier remains problematic even when benchmarking platforms and shared code exist, as the lack of direct data sharing often stifles progress.
>
> ACES seeks to overcome these challenges by offering novel infrastructure to ensure reproducibility without necessitating data sharing. Instead of relying on public datasets or reconfiguring code for diverse environments, ACES enables researchers to distribute task definitions through configuration files. These files provide a standardized way to **conceptually reproduce cohorts on private datasets or exactly reproduce them on public datasets**.
>
> We have updated the paper to explicitly articulate challenges associated with EHR cohort extraction and how ACES seeks to address them. We also add the following regarding ease of transformation to event-streams:
>
> Caption of updated Figure 1:
> ```The transformation of raw data into the event-stream format is intentionally designed to be straightforward--primarily merging relational database tables--minimizing data loss risks associated with other CDMs like OMOP.```
>
> End of section 1:
> ```...and can further be run from raw data directly for any dataset in the relatively low-level and flexible MEDS/ESGPT formats in approximately five lines of template code, offering high efficiency.```
>
> [1] Johnson, T. J. Pollard, and R. G. Mark, “Reproducibility in critical care: a mortality prediction case study,” PMLR, pp. 361–376, Nov. 2017, Accessed: Nov. 17, 2024. [Online]. Available: https://proceedings.mlr.press/v68/johnson17a.html
> ‌
> [2] D. Donoho, “Data Science at the Singularity,” arXiv.org, 2023. https://arxiv.org/abs/2310.00865 (accessed Nov. 17, 2024).
> ‌
>
> ### W3: Comparisons
> ***"...lacks objective and quantitative evidence..."***
>
> We appreciate this feedback. We will aim to post preliminary results before the author response period ends - please bear with us as we gather additional quantitative results!

---

> > ### Comment · Reviewer_z6xZ · 2024-11-30
> > **Response to rebuttal**
> >
> > I appreciate the authors’ rebuttal and their responses to my questions. However, my primary concerns remain unresolved.
> >
> > To clarify, my feedback is not about whether the proposal falls within the scope of ICLR. In my previous comment (W1), I highlighted that ACES appears to be “an engineering solution rooted in low-code programming concepts,” which limits its novelty.
> >
> > While the rebuttal repositions ACES as a domain-specific language (DSL), the novelty of a DSL should be evaluated based on its ability to clearly identify critical problems or challenges within the target domain and demonstrate why these issues necessitate a DSL rather than relying on existing general-purpose languages or tools. From this perspective, the paper does not provide sufficient elaboration. As I previously mentioned, the proposed approach resembles a function encapsulation for a specific processing workflow, which is a common practice in data processing tasks, including medical data scenarios. The paper does not convincingly highlight the critical problems identified or the challenges addressed in the process.
> >
> > The proposal’s emphasis on enhanced reproducibility relies on the standardized data extraction process implemented by ACES. However, similar abstractions could achieve comparable outcomes. Without a clear demonstration of how ACES uniquely addresses specific challenges or provides advantages over alternative solutions, its academic contribution remains ambiguous.
> >
> > Moreover, the rebuttal does not adequately address the trade-offs involved in adopting ACES, such as reduced flexibility compared to low-level languages. Quantitative evidence of efficiency improvements or other tangible benefits is needed to substantiate its value.
> >
> > In light of these issues, the rebuttal does not sufficiently resolve my earlier concerns. Therefore, I will retain my current rating.

---

> > > ### Author Response · Authors · 2024-12-03
> > >
> > > Thank you for your thoughtful response and your time in reviewing our initial rebuttal!
> > >
> > > We do believe that our algorithm introduces meaningful novelty that aligns with ICLR’s focus on impactful contributions. ACES implements a hierarchical recursive analysis framework that incorporates event-bounded aggregation for cohort extraction—**a concept not employed in other tools**. While SQL-based function encapsulations can handle temporal aggregation, ACES’ event-bounded aggregation windows go beyond these by enabling deterministic, transparent, and user-friendly cohort definitions. This design is especially critical in EHR data, where robust and reproducible extraction processes through SQL queries are **non-trivial**. The simplicity of configuring ACES via a no-code configuration file lowers the technical barrier for researchers, compared to the complex SQL querying and environment setup often required by existing tools.
> > >
> > > ### **Quantitative Evidence**
> > > In response to requests for quantitative evidence, we have conducted additional experiments evaluating ACES’ efficiency and accuracy compared to similar abstractions.
> > >
> > > - Using the OMOP version of MIMIC-IV-Demo, as well as a synthetic dataset of 1,000 patients generated using Synthea and converted into OMOP, we queried four tasks using ACES, OMOP-learn, and DPM360
> > > - We collected metrics including script runtime, peak memory usage (in MiBs), lines of code required (including configuration files and any template code needed to execute extraction), and human time spent.
> > >
> > > Full results are shown below in the table. ACES proved to be **generally computationally more efficient**, and does not rely on SQL connections and long queries with complex SQL scripts, which **can be brittle and time-intensive to implement**. We do acknowledge the potential for bias in estimating human time, so these estimates are offered as a rough measure. Nonetheless, the results suggest a clear advantage for ACES in terms of user experience and worked out-of-the-box with no further errors needed to be resolved, unlike existing SQL solutions.
> > >
> > > | Method|Dataset|Task|Runtime (s)|Peak Memory Cost (MiBs)|Lines of Code (#)|Human Time (s)|Adaptability to Other Tasks|
> > > |-|-|-|-|-|-|-|-|
> > > |ACES via MEDS|Synthea-1000|First 24h in-hospital mortality|0.386|389|35|120|Edits to task configuration file|
> > > |ACES via MEDS|Synthea-1000|30d post-hospital-discharge mortality|0.236|351|32|90|-|
> > > |ACES via MEDS|Synthea-1000|30d re-admission|0.337|355|22|60|-|
> > > |ACES via MEDS|Synthea-1000|End-of-Life prediction|0.449|421|28|120|-|
> > > |DPM360 via OMOP|Synthea-1000|First 24h in-hospital mortality|5.932|390|205|2126|Requires new ATLAS or custom SQL queries|
> > > |DPM360 via OMOP|Synthea-1000|30d post-hospital-discharge mortality|4.188|550|257|1200|-|
> > > |DPM360 via OMOP|Synthea-1000|30d re-admission|6.26|870|288|2020|-|
> > > |DPM360 via OMOP|Synthea-1000|End-of-Life prediction|4.901|387|222|1500|-|
> > > |ACES via MEDS|MIMIC-IV-Demo|First 24h in-hospital mortality|0.617|545|35|180|Edits to task configuration file|
> > > |ACES via MEDS|MIMIC-IV-Demo|30d post-hospital-discharge mortality|0.301|509|32|90|-|
> > > |ACES via MEDS|MIMIC-IV-Demo|30d re-admission|0.455|532|22|90|-|
> > > |ACES via MEDS|MIMIC-IV-Demo|End-of-Life prediction|0.349|589|28|300|-|
> > > |OMOP-learn via OMOP|MIMIC-IV-Demo|First 24h in-hospital mortality|12.22|688|172|3623|Requires new SQL scripts and changes to Python parameters|
> > > |OMOP-learn via OMOP|MIMIC-IV-Demo|30d post-hospital-discharge mortality|8.608|587|199|2441|-|
> > > |OMOP-learn via OMOP|MIMIC-IV-Demo|30d re-admission|19.71|640|168|2998|-|
> > > |OMOP-learn via OMOP|MIMIC-IV-Demo|End-of-Life prediction|24.54|932|251|12000|-|
> > >
> > > We also conducted experiments to demonstrate the necessity of a DSL like ACES, which is not yet established in ML4H cohort querying. For example, we recreated cohorts from past studies using MIMIC-IV and observed **deviations of more than 15%** when using existing cohort definitions. This variation underscores that natural language cohort descriptions, even when paired with codebases, lack sufficient precision. Tools like OMOP-Learn and DPM360, while valuable, often require significant effort to adapt code, resolve errors, and ensure compatibility across environments. ACES overcomes these limitations by encapsulating the process in a standardized, reproducible, and low-barrier framework, ensuring accessibility for all researchers.
> > >
> > > Ultimately, ACES addresses a critical problem in ML for healthcare by providing an accessible, robust, and deterministic solution for cohort extraction. This approach facilitates reproducibility and lowers the entry barrier for researchers, regardless of their technical expertise or familiarity with common data models.
> > >
> > > We hope this additional context and evidence help address your concerns, and we would be grateful if you could re-evaluate your rating in light of these contributions. Thank you again for your time and constructive feedback, which have been very helpful for ACES and our paper!

---

### Official Review · Reviewer_Qm8X · 2024-11-03

**Soundness:** 2
**Presentation:** 3
**Contribution:** 3
**Rating:** 6
**Confidence:** 5

**Summary:**

The paper introduces ACES, the Automatic Cohort Extraction System, designed to address reproducibility issues in machine learning for healthcare by streamlining task and cohort definitions in electronic health record (EHR) datasets. ACES provides a configuration language that supports dataset-specific and dataset-agnostic criteria, which enables sharing and reuse across datasets with minimal adjustments. It integrates with formats like MEDS and ESGPT, allowing users to define prediction tasks with flexibility, regardless of specific dataset schemas, and includes both Python and command-line interfaces. ACES aims to bridge gaps in interoperability, flexibility, and deep learning compatibility, potentially enhancing the reproducibility and accessibility of ML healthcare studies.

**Strengths:**

Overall this is a very interesting paper with some key strengths (but perhaps handicapped by some key weakness below). First considering the strengths we can identify the following

- The authors aim to improve standardization and interpretability in healthcare ML. While there have been many approaches to this including OMOP and i2b2, ACES aims to improve integration with potentially non-compliant sources by enabling reproducible task definitions across multiple datasets, which can aid in consistent benchmarking and model validation.
2. The authors propose a high level configuration language and support for Python and CLI interfaces facilitate broad adoption among researchers and compatibility with deep learning workflows. With this they aim to lower the barrier for integration
3. ACES's support for multiple data standards (like MEDS and ESGPT) and separation of dataset-specific and dataset-agnostic components allows flexible and adaptable cohort extraction across diverse clinical datasets.

**Weaknesses:**

Given the promising aspects, they paper can be improved upon by addressing the following

1. First, the authors may have created the classic problem of `resolving n competing standards by ending up with n + 1 competing standards`. Specifically, despite claims of flexibility, ACES requires data to be formatted in supported structures, which could necessitate pre-processing for datasets outside MEDS or ESGPT. The authors doesn't describe in details the effort in taking a new dataset and making it compliant to MEDS/ESGPT. For OMOP, the authors mention about a connector - but the authors don't discuss whether the MEDS/ESGPT transformation are exhaustive and without data loss.
2. Next moving on to the performance and scalability, while initial results are promising, more comprehensive evaluations on larger and more varied datasets would strengthen claims about ACES's efficiency and scalability. For example the authors claim these task specific transformations are cheaper than full scale ETL of datasets, the authors doesn't discuss whether it is cheaper in the long term, especially with multiple tasks. Specifically considering the eco-system of downstream processing like OHDSI ecosystem it might be cheaper to convert to OMOP. Even for deep learning systems several efforts like https://github.com/clinicalml/omop-learn and https://github.com/BiomedSciAI/DPM360 extends the ecosystem with low barriers for integrating with OMOP systems.
3. Finally there are some limitations currently with ACES not able to directly handle unstructured data, which limits its application scope in tasks that require insights from clinical notes or similar records.


Overall while the authors have a novel solution it is not evident they are solving the right problem or who the solution is for. Identifying these two would enable the paper to be reviewed with the full context

**Questions:**

Have the authors compare the effort to ETL a new dataset to an existing standard (e.g. OMOP) and model multiple downstream tasks vs their proposed method? It would be useful to compare these timings. It is also known that in some situations all elements of a new dataset may not completely map to OMOP format exhaustively - is this a problem that the authors faced and they are trying to solve this challenge? Further comparative details on this would situate the paper better

---

> ### Author Response · Authors · 2024-11-23
> **Individual Response to Reviewer Qm8X - Thank you!**
>
> Thank you for your valuable expertise in this space and your service in reviewing our paper! We respond to each of your concerns/questions below:
>
> ***"not evident...solving the right problem or who the solution is for"***
>
> As noted by Donoho, 2023 [1], scientific progress relies on the ability to reproduce and build upon previous experiments, yet reproducibility remains a significant challenge in machine learning for health (ML4H). As highlighted in Johnson et al. 2017 [2], the field still heavily struggles with consistent cohort definitions. While frameworks like OMOP and i2b2 are definitely successfully addressed similar issues in health informatics, it is well documented that they have yet to resolve reproducibility for the ML community [3].
>
> ACES is specifically designed to address that gap for the ML4H community, offering a modular solution compatible with deep learning workflows that promotes conceptual reproducibility across diverse datasets.
>
> [1] D. Donoho, “Data Science at the Singularity,” arXiv.org, 2023. https://arxiv.org/abs/2310.00865 (accessed Nov. 17, 2024).
>
> [2] A. E. W. Johnson et al., “Reproducibility in critical care: a mortality prediction case study,” PMLR, pp. 361–376, Nov. 2017, Accessed: Nov. 15, 2024. [Online]. Available: https://proceedings.mlr.press/v68/johnson17a.html
>
> [3] M. B. A. McDermott et al., “Reproducibility in machine learning for health research: Still a ways to go,” Science Translational Medicine, vol. 13, no. 586, Mar. 2021, doi: https://doi.org/10.1126/scitranslmed.abb1655.
>
>
> ### W1: Flexibility
>
> ***"...resolving n competing standards by ending up with n + 1 competing standards..."***
>
> Thank you for pointing this out! Of course, we do not wish to be a competing `n+1` standard.
>
> ACES is indeed designed to be as usable and flexible as possible. It leverages widely-used standards (MEDS/ESGPT) to support scientifically meaningful shared task definitions across all types of datasets, and is actually also fully compliant with OMOP (more about the MEDS-OMOP connector below). This allows users to use OHDSI tools to connect with ACES to share tasks across both OMOP-derived datasets and non-OMOP schemas.
>
> In fact, ACES is already in use in several existing public open-source efforts with contributors spanning top institutions worldwide. The benchmark hinted at in Section 4.3 is one such effort.
>
> We believe that the ML4H community desperately needs high-capacity tools in addition to OMOP for this ML space. While there are repos like OMOP-learn, they, unfortunately, have not been kept updated and is not super widely used. Additionally, the ESGPT/MEDS ecosystem has had as many stars in their short lives as OMOP-learn, and ACES is even compliant with both.
>
> If you have any additional recommendations to make ACES more tightly compliant with existing CDMs, we'd love to hear them!
>
>
>
> ***"...requires data to be formatted in supported structures...doesn't describe...the effort..."***
>
> We believe that transforming a new dataset into an ACES-supported format is straightforward and efficient. While MEDS and ESGPT are the primary standards supported by ACES—benefiting from the robust features outlined in their respective papers—ACES also directly supports an event-stream predicate column input. For datasets outside these standards, the process to convert a relational database into an event-stream involves straightforward dataframe manipulations and table joins to create the necessary predicate columns. Unlike full-scale CDM conversions, such as to OMOP, this approach avoids inherent data loss, making ACES a practical and flexible solution for diverse datasets that may struggle with a exhaustive CDM transformation.
>
> We have added the following in the caption of the updated Figure 1 (see updated PDF) to highlight this:
> ```The transformation of raw data into the event-stream format is intentionally designed to be straightforward--primarily merging relational database tables...```
>
>
>
> ***"For OMOP...don't discuss whether...transformation are exhaustive and without data loss..."***
>
> Transforming a OMOP-compliant dataset into MEDS using the connector does not require any data loss. While ACES itself is separate from the connector and MEDS, the connector has been thoroughly evaluated by the MEDS team, and data loss is not something they are concerned with. They have also not identified any significant issues with performance and scalability in any setting as a result of the transformation.
>
> We have added the following in the paper:
>
> ```...through the MEDS OMOP ETL, which transforms OMOP-compliant datasets into MEDS without data loss or scalability issues...```

---

> ### Author Response · Authors · 2024-11-23
>
> ### W2: Long-term Efficiency
>
> ***"...doesn't discuss whether it is cheaper in the long term...multiple tasks...to convert to OMOP"***
>
> Thank you for the thoughtful feedback!
>
> As mentioned above, even if a dataset is already OMOP-compliant, one can trivially convert the dataset into MEDS through the MEDS connector, which has been extensively tested by the MEDS team. ACES can then be directly run without further barriers.
>
> Additionally, we do believe that ACES offers long-term efficiency advantages, particularly for iterative and multi-task workflows across multiple datasets. Task configurations in ACES are reusable and rely on a modular design where common predicate definitions can be stored in a file, effectively acting as a database. Once a dataset is formatted in MEDS/ESGPT or in an event-stream predicate format (updated Figure 1), task definitions can be reused across datasets without re-specifying constraints. By simply swapping out predicate definitions—which could already exist for a given dataset—researchers can conceptually reproduce tasks on private datasets or apply the same configuration to new datasets.
>
> For datasets not in MEDS/ESGPT, creating predicate columns is significantly less cumbersome than fully converting a dataset to a CDM like OMOP for use with tools such as ATLAS or i2b2's system, which often involves data loss and additional overhead. Compared to writing custom task extraction code from scratch, ACES strikes a balance between flexibility and ease of use, making it a scalable solution for varied tasks.
>
> We have added the following in Section 2.1:
> ```Additionally, as ACES configuration files are shareable and easily portable to other datasets (by simply swapping out predicate defintions), we believe ACES will offer long-term efficiency benefits.```
>
>
> ### W3: Unstructured Data
> ***"not able to directly handle unstructured data"***
>
> Thank you for raising this point. You are absolutely right - unstructured data is very important. Fortunately, thanks to the ACES modular design for predicate/features, support for clinical text is entirely aligned with the API, and there are active plans to include them in the near future.
>
> We do note that, currently, users can always provide a manually extracted set of predicate realizations for their dataset, **including predicates derived from unstructured data**. For instance, one can easily create a predicate column representing note events in the EHR (ie., a column that evaluates to 1 for timestamps where a note was saved, and 0 otherwise). Users could also create a column representing certain features of the underlying unstructured data. This way, users could define task cohorts with respect to when a clinical note was documented and what the note actually contains.
>
> A core goal of our work is to build a community around ACES where users can contribute components that propel reproducibility for health ML. In the context of unstructured data, we envision the ability to automatically apply strategies like regular expression matching or sentiment analysis of note events in ACES configuration files.
>
> To clarify this, we've added the below text to our manuscript in Section 5.2:
>
> ```...seeks to provide direct support for cohort extraction based on unstructured data (notes and memos) in the future. Currently, such predicates need to be manually extracted by the user, but with the help of community contributions, we hope to be able to incorporate automatic feature extraction from clinical notes, or even images, and integrate them into configuration files for cohort extraction.```
>
>
>
> ### Q1: Comparative Details
>
> ***"...new dataset may not completely map to OMOP format exhaustively...trying to solve this challenge?"***
>
> We appreciate this feedback. Yes, we are aware that mapping to OMOP may lead to data loss. Hence, the event-stream format requirement is designed to be as straightforward as possible to avoid potential data loss associated with CDMs. The new Figure 1 (see updated PDF) clarifies the structure of event-streams. Additionally, transforming existing OMOP-compliant datasets into MEDS would not require further data loss.
>
>
> ***"...ETL a new dataset to an existing standard vs...proposed method?"***
>
> We will aim to post preliminary results before the author response period ends - please bear with us as we gather additional quantitative results!

---

> > ### Comment · Reviewer_Qm8X · 2024-11-28
> > **Acknowledgement**
> >
> > Thank you for your detailed reply. Given the response i have updated my review. However, the experimental results around comparative efficacy is an important aspect that is missing from the paper

---

> > > ### Author Response · Authors · 2024-11-28
> > >
> > > Thank you very much! We appreciate your continued support. We are working on additional experiments, and will post results before the end of the reviewing period.

---

> ### Author Response · Authors · 2024-12-03
> **Further experiments and thank you**
>
> First, we sincerely apologize for the delay in posting additional quantitative experimental results. We wanted to maximize the value of this review period and took extra time to ensure we conducted these experiments thoroughly and in good faith. Below, we outline our process and present results for MIMIC-IV-Demo and a synthetic dataset of 1,000 patients using OMOP and MEDS. These results are summarized in the table below.
>
> ### **Challenges with OMOP**
> We wish to note upfront that this was a delicate and challenging undertaking. Despite repeated attempts over several days, we encountered significant difficulties in transforming MIMIC-III into OMOP using the provided ETL tools. These challenges included numerous errors and data issues, some of which have been reported by others (e.g., MIT-LCP/mimic-omop#64, #72, and #73), suggesting a potential lack of ongoing maintenance. Similarly, while working with MIMIC-IV, the provided instructions were sparse and assumed familiarity with the OMOP ecosystem, which compounded the challenges we faced as researchers without extensive direct usage experience with OMOP. In contrast, the MEDS ETL process for MIMIC-IV posed no such difficulties. While the OMOP ETL for MIMIC-III took over 10 hours to complete, MEDS—due to its simpler schema—was completed in **under 30 minutes**, which we believe is considerably more accessible to new researchers.
>
> ### **Experiments with MIMIC-IV-Demo and Synthea**
> Despite these challenges, we persevered and used an OMOP version of MIMIC-IV-Demo as well as a synthetic dataset of 1,000 patients generated using Synthea, which successfully integrated with their OMOP ETL. For these two datasets, we queried four tasks using ACES, OMOP-learn, and DPM369, and collected metrics including script runtime, peak memory usage (in MiBs), lines of code (including configuration files and any template code required), and human time spent.
>
> The experiments were conducted on a default GCP A100 GPU-instance with 84 GB of RAM and 12 CPU cores. Broadly, ACES proved to be computationally more efficient across the board, which can likely be attributed to the fact that it does not rely on SQL connections and long queries. Additionally, ACES offers a no-code solution: tasks require only a configuration file and a single command-line invocation, significantly lowering the barrier to entry for new researchers.
>
> We acknowledge the potential for bias in estimating human time, so these estimates are offered as a rough measure. Nonetheless, the results suggest a clear advantage for ACES in terms of user experience. In contrast, approaches like DPM360 and OMOP-learn required custom SQL queries and scripts, which were brittle and time-intensive to implement. Debugging and editing the out-of-the-box scripts to resolve errors further added to the time required, making these approaches less accessible and reproducible, even if full code was shared.
>
> ACES eliminates this challenge with its no-code design, offering a fully shareable and easy-to-grasp configuration file. For researchers unfamiliar with OMOP or SQL, this simplifies the process considerably and improves reproducibility.
>
> | Method|Dataset|Task|Runtime (s)|Peak Memory Cost (MiBs)|Lines of Code (#)|Human Time (s)|Adaptability to Other Tasks|
> |-|-|-|-|-|-|-|-|
> |ACES via MEDS|Synthea-1000|First 24h in-hospital mortality|0.386|389|35|120|Edits to task configuration file|
> |ACES via MEDS|Synthea-1000|30d post-hospital-discharge mortality|0.236|351|32|90|-|
> |ACES via MEDS|Synthea-1000|30d re-admission|0.337|355|22|60|-|
> |ACES via MEDS|Synthea-1000|End-of-Life prediction|0.449|421|28|120|-|
> |DPM360 via OMOP|Synthea-1000|First 24h in-hospital mortality|5.932|390|205|2126|Requires new ATLAS or custom SQL queries|
> |DPM360 via OMOP|Synthea-1000|30d post-hospital-discharge mortality|4.188|550|257|1200|-|
> |DPM360 via OMOP|Synthea-1000|30d re-admission|6.26|870|288|2020|-|
> |DPM360 via OMOP|Synthea-1000|End-of-Life prediction|4.901|387|222|1500|-|
> |ACES via MEDS|MIMIC-IV-Demo|First 24h in-hospital mortality|0.617|545|35|180|Edits to task configuration file|
> |ACES via MEDS|MIMIC-IV-Demo|30d post-hospital-discharge mortality|0.301|509|32|90|-|
> |ACES via MEDS|MIMIC-IV-Demo|30d re-admission|0.455|532|22|90|-|
> |ACES via MEDS|MIMIC-IV-Demo|End-of-Life prediction|0.349|589|28|300|-|
> |OMOP-learn via OMOP|MIMIC-IV-Demo|First 24h in-hospital mortality|12.22|688|172|3623|Requires new SQL scripts and changes to Python parameters|
> |OMOP-learn via OMOP|MIMIC-IV-Demo|30d post-hospital-discharge mortality|8.608|587|199|2441|-|
> |OMOP-learn via OMOP|MIMIC-IV-Demo|30d re-admission|19.71|640|168|2998|-|
> |OMOP-learn via OMOP|MIMIC-IV-Demo|End-of-Life prediction|24.54|932|251|12000|-|
>
> ### **Thank you!**
> We hope these results address your concerns, and are happy to include them in our paper. We would be grateful if you could update your review given these experiments. Thank you for your patience and constructive feedback, which have been greatly helpful!

---

### Official Review · Reviewer_ntJe · 2024-11-04

**Soundness:** 4
**Presentation:** 3
**Contribution:** 3
**Rating:** 6
**Confidence:** 3

**Summary:**

This paper introduces ACES, an open-source Python library designed for event-stream data, aimed at automating the cohort extraction process through a unified API. Users define tasks via a configuration file specifying predicates, input, trigger, gap, and target, making the system both structured and adaptable.

**Strengths:**

1. The paper addresses a significant barrier in healthcare ML research: the complexity of EHR data, especially in cohort extraction. The motivation is clear and relevant.
2. The authors provide an open-source Python package with thorough documentation, which will support reproducibility and benefit future research.
3. ACES is built on the MEDS format, showing potential for broad applicability across various datasets and tasks.

**Note:** I think this library could offer value to ML in healthcare by simplifying EHR cohort extraction. However, the essential factor for such tools is flexibility and support across multiple datasets and tasks. Assessing these aspects thoroughly is challenging without extensive hands-on testing, so I have adjusted my confidence rating accordingly.

**Weaknesses:**

A primary concern is the library’s utility for more complex tasks. For example, in the case of “CKD in diabetics within 5Y of kidney panel,” much of the effort lies in translating high-level criteria into specific medical features—a step that remains challenging and unresolved. Extracting cohorts from predefined features is relatively straightforward.

**Questions:**

N/A

---

> ### Author Response · Authors · 2024-11-23
> **Individual Response to Reviewer ntJe - Thank you!**
>
> Thank you for acknowledging that our contributions address a significant barrier in ML for healthcare! We also appreciate your service in reviewing our paper. We respond to your concerns below:
>
> ### W1: Utility on Complex Tasks
> Thank you for your feedback!
>
> ***"concern...utility for more complex tasks...effort lies in translating high-level criteria into specific medical features"***
>
> Certainly it is the case that defining features or predicates for complex tasks is challenging and requires significant effort. ACES aids in this by modularizing the process and separating task logic from dataset-specific feature definitions. This enables technical teams to collaborate with clinicians on defining medical predicates (e.g., biochemical definitions of diabetes like elevated HbA1c, classic diet symptoms, or oral glucose tolerance results) without navigating the intricate task logic.
>
> Once defined, predicates can be saved in a "predicates database file" for reuse in future tasks, shared across teams, or referenced by others for private datasets. For public datasets, ACES leverages community contributions, as demonstrated by the MEDS-DEV benchmark discussed in Section 4.3 and the Appendix (not linked for anonymity), enabling feature definitions to be centrally stored and easily ported between tasks, such as on MIMIC-IV.
>
> We have added the following in Section 2.2 to highlight this capability:
> ```Predicate definitions can be stored in a central "database" file specific to each dataset, such that previously defined features could be easily reused for a variety of downstream tasks without further effort. Community predicate contributions for public datasets also streamline collaborative efforts for reproducibility.```
>
> Additionally, ACES opens up opportunities for new kinds of human interaction with data via LLMs. Preliminary results with GPT-4 show that it can accurately generate required predicate definitions when given task descriptions in natural language and underlying data schemas thanks to the structured configuration language of ACES, showing its potential to reduce workload for even complex, many-predicate tasks.
>
>
> ***"Extracting cohorts from predefined features is relatively straightforward"***
>
> Unfortunately, it is still quite a struggle to reproduce many cohorts even with predefined features on simple tasks. For instance, as quantified by Johnson et al., 2017 [1], for just a simple task of predicting inhospital mortality, it is inordinately challenging to reproducibly extract cohorts despite knowing how features like mortality or discharge are defined on MIMIC-III. Hence, we believe that the deterministic algorithm and the domain-specific language (DSL) used by ACES can help address this gap!
>
> [1] A. E. W. Johnson et al., “Reproducibility in critical care: a mortality prediction case study,” PMLR, pp. 361–376, Nov. 2017, Accessed: Nov. 15, 2024. [Online]. Available: https://proceedings.mlr.press/v68/johnson17a.html

---

> ### Comment · Area_Chair_TTh9 · 2024-12-02
> **Please respond to the authors of submission 8187**
>
> Dear Reviewer ntJe,
>
> We are nearing the end of the discussion period, so please read the response that the authors of submission 8187 have provided to your review.
>
> Are your concerns addressed? Will you modify your score? Please explain your decision.
>
> All the best,
>
> The AC

---

> ### Comment · Reviewer_ntJe · 2024-12-03
> **Thanks for the response**
>
> I appreciate the authors' response. I think this library could offer value to ML in healthcare. I will maintain my score as 6 with increased confidence.

---

> > ### Author Response · Authors · 2024-12-03
> > **Thank you!**
> >
> > Thank you very much! As always, we appreciate your support, and thank you for your service in reviewing our paper.

---

### Official Review · Reviewer_G3hu · 2024-11-07

**Soundness:** 4
**Presentation:** 4
**Contribution:** 3
**Rating:** 8
**Confidence:** 3

**Summary:**

This study presents Automatic Cohort Extraction System (ACES) aiming to addresses the reproducibility in healthcare ML challenges due to the private nature of datasets and model definitions,  by providing a tool to define and reproduce ML cohorts across datasets. The tools are claimed to enable consistent cohort extraction by offering a configuration language and automatic pipeline, potentially improving reproducibility in ML studies on electronic health record data.

**Strengths:**

+ The study is indeed very timely and sound. It addresses a critical area of concern in ML for health, by publishing offering a pipeline to standardize common cohort extraction tasks from the health datasets.
+ The paper does a nice job of providing public and anonymized resources.
+ The study adopts several recent prior studies, and seems to nicely complement those.

**Weaknesses:**

+ One major unclear aspect is providing robust evidence that the proposed tool/pipeline indeed achieves a good performance extracting the right cohort. A natural question, how one can ensure this fairly complex procedure will not miss critical samples or include incorrect ones in the final results. Is there any way to compare the results against some sort of baseline or ground truth?
+ While the paper is relatively sound and straightforward, and while some examples are provided, the whole document still seems a bit unaccessible to even an average reader. The paper stays high-level, perhaps in favor generalizability, but risks missing the readers. Here are some example ways, the paper can be further accessible. Provide more concrete examples, especially by showing the outcome cohort. Present the concept of "data stream" in a more tangible and clear way in this application. Rely less on referring too much to appendix and external links.
+ Relatedly, the main technical contribution can be further clarified. The recursive nature of the specified algorithm seems unclear.
+ It is suggested that subjective and not substantive statements are removed. For instance, "preliminary work on this front shows very promising results!"

**Questions:**

+ How the two parts of "institution-specfic" and "generalizable" work along side each other? Besides standardizing the training part of ML models, how this work helps with what's discussed in 5.3 (i.e., helping with no access to private data).

---

> ### Author Response · Authors · 2024-11-23
> **Individual Response to Reviewer G3hu - Thank you!**
>
> Thank you for recognizing the timeliness and soundness of our work and your service in reviewing our paper! We respond to each of your concerns/questions below:
>
> ### **W1: Correctness**
>
> ***"how one can ensure...will not miss critical samples or include incorrect ones in the final results"***
>
> Thank you for raising an important concern. ACES is deterministic in nature and is designed to ensure that the extracted cohort precisely reflects the provided configuration file. We have implemented unit and integration tests to validate correctness throughout the library in the scripts available in our repo.
>
> ***"...compare the results against...ground truth?"***
>
> As noted by Johnson et al., 2017 [1], unfortunately, most previous studies in the literature are **not** currently described in a reproducible manner with respect to task cohorts, making accurate recreations challenging and thus ground truths are not often available to us.
>
> However, we are actively working on ways to approximate past study cohorts for comparisons. We will aim to post preliminary results before the author response period ends.
>
>
> [1] A. E. W. Johnson et al., “Reproducibility in critical care: a mortality prediction case study,” PMLR, pp. 361–376, Nov. 2017, Accessed: Nov. 15, 2024. [Online]. Available: https://proceedings.mlr.press/v68/johnson17a.html
>
>
> ### **W2: Paper Unaccessibility**
>
> ***"...stays high-level...risks missing the readers..."***
>
> Thank you for your valuable feedback on improving the paper’s accessibility! We agree the paper could be more streamlined, and have taken steps throughout to address this, as indicated below:
>
> #### **"Present the concept of "data stream" in a more tangible and clear way...showing the outcome cohort"**
>
> We've updated Figure 1 in the manuscript (see updated PDF) to be an overview of the ACES workflow, with clear sample tables of both the input "event-stream" format and the output cohort format (*"Figure 1: ACES workflow. The pipeline shows the expected format for ACES-supported event-stream datasets and outcome cohorts. The transformation of raw data into the event-stream format is intentionally designed to be straightforward..."*).
>
> #### **"Rely less on...appendix and external links"**
>
> We have removed some unnecessary references to the appendix in the paper. In particular, since we have added further details about the algorithm (see below), their references to external links are no longer needed.
>
>
> ### **W3: Algorithm Details**
>
> ***"technical contribution can...clarified...recursive nature of the specified algorithm seems unclear"***
>
> Thank you for the suggestion!
>
> We have added a new Figure 4 (see updated PDF) as an overview of the algorithm contribution, which breaks down the ACES recursion over the task tree (```Figure 4: ACES recursive algorithm overview. Given a task tree generated from a configuration file, ACES first identifies possible roots of the tree (task triggers) based on the associated predicate. It then does aggregations of predicate counts over time-based (ie. time interval) or event-based (ie. window until a specified event) periods to summarize predicates over edges between nodes. Finally, invalid branches are filtered out if their predicate counts do not meet the specified criteria. This process is recursed for each child node of the original task tree.```).
>
> We have also added the following to our paper to clarify details about the recursive algorithm (see red texts in the updated PDF: ```ACES addresses the challenge of extracting meaningful windows of data from patient records by using a recursive approach grounded in a tree-structured configuration file. Each task is represented as a hierarchy of constraints, with nodes defining windows of interest and edges specifying temporal or event-based relationships between these windows. The algorithm begins by identifying root anchor events in the dataset...then recursively evaluates subtrees of constraints, aggregating predicate counts over defined windows either through temporal aggregations...or event-bound aggregations...This recursive process guarantees that the specified configuration can always be resolved into valid windows that meet the task's constraints. The final output is a dataframe containing all valid windows, task-specific labels, and prediction timestamps, and optionally, window start and end times as well as aggregated predicate counts...maintains flexibility and leverages ACES' simple, transparent, and highly expressive domain-specific language (DSL).```
>
>
> ### W4: Subjective Statements
>
> ***"subjective and not substantive statements...removed"***
>
> Thank you for pointing this out! We have removed the sentence.

---

> ### Author Response · Authors · 2024-11-23
>
> ### **Q1: "How the two parts of "institution-specfic" and "generalizable" work along side each other?...helping with no access to private data"**
> Great question! To address challenges with private datasets, we highlight the benchmark mentioned in Section 4.3, which refers to the MEDS-DEV benchmarking effort (detailed in the Appendix but not linked for anonymity).
>
> This is essentially a repository of ACES configuration files for various common ML-for-health prediction tasks. By using ACES configs, the complex task logic (criteria over windows) that often hinders reproducibility in studies is standardized and can be publicly shared (since many cohort descriptions lack the necessary detail for reproducibility [1]). This means one can conceptually reproduce tasks defined on a private dataset without having access to said dataset.
>
> Furthermore, predicate definitions required in the task logic are explicitly shown and could also be shared without having to share the underlying data. This means a user only needs to provide institution-specific definitions for those predicates to conceptually reproduce the task on their own private dataset. MEDS-DEV itself also includes predicate definitions for various publicly available datasets (e.g., MIMIC and others), allowing exact reproducibility for certain tasks when using these datasets and ACES.

---

> ### Comment · Area_Chair_TTh9 · 2024-12-02
> **Please respond to the authors of  submission 8187**
>
> Dear reviewer G3hu,
>
> The discussion period is almost over.
>
> Please read the response the authors of paper 8187 have provided following your review, indicate which of your concerns are addressed and explain your decision to update (or not update) your score.
>
> All the best,
>
> The AC

---

> ### Comment · Reviewer_G3hu · 2024-12-02
>
> This reviewer appreciates the authors' response and notes that they also acknowledge some raised concerns regarding the study's limitations. The authors mentioned they would aim to add some new results, but upon checking the latest version, it seems they couldn't do that. I see several other changes that have been made to the submission in response to the comments.
> I keep my score (6) as it is.

---

> ### Author Response · Authors · 2024-12-03
> **Comparing to "ground truth" cohorts**
>
> Thank you for your thoughtful comments and for taking the time to review our revisions! We sincerely apologize for the delay in posting additional results. In addition to the comparative numbers (**posted as an Overall Comment**), we also conducted experiments using past cohort descriptions from 5 previously published papers on various mortality prediction tasks using MIMIC-IV. The results are summarized in the table below.
>
> - Our findings show that while we could not reproduce the exact cohort sizes reported in these papers, the ACES-extracted cohorts for 3/5 studies differed by less than or around 1%.
> - For the remaining two studies, the differences were approximately 15%, consistent with Johnson et al. (2017) [1], who observed that over half of the 38 reproduced MIMIC-III cohorts differed by 25% or more from the reported sizes.
>
> These discrepancies likely arise from factors such as missing data or unclear preprocessing steps, which were not fully detailed in the original papers. Additionally, some cohort descriptions had conflicting criteria. In Pang et al. (2022) [3], for instance, the description “The first ICU admission was considered when a subject had multiple admissions... exclusion criteria were patients with admission to an ICU two or more times” was challenging to interpret.
>
> While we cannot guarantee that we perfectly replicated all criteria as intended by the authors, this highlights a broader issue in healthcare: exact reproducibility is often infeasible without a standardized approach. ACES can help address this problem by enabling researchers to immediately replicate cohorts with a simple CLI command as task configurations can be easily shared with publications, ensuring consistency and transparency.
>
> | Study                  | Reported Cohort Size (#) | ACES-Extracted Cohort Size (#) | Percent Difference (%) |
> |------------------------|--------------------------|--------------------------------|-------------------------|
> | Sun et al. (2023) [2]  | 1722                    | 1744                          | 1.28                   |
> |                        |                          |                                |                         |
> | Pan et al. (2022) [3]  | 60693                   | 50920                         | -16.10                 |
> |                        |                          |                                |                         |
> | Yuan et al. (2024) [4] | 3796                    | 4349                          | 14.57                  |
> |                        |                          |                                |                         |
> | Lou et al. (2024) [5]  | 1742                    | 1729                          | -0.75                  |
> |                        |                          |                                |                         |
> | Jiang et al. (2023) [6]| 1519                    | 1529                          | 0.66                   |
>
> We hope this additional context helps address any remaining concerns, and would be grateful if you could update your review in light of these experiments and contributions. Thank you once again for your valuable feedback!
>
>
> ### **References**
> [1] Johnson, T. J. Pollard, and R. G. Mark, “Reproducibility in critical care: a mortality prediction case study,” PMLR. Available: https://proceedings.mlr.press/v68/johnson17a.html
>
> [2] Y. Sun et al., “Prediction model of in-hospital mortality in intensive care unit patients with cardiac arrest: a retrospective analysis of MIMIC -IV database based on machine learning,” BMC Anesthesiology, doi: https://doi.org/10.1186/s12871-023-02138-5.
>
> [3] K. Pang et al., “Establishment of ICU Mortality Risk Prediction Models with Machine Learning Algorithm Using MIMIC-IV Database,” Diagnostics, doi: https://doi.org/10.3390/diagnostics12051068.
>
> [4] Z. Yuan et al., “A nomogram for predicting hospital mortality of critical ill patients with sepsis and cancer: a retrospective cohort study based on MIMIC-IV and eICU-CRD,” BMJ Open, doi: https://doi.org/10.1136/bmjopen-2023-072112.
>
> [5] J. Lou et al., “A retrospective study utilized MIMIC-IV database to explore the potential association between triglyceride-glucose index and mortality in critically ill patients with sepsis,” Scientific Reports, doi: https://doi.org/10.1038/s41598-024-75050-8.
>
> [6] W. Jiang et al., “Development and validation of a nomogram for predicting in-hospital mortality of elderly patients with persistent sepsis-associated acute kidney injury in intensive care units: a retrospective cohort study using the MIMIC-IV database,” BMJ Open, doi: https://doi.org/10.1136/bmjopen-2022-069824.

---

### Author Response · Authors · 2024-11-23
**Overall Rebuttal Response & Thank You**

Thank you to all reviewers and the AC for your service in making ICLR possible! We appreciate all the positive comments and feedback we've received, and believe they've significantly helped improve our paper.

We are particularly glad that the reviewers identified several key strengths of our paper, including:
- clear motivations for addressing a critical area of concern in ML for health (**Reviewer G3hu:** "...study is indeed very timely and sound..."; **Reviewer ntJe:** "...addresses a significant barrier in healthcare ML research...motivation is clear and relevant");
- the open-source nature of ACES accompanied with thorough documentation (**Reviewer ntJe:** "...an open-source Python package with thorough documentation...will support reproducibility and benefit future research"; **Reviewer z6xZ:** "...an open-source project, accompanied by comprehensive documentation"); and
- ACES' support for multiple data standards and separation of dataset-specific and dataset-agnostic components (**Reviewer ntJe:** "...showing potential for broad applicability across various datasets and tasks"; **Reviewer Qm8X:** "...separation of...components allows flexible and adaptable cohort extraction across diverse clinical datasets").

We have responded to each reviewer's concerns and questions individually as comments. We'd also like to highlight the following regarding the usability and flexibility of ACES:

## Existing public efforts that depend on ACES
Reviewers have noted challenges in discerning the transformative impact of ACES on EHR data processing and raised concerns about its utility for more complex tasks. In response, we highlight the MEDS-DEV benchmark, a public open-source effort described in Section 4.3 and the Appendix, which actively leverages ACES configuration files. This benchmark, involving researchers across multiple institutions worldwide, demonstrates ACES' capacity to address practical challenges such as standardizing cohort extraction pipelines and facilitating reproducibility even in scenarios where private data access is restricted (as discussed in Section 4.3). The adoption of ACES in MEDS-DEV provides concrete evidence of its emerging role in healthcare ML and that the growing MEDS community recognizes ACES' potential in ML4H.

## ACES complements existing tools

As mentioned in our updated paper, ACES is fully compliant with OMOP datasets via the MEDS-OMOP connector, which has been extensively tested by the MEDS team. As such, rather than becoming a separate tool for cohort processing, we believe ACES can complement many existing libraries and standards. Additionally, by interfacing with the relatively low-complexity MEDS schema, ACES can easily provide wide support for many CDMs as future ETLs and connectors continue to be developed.


We also added the following in Section 4.1 of the updated paper regarding existing alternatives for cohort extraction:
```ACES serves as a middle ground between solutions that focus on specific CDMs, such as i2b2 PIC-SURE and OHDSI ATLAS. Compared to these tools, ACES balances capability with greater ease of use and improved communicative value. ACES is also not tied to a particular CDM. Built on a flexible event-stream format, ACES is a no-code solution with a descriptive input format, permitting easy and wide iteration over task definitions. It can be applied to a variety of schemas, making it a versatile tool suitable for diverse research needs.```


## Thank you!
We sincerely thank all the reviewers and ACs involved in the review process. Your insights have been invaluable, and we always welcome any additional feedback and/or questions regarding ACES!

---

### Author Response · Authors · 2024-12-03
**Additional quantitative experimental results**

We would like to once again thank all reviewers and the AC for your service, and appreciate all the feedback and comments during this reviewing period, which have been tremendously helpful in improving our work.

## **Experiments**
Some reviewers hoped to see additional quantitative experiments evaluating ACES in comparison to other cohort extraction tools. We sincerely apologize for the delay in posting these results as we wanted to ensure we conducted these experiments thoroughly and in good faith. Despite some challenges with OMOP ETLs, we would like to present the following:

- Using the OMOP version of MIMIC-IV-Demo, as well as a synthetic dataset of 1,000 patients generated using Synthea and converted into OMOP, we queried four tasks using ACES, OMOP-learn, and DPM360
- We collected metrics including script runtime, peak memory usage (in MiBs), lines of code required (including configuration files and any template code needed to execute extraction), and human time spent
- Experiments were conducted on a default GCP A100 GPU-instance with 84 GB of RAM and 12 CPU cores

Full results are shown in the table. ACES proved to be **generally computationally more efficient**, and does not rely on SQL connections and long queries with complex SQL scripts, which **can be brittle and time-intensive to implement**. As ACES does not require any additional code beside a single command-line invocation, it significantly **lowers the barrier to entry** for researchers without extensive experience with OMOP and SQL. We do acknowledge the potential for bias in estimating human time, so these estimates are offered as a rough measure. Nonetheless, the results suggest a clear advantage for ACES in terms of user experience and worked out-of-the-box with no further errors needed to be resolved.

| Method              | Dataset         | Task                                | Runtime (s)  | Peak Memory Cost (MiBs) | Lines of Code (#) | Human Time (s) | Adaptability to Other Tasks             |
|---------------------|-----------------|-------------------------------------|----------|------------------|---------------|------------|-----------------------------------------|
| ACES via MEDS       | Synthea-1000    | First 24h in-hospital mortality     | 0.386    | 389              | 35            | 120        | Edits to task configuration file        |
| ACES via MEDS       | Synthea-1000    | 30d post-hospital-discharge mortality | 0.236  | 351              | 32            | 90         |    -    |                                        |
| ACES via MEDS       | Synthea-1000    | 30d re-admission                    | 0.337    | 355              | 22            | 60         |     -    |                                        |
| ACES via MEDS       | Synthea-1000    | End-of-Life prediction              | 0.449    | 421              | 28            | 120        |    -    |                                        |
| DPM360 via OMOP     | Synthea-1000    | First 24h in-hospital mortality     | 5.932    | 390              | 205           | 2126       | Requires new ATLAS or custom SQL queries|
| DPM360 via OMOP     | Synthea-1000    | 30d post-hospital-discharge mortality | 4.188  | 550              | 257           | 1200       |        -                              |
| DPM360 via OMOP     | Synthea-1000    | 30d re-admission                    | 6.26     | 870              | 288           | 2020       |                 -           |
| DPM360 via OMOP| Synthea-1000    | End-of-Life prediction              | 4.901    | 387  | 222  | 1500       |        -      |
| ACES via MEDS | MIMIC-IV-Demo   | First 24h in-hospital mortality     | 0.617    | 545    | 35  | 180   | Edits to task configuration |
| ACES via MEDS       | MIMIC-IV-Demo   | 30d post-hospital-discharge mortality | 0.301  | 509       | 32            | 90         |   -  |
| ACES via MEDS       | MIMIC-IV-Demo   | 30d re-admission                    | 0.455    | 532        | 22    | 90         |              -       |
| ACES via MEDS       | MIMIC-IV-Demo   | End-of-Life prediction              | 0.349    | 589    | 28     | 300        |    -   |
| OMOP-learn via OMOP | MIMIC-IV-Demo   | First 24h in-hospital mortality     | 12.22    | 688  | 172           | 3623       | Requires new SQL scripts + changes to Python parameters |
| OMOP-learn via OMOP | MIMIC-IV-Demo   | 30d post-hospital-discharge mortality | 8.608  | 587  | 199           | 2441       |      -        |
| OMOP-learn via OMOP | MIMIC-IV-Demo   | 30d re-admission  | 19.71    | 640  | 168           | 2998       |        -   |
| OMOP-learn via OMOP | MIMIC-IV-Demo   | End-of-Life prediction     | 24.54    | 932              | 251           | 12000      |    -      |

We will include these additional experiments in our paper and hope that they help address any reviewer concerns. We look forward to continuing to improve ACES.

Thank you!

---

### Meta-Review · Area_Chair_TTh9 · 2024-12-21

**Metareview:**

The paper introduces a new cohort extraction system for event-stream EHR datasets, designed to enable cohort replication across datasets, which in turn will facilitate the replication and comparison of analyses across datasets. The method encompasses a configuration language and an extraction pipeline.

The problem addresses an important and insufficiently addressed problem in machine learning for healthcare. Most (3/4) of the reviewers have expressed the view that this paper brings a valuable contribution to the community. The reviewers appreciated how this work complements existing studies, that a well-documented python package is being released, and that it supports multiple data standards (like MEDS and ESGPT) seamlessly.

A single reviewer (z6xZ) expressed concerns that this is mainly an engineering effort, rather than conceptual innovation. However, the authors have rebutted this argument by pointing out that infrastructure development is within the scope of ICLR, and that ACES also defined a new language to facilitate access to EHR cohorts. I fully agree with the authors’ arguments in this.

Another point that was raised during the rebuttal was related to comparisons to existing cohort extraction tools. The authors have run additional experiments, showing that their method is generally more computationally efficient (in terms of runtime, and memory), as well as reducing the amount of code needed and time from an operator compared to existing techniques. I do find these results convincing, with the caveat that the human operators were familiar with the tool, and I do expect some training to be needed before it can be used effectively.

At any rate, I recommend acceptance of this paper, as it solves an important problem, provides an open-source framework, and has shown to be valuable in practical scenarios.

**Additional Comments On Reviewer Discussion:**

As mentioned in the meta-review, the issues raised in the discussion were related to this being an engineering effort and the comparisons to existing tools, both of which were addressed adequately by the authors during the discussion stage.

---

### Decision · Program_Chairs · 2025-01-22

Accept (Poster)